# Second-Order Optimality in Non-Convex Decentralized Optimization via Perturbed Gradient Tracking

**Isidoros Tziotis, Constantine Caramanis, Aryan Mokhtari**
Department of Electrical and Computer Engineering
The University of Texas at Austin
{isidoros_13,constantine,mokhtari}@utexas.edu

## Abstract

In this paper we study the problem of escaping from saddle points and achieving second-order optimality in a decentralized setting where a group of agents collaborate to minimize their aggregate objective function. We provide a non-asymptotic (finite-time) analysis and show that by following the idea of perturbed gradient descent, it is possible to converge to a second-order stationary point in a number of iterations which depends linearly on dimension and polynomially on the accuracy of second-order stationary point. Doing this in a communication-efficient manner requires overcoming several challenges, from identifying (first order) stationary points in a distributed manner, to adapting the perturbed gradient framework without prohibitive communication complexity. Our proposed Perturbed Decentralized Gradient Tracking (PDGT) method consists of two major stages: (i) a gradient-based step to find a first-order stationary point and (ii) a perturbed gradient descent step to escape from a first-order stationary point, if it is a saddle point with sufficient curvature. As a side benefit of our result, in the case that all saddle points are non-degenerate (strict), the proposed PDGT method finds a local minimum of the considered decentralized optimization problem in a finite number of iterations.

## 1  Introduction

Recently, we have witnessed an unprecedented increase in the amount of data that is gathered in a distributed fashion and stored over multiple agents (machines). Moreover, the advances in data-driven systems such as Internet of Things, health-care, and multi-agent robotics demand for developing machine learning frameworks that can be implemented in a distributed manner. Simultaneously, convex formulations for training machine learning tasks have been replaced by nonconvex representations such as neural networks. These rapid changes call for the development of a class of communication-efficient algorithms to solve *nonconvex decentralized* learning problems.

In this paper, we focus on a nonconvex decentralized optimization problem where a group of $m$ agents collaborate to minimize their aggregate loss function, while they are allowed to exchange information only with their neighbors. To be more precise, the agents (nodes) aim to solve

$$\min_{\mathbf{x} \in \mathbb{R}^d} f(\mathbf{x}) = \frac{1}{m} \sum_{i=1}^{m} f_i(\mathbf{x}), \tag{1}$$

where $f_i : \mathbb{R}^d \to \mathbb{R}$ is the objective function of node $i$ which is possibly nonconvex. Finding the global minimizer of this problem, even in the centralized setting where all the functions are available at a single machine, is hard. Given this hardness result, we often settle for finding a stationary point of Problem (1). There have been several lines of work on finding an approximate first-order

stationary point of this distributed problem, i.e., finding a set of local solutions $\tilde{\mathbf{x}}_1, \ldots, \tilde{\mathbf{x}}_m$ where their average $\tilde{\mathbf{x}}_{avg}$ has a small gradient norm $\|\nabla f(\tilde{\mathbf{x}}_{avg})\|$ and a small consensus error $\sum_{i=1}^{m} \|\tilde{\mathbf{x}}_i - \tilde{\mathbf{x}}_{avg}\|$. Achieving first-order optimality, however, in nonconvex settings may not lead to a satisfactory solution as it could be a poor saddle point. Therefore, finding a second-order stationary point could improve the quality of the solution. In fact, when all saddle points are non-degenerate finding a second-order stationary point implies convergence to a local-minimum, and in several problems including matrix completion [1], phase retrieval [2], and dictionary learning [3] local minima are global minima.

While convergence to a second-order stationary point for the centralized setting has been extensively studied in the recent literature, the non-asymptotic complexity analysis of finding such a point for decentralized problems (under standard smoothness assumptions) has thus far evaded solution, in part because of significant additional challenges presented by communication limitations. A major difference between the centralized and the decentralized framework lies in the exchange of information between the nodes. Exchanging Hessian information is, of course, prohibitively expensive. Furthermore, turning to approximating schemes has the potential to create catastrophic problems for the algorithm, as small errors in approximation across the nodes could lead to inconsistent updates that could reverse progress made by prior steps. Moreover, escaping from first-order stationary points requires identifying that the algorithm has reached such a point, and accomplishing even this basic step in a communication-efficient manner presents challenges.

**Contributions.** In this paper we develop a novel gradient-based method for escaping from saddle points in a decentralized setting and characterize its overall communication cost for achieving a second-order stationary point. The proposed Perturbed Decentralized Gradient Tracking (PDGT) algorithm consists of two major steps: (i) A local decentralized gradient tracking scheme to find a first-order stationary point, while maintaining consensus by averaging over neighboring iterates; (ii) A perturbed gradient tracking scheme to escape from saddle points that are non-degenerate. We show that to achieve an $(\epsilon, \gamma, \rho)$-second-order stationary point (see Definition 2) the proposed PDGT algorithm requires at most $\tilde{\Theta}\left(\max\left\{\frac{f(\mathbf{x}^0)-f^*}{(1-\sigma)^2 \min\{\epsilon^2, \rho^2\}\gamma^3}, \frac{d}{\gamma^6}\right\}\right)$ rounds of communication, where $d$ is dimension, $f(\mathbf{x}^0)$ is the initial objective function value, $f^*$ is the optimal function value, and $\sigma$ is the second largest eigenvalue of mixing matrix in terms of absolute norm which depends on the connectivity of the underlying graph. To the best of our knowledge, this result provides the first non-asymptotic guarantee for achieving second-order optimality in decentralized optimization under standard smoothness assumptions.

## 1.1 Related Work

**Centralized settings.** Convergence to a first-order stationary point for centralized settings has been extensively studied in the nonconvex literature [4–13]. A recent line of work focuses on improving these guarantees and achieving second-order optimality in a finite number of iterations. These schemes can be divided into three categories: (i) fully gradient-based methods which use the perturbation idea for escaping from saddle points once iterates reach a point with small gradient norm [14–16]; (ii) methods which utilize the eigenvector corresponding to the smallest eigenvalue of the Hessian to find an escape direction [5, 6, 17–21]; and (iii) trust-region [22, 23] and cubic regularization algorithms [24–26] which require solving a quadratic or cubic subproblem, respectively, at each iteration. These methods, however, cannot be applied to decentralized settings directly as they require access to the gradient or Hessian of the global objective function.

**First-order optimality in decentralized settings.** Recently, several iterative methods have been introduced and studied for achieving first-order optimality in decentralized settings. In particular, [27–29] show convergence to a first-order stationary point by leveraging successive convex approximation techniques and using dynamic consensus protocols. Also, a similar guarantee has been established for several well-known decentralized algorithms including distributed gradient descent [30, 31], primal-dual schemes [32–34], gradient tracking methods [35, 36], and decentralized alternating direction method of multipliers (ADMM) [37].

**Second-order optimality in decentralized settings.** Finding a second-order stationary point in a distributed setting has been studied by several works [38–41], but they all only provide asymptotic guarantees. The most related work to our submission is [42] which studies non-asymptotic convergence of stochastic gradient-based diffusion method for decentralized settings. However, the result of this work is obtained under two relatively less common assumptions. First, it requires a bounded

gradient disagreement condition which ensures that the local gradients $\nabla f_i$ are not far from the global gradient $\nabla f$ (Assumption 3 in [42]). Second, it assumes that the computed stochastic gradient near a saddle point is such that there is gradient noise present along some descent direction, spanned by the eigenvectors corresponding to the negative eigenvalues of the Hessian, i.e., stochastic gradient leads to an escape direction (Assumption 7 in [42]). Both these assumptions, and, in particular, the second one may not hold in general decentralized settings, and they both significantly simplify the analysis of escaping from saddle points. Unlike [42], the theoretical results presented here do not require assuming these restrictive conditions, and our paper provides the first non-asymptotic guarantee for achieving second-order optimality in decentralized settings, under standard smoothness assumptions. In fact, the conditions that we assume for proving our results are identical to the ones used in [15] for the analysis of perturbed gradient method in the centralized setting.

## 2   Preliminaries

The problem in (1) is defined over a set of $m$ connected agents (nodes) where each one has access to a component of the objective function. We denote the underlying undirected connectivity graph by $\mathcal{G} = \{V, E\}$, where $V = \{1, \ldots, m\}$ is the set of vertices (nodes) and $E$ is the set of edges. As this graph is undirected, if node $i$ can send information to node $j$, then the reverse communication is also possible. We call two nodes neighbors if there exists an edge between them. We further denote the neighborhood of node $i$ by $\mathcal{N}_i$, which also includes node $i$ itself.

Since the optimization variable $\mathbf{x}$ in (1) appears in each summand of the objective function, this problem is not decomposable into subproblems that can be solved simultaneously over nodes of the network. To make the objective function separable we introduce $m$ local variables $\mathbf{x}_i \in \mathbb{R}^d$, and instead of minimizing $\frac{1}{m}\sum_{i=1}^m f_i(\mathbf{x})$ in (1), we minimize the objective function $\frac{1}{m}\sum_{i=1}^m f_i(\mathbf{x}_i)$. To ensure that these two problems are equivalent, we enforce the local decision variables to be equal to each other. Since the graph is connected, this condition can be replaced by consensus among neighboring nodes, and therefore the resulting problem can be written as

$$\min_{\mathbf{\underline{x}}=[\mathbf{x}_1;\mathbf{x}_2;...;\mathbf{x}_m]\in\mathbb{R}^{md}} F(\mathbf{\underline{x}}) := \frac{1}{m}\sum_{i=1}^m f_i(\mathbf{x}_i) \qquad s.t. \quad \mathbf{x}_i = \mathbf{x}_j, \quad \forall (i,j) \in E. \qquad (2)$$

Note that in (2) we have introduced the notation $\mathbf{\underline{x}} \in \mathbb{R}^{md}$ to indicate the concatenation of all local variables $\mathbf{\underline{x}} := [\mathbf{x}_1; \mathbf{x}_2; ...; \mathbf{x}_m]$ and defined the function $F : \mathbb{R}^{md} \to \mathbb{R}$ as $F(\mathbf{\underline{x}}) := \frac{1}{m}\sum_{i=1}^m f_i(\mathbf{x}_i)$. It can be verified that $\mathbf{x}^*$ is an optimal solution of Problem (1) if and only if $\mathbf{\underline{x}}^* := [\mathbf{x}^*; \ldots; \mathbf{x}^*]$ is an optimal solution of Problem (2). In the rest of the paper, therefore, we focus on solving Problem (2) as its objective function is node-separable. We should mention that solving this problem is still challenging as the constraints of this problem are coupled.

In this paper, we only assume standard smoothness conditions for the local objective functions $f_i$ to establish our theoretical guarantees.

**Assumption 1.** *The local functions $f_i$ have Lipschitz continuous gradient with constant $L_1$, i.e., for all $i \in \{1, \ldots, m\}$ and any $\mathbf{x} \in \mathbb{R}^d$ and $\mathbf{x}' \in \mathbb{R}^d$ we have $\|\nabla f_i(\mathbf{x}) - \nabla f_i(\mathbf{x}')\| \leq L_1 \|\mathbf{x} - \mathbf{x}'\|$.*

**Assumption 2.** *The local functions $f_i$ have Lipschitz continuous Hessian with constant $L_2$, i.e., for all $i \in \{1, \ldots, m\}$ and any $\mathbf{x} \in \mathbb{R}^d$ and $\mathbf{x}' \in \mathbb{R}^d$ we have $\|\nabla^2 f_i(\mathbf{x}) - \nabla^2 f_i(\mathbf{x}')\| \leq L_2 \|\mathbf{x} - \mathbf{x}'\|$.*

The gradient Lipschitz continuity condition in Assumption 1 is customary for the analysis of gradient-based methods. The condition in Assumption 2 is also required to ensure that the function is well-behaved near its saddle stationary points.

Finding an optimal solution of (1) or (2) is hard since the local functions $f_i$ are nonconvex. Hence, we settle for finding a stationary point. In the centralized unconstrained case, a first-order stationary point of function $f$ satisfies $\|\nabla f(\hat{\mathbf{x}})\| = 0$, and an approximate $\epsilon$-first-order stationary point is defined as $\|\nabla f(\hat{\mathbf{x}})\| \leq \epsilon$. For the constrained decentralized problem in (2) the notion of first-order stationarity should address both stationarity and feasibility as we state in the following definition.

**Definition 1.** *A set of vectors $\{\hat{\mathbf{x}}_i\}_{i=1}^m$ is an $(\epsilon, \rho)$-first-order stationary point of Problem (2) if*

$$\left\| \frac{1}{m}\sum_{i=1}^m \nabla f_i(\hat{\mathbf{x}}_i) \right\| \leq \epsilon, \qquad \frac{1}{m}\sum_{i=1}^m \left\| \hat{\mathbf{x}}_i - \frac{1}{m}\sum_{j=1}^m \hat{\mathbf{x}}_j \right\| \leq \rho. \qquad (3)$$

---

**Algorithm 1:** PDGT algorithm

---

1: **Input:** $\mathbf{x}^0, \nabla f(\mathbf{x}^0), \epsilon, \gamma, \rho, \delta_1, \delta_2$

2: Set $\mathbf{x}_i = \mathbf{x}^0, \quad \mathbf{y}_i = \nabla f(\mathbf{x}^0), \quad T_1 = \tilde{\Theta}\left(\frac{f(\mathbf{x}^0)-f^*}{(1-\sigma)^2 \min\{\epsilon^2, \rho^2\}}\right), \quad T_2 = \tilde{\Theta}\left(\frac{d\log(1/\gamma\delta_2)}{\gamma^3}\right),$

     $\eta_1 = \tilde{\Theta}\left((1-\sigma)^2\right), \quad \eta_2 = \tilde{\Theta}\left(\frac{\gamma^2}{d(1-\sigma)}\right), \quad \mathcal{R} = \tilde{\Theta}\left(\gamma^{\frac{3}{2}}\right), \quad B = \tilde{\Theta}\left(\gamma^3\right);$

3: Call    $(\underline{\tilde{\mathbf{x}}})$ = PDGT Phase I $(\underline{\mathbf{x}}, \underline{\mathbf{y}}, \eta_1, T_1, \delta_1)$;

4: Call    $(\underline{\hat{\mathbf{x}}}, \hat{\mathbf{y}}, S)$ = PDGT Phase II $(\underline{\tilde{\mathbf{x}}}, \eta_2, T_2, \mathcal{R}, B)$;

5: **if** $S = 1$ **then**

6:     Return $\underline{\hat{\mathbf{x}}}$ as a second-order stationary point and stop;

7: **else**

8:     Set $\underline{\mathbf{x}} = \hat{\mathbf{x}}$, $\underline{\mathbf{y}} = \hat{\mathbf{y}}$ and go to Step 3;

9: **end if**

---

The first condition in the above definition ensures that the gradient norm is sufficiently small, while the second condition ensures that the iterates are close to their average. It can be shown that if $[\hat{\mathbf{x}}_1, \ldots, \hat{\mathbf{x}}_m]$ is an $(\epsilon, \rho)$-first-order stationary point of Problem (2), then their average $\hat{\mathbf{x}}_{avg} := \frac{1}{m}\sum_{i=1}^{m}\hat{\mathbf{x}}_i$ is an $(\epsilon + L_1\rho)$-first-order stationary point of Problem (1), i.e., $\|\frac{1}{m}\sum_{i=1}^{m}\nabla f_i(\hat{\mathbf{x}}_{avg})\| \leq \epsilon + L_1\rho$. The proof of this claim is available in the supplementary material.

The same logic holds for second-order stationary points. In the centralized case, $\mathbf{x}$ is an $(\epsilon, \gamma)$-second-order stationary point if $\|\nabla f(\hat{\mathbf{x}})\| \leq \epsilon$ and $\nabla^2 f(\hat{\mathbf{x}}) \succeq -\gamma\,\mathbf{I}$. Similarly, we define a second-order stationary point of Problem (2) with an extra condition that enforces consensus approximately.

**Definition 2.** *A set of vectors* $\{\hat{\mathbf{x}}_i\}_{i=1}^{m}$ *is an* $(\epsilon, \gamma, \rho)$-*second-order stationary point of Problem* (2) *if*

$$\left\|\frac{1}{m}\sum_{i=1}^{m}\nabla f_i(\hat{\mathbf{x}}_i)\right\| \leq \epsilon, \qquad \frac{1}{m}\sum_{i=1}^{m}\nabla^2 f_i(\hat{\mathbf{x}}_i) \succeq -\gamma\,\mathbf{I}, \qquad \frac{1}{m}\sum_{i=1}^{m}\left\|\hat{\mathbf{x}}_i - \frac{1}{m}\sum_{j=1}^{m}\hat{\mathbf{x}}_j\right\| \leq \rho. \quad (4)$$

Note that under Assumptions 1 and 2, it can be shown that if the local solutions $[\hat{\mathbf{x}}_1, \ldots, \hat{\mathbf{x}}_m]$ form an $(\epsilon, \gamma, \rho)$-second-order stationary point of Problem (2), then their average $\hat{\mathbf{x}}_{avg} := \frac{1}{m}\sum_{i=1}^{m}\hat{\mathbf{x}}_i$ is an $(\epsilon + L_1\rho, \gamma + L_2\rho)$-second-order stationary point of Problem (1), i.e., $\|\frac{1}{m}\sum_{i=1}^{m}\nabla f_i(\hat{\mathbf{x}}_{avg})\| \leq \epsilon + L_1\rho$ and $\frac{1}{m}\sum_{i=1}^{m}\nabla^2 f_i(\hat{\mathbf{x}}_{avg}) \succeq -(\gamma + L_2\rho)\,\mathbf{I}$. For proof check the supplementary material.

## 3 Perturbed Decentralized Gradient Tracking Algorithm

We now present our proposed Perturbed Decentralized Gradient Tracking (PDGT) algorithm. The PDGT method presented in Algorithm 1 can be decomposed into two phases. Phase I of our method uses the gradient tracking ideas proposed in [35,36] to show convergence to some first-order stationary point. Using this scheme for our setup, however, requires overcoming the following hurdle: The nodes do not have access to the global gradient and thus even the task of realizing that they lie close to such a point is not trivial. Moreover, the consensus error is cumulative over the graph and tracking this quantity for each node is an additional challenge. In prior work, it has been shown that there exists an iterate that achieves first-order optimality without explicitly introducing a mechanism for identifying such an iterate. In this paper, we address this issue by utilizing an average consensus protocol as a subroutine of Phase I, which coordinates the nodes and finds with high probability and negligible communication overhead the correct index achieving first-order optimality.

Phase II of PDGT utilizes ideas from centralized perturbed gradient descent developed in [15], in order to escape saddle points. Adapting these ideas to the decentralized setting poses several challenges. A naive use of an approximation scheme could produce further issues as the noise could lead different nodes to take different escaping directions, potentially canceling each other out. Further, in order to control the consensus error and the gradient tracking disagreement we adopt a significantly smaller step size than the one used in the centralized case. Finally, using a common potential function both for Phase I and Phase II derives an interesting tradeoff between the corresponding stepsizes. Taking into account all these challenges we design PDGT to guarantee escaping from strict saddle points. In particular, we show that at the end of the second phase, either a carefully chosen potential function decreases - PDGT escapes from a saddle point - and we go back to Phase I, or an approximate

**Algorithm 2:** PDGT algorithm: Phase I

---

1: **Input:** $\underline{\mathbf{x}}, \underline{\mathbf{y}}, \eta_1, T_1, \delta_1$
2: **Initialization:** $\underline{\mathbf{x}}^0 = \underline{\mathbf{x}}, \quad \underline{\mathbf{y}}^0 = \underline{\mathbf{y}};$
3: **for** $r = 1, \ldots, T_1$ **do**
4:      Compute $\mathbf{x}_i^r = \sum_{j \in \mathcal{N}_i} w_{ij} \mathbf{x}_j^{r-1} - \eta_1 \mathbf{y}_i^{r-1};$                $\forall i = 1, \ldots, m$
5:      Compute $\mathbf{y}_i^r = \sum_{j \in \mathcal{N}_i} w_{ij} \mathbf{y}_j^{r-1} + \nabla f_i(\mathbf{x}_i^r) - \nabla f_i(\mathbf{x}_i^{r-1});$      $\forall i = 1, \ldots, m$
6:      Exchange $\mathbf{x}_i^r$ and $\mathbf{y}_i^r$ with neighboring nodes;             $\forall i = 1, \ldots, m$
7: **end for**
8: **for** $j = 1 : \log(\frac{1}{\delta_1})$ **do**
9:      Choose index $\tilde{t}_j \sim [0, T_1]$ uniformly at random and run Consensus Protocol on $\tilde{t}_j$ to find first order stationary point $\underline{\tilde{\mathbf{x}}}$ with small gradient tracking disagreement;
10: **end for**
     **Result:** Returns first order stationary point $\underline{\tilde{\mathbf{x}}}$ with probability at least $1 - \delta_1$

---

second-order stationary point has been reached and the exact iterate is reported. Next, we present the details of both phases of PDGT.

**Phase I.** Consider $\nabla f_i(\mathbf{x}_i)$, the local gradient of node $i$, and define $\mathbf{y}_i \in \mathbb{R}^d$ as the variable of node $i$ which is designed to track the global average gradient $\frac{1}{m} \sum_{i=1}^m \nabla f_i(\mathbf{x}_i)$. The algorithm proceeds to update the iterates $\mathbf{x}_i$ based on the directions of $\mathbf{y}_i$. More specifically, at each iteration $r$, each agent $i$ first updates its local decision variable by averaging its local iterate with the iterates of its neighbors and descending along the negative direction of its gradient estimate $\mathbf{y}_i^{r-1}$, i.e.,

$$\mathbf{x}_i^r = \sum_{j \in \mathcal{N}_i} w_{ij} \mathbf{x}_j^{r-1} - \eta_1 \mathbf{y}_i^{r-1}, \tag{5}$$

where $\eta_1$ is the stepsize and $w_{ij}$ is the weight that node $i$ assigns to the information that it receives from node $j$. We assume that $w_{ij} > 0$ only for the nodes $j$ that are in the neighborhood of node $i$, which also includes node $i$ itself. Further, the sum of these weights is 1, i.e., $\sum_{j \in \mathcal{N}_i} w_{ij} = 1$.

Once the local $\mathbf{x}_i$'s are updated, each agent $i$ computes its local gradient $\nabla f_i(\mathbf{x}_i^r)$ evaluated at its current iterate $\mathbf{x}_i^r$. Then, the nodes use the gradient tracking variable $\mathbf{y}_i^{r-1}$ received from their neighbors in the previous round to update their gradient tracking vector according to the update

$$\mathbf{y}_i^r = \sum_{j \in \mathcal{N}_i} w_{ij} \mathbf{y}_j^{r-1} + \nabla f_i(\mathbf{x}_i^r) - \nabla f_i(\mathbf{x}_i^{r-1}), \tag{6}$$

Note that the update in (6) shows that node $i$ computes its new global gradient estimate by combining its previous local estimate with the ones communicated by its neighbors as well as the difference of its two consecutive local gradients. Once the local gradient tracking variables are updated, nodes communicate their local models $\mathbf{x}_i^r$ and local gradient tracking vectors $\mathbf{y}_i^r$ with their neighbors.

After running the updates in (5) and (6) for $T_1$ rounds, we can ensure that we have visited a set of points $[\mathbf{x}_1, \ldots, \mathbf{x}_m]$ that construct a first-order stationary point of Problem (2) (see Theorem 1); however, nodes are oblivious to the time index of those iterates. To resolve this issue all nodes sample a common time index $r \in \{1, \ldots, T_1\}$ and run an average consensus protocol among themselves to compute the expression $\left\| \frac{1}{m} \sum_{i=1}^m \nabla f_i(\tilde{\mathbf{x}}_i) \right\|^2 + \frac{1}{m} \sum_{i=1}^m \| \tilde{\mathbf{x}}_i - \frac{1}{m} \sum_{j=1}^m \tilde{\mathbf{x}}_j \|^2$ for that time index. By repeating this process at most $\log(\frac{1}{\delta_1})$ times, the output of the process leads to a set of points satisfying first-order optimality with probability at least $1 - \delta_1$. The details of this procedure are provided in the appendix. Note that the consensus procedure is standard and known to be linearly convergent. Hence, the additional cost of running the consensus protocol $\log(\frac{1}{\delta_1})$ times is negligible compared to $T_1$; see Theorem 1 for more details.

**Phase II.** In the second phase of PDGT we are given a set of variables denoted by $\underline{\tilde{\mathbf{x}}} = [\tilde{\mathbf{x}}_1, \ldots, \tilde{\mathbf{x}}_m]$ which is a first-order stationary point. The goal is to escape from it, if it is a strict saddle, i.e., the smallest eigenvalue of the Hessian at this point is sufficiently negative. Initialized with a first-order stationary point $\underline{\tilde{\mathbf{x}}}$ the algorithm injects the same noise $\xi$ picked uniformly from a ball of radius $\mathcal{R} = \tilde{O}(\gamma^{\frac{3}{2}})$, to all the local iterates $\tilde{\mathbf{x}}_i$. Thus for all $i$ we have $\mathbf{x}_i^0 = \tilde{\mathbf{x}}_i + \xi$. After initialization

---

**Algorithm 3:** PDGT algorithm: Phase II

---

1: **Input:** $\underline{\tilde{\mathbf{x}}}, \eta_2, T_2, \mathcal{R}, B$
2: All nodes sample a vector $\xi \sim$ uniform ball of radius $\mathcal{R}$ using the same seed;
3: Set $\mathbf{x}_i^0 = \tilde{\mathbf{x}}_i + \xi$ and run Average Consensus on $\nabla f_i(\mathbf{x}_i^0)$ to set $\mathbf{y}_i^0 = \frac{1}{m}\sum_{i=1}^{m} \nabla f_i(\mathbf{x}_i^0)$;
4: **for** $r = 1, \ldots, T_2$ **do**
5:   Compute $\mathbf{x}_i^r = \sum_{j \in \mathcal{N}_i} w_{ij}\mathbf{x}_j^{r-1} - \eta_2 \mathbf{y}_i^{r-1}$;                                     $\forall i = 1, \ldots, m$
6:   Compute $\mathbf{y}_i^r = \sum_{j \in \mathcal{N}_i} w_{ij}\mathbf{y}_j^{r-1} + \nabla f_i(\mathbf{x}_i^r) - \nabla f_i(\mathbf{x}_i^{r-1})$;       $\forall i = 1, \ldots, m$
7:   Exchange $\mathbf{x}_i^r$ and $\mathbf{y}_i^r$ with neighboring nodes;                            $\forall i = 1, \ldots, m$
8: **end for**
9: Run Average Consensus Protocol for iterates $\underline{\mathbf{x}}^{T_2}$ and $\underline{\tilde{\mathbf{x}}}$;
10: **if** $H(\underline{\mathbf{x}}^{T_2}, \underline{\mathbf{y}}^{T_2}) - H(\underline{\tilde{\mathbf{x}}}, \underline{\tilde{\mathbf{y}}}) > -B$ **then**
11:   Return approximate second-order stationary point $\underline{\tilde{\mathbf{x}}} = [\tilde{\mathbf{x}}_1, \ldots, \tilde{\mathbf{x}}_m]$ and set $S = 1$;
12: **else**
13:   Return $\underline{\mathbf{x}}^{T_2} = [\mathbf{x}_1^{T_2}, \ldots, \mathbf{x}_m^{T_2}]$,   $\underline{\mathbf{y}}^{T_2} = [\mathbf{y}_1^{T_2}, \ldots, \mathbf{y}_m^{T_2}]$ and set $S = 0$;
14: **end if**

---

all nodes follow the updates in (5) and (6) with stepsize $\eta_2$, for $T_2$ rounds. If the initial point was a strict saddle then at the end of this process the iterates escape from it; as a result our properly chosen potential function $H$ (formally defined in (9) in Section 4) decreases substantially and then we revisit Phase I. If the potential function $H$ does not decrease sufficiently, then we conclude that $\underline{\tilde{\mathbf{x}}} = [\tilde{\mathbf{x}}_1, \ldots, \tilde{\mathbf{x}}_m]$ is a second-order stationary point of Problem (2). More precisely, choosing a proper stepsize $\eta_2$ and running PDGT for $T_2 = \tilde{O}(d\gamma^{-3})$ iterations decreases the potential function $H$ by at least $B = \tilde{O}(\gamma^3)$, with probability $1 - \delta_2$, where $T_2$ has only a polylogarithmic dependence on $\delta_2$. If the potential function is not substantially decreased then we confidently report $\underline{\tilde{\mathbf{x}}}$ as an approximate second-order stationary point. Note that $S$ is our indicator, tracking whether we have encountered some approximate second-order stationary point or not. Further, the average consensus protocol is utilized in the second phase both to initialize the gradient tracking variables and to evaluate the potential function $H$ at the iterates $\underline{\mathbf{x}}^{T_2}$ and $\underline{\tilde{\mathbf{x}}}$. Since the communication cost of the average consensus protocol is logarithmic in $\gamma^{-1}$, it is negligible compared to $T_2$. Hence, the number of communication rounds for Phase II is $\tilde{O}(d\gamma^{-3})$. Check Theorem 2 for more details.

## 4   Theoretical Results

In this section, we study convergence properties of our proposed PDGT method. First, we characterize the number of rounds $T_1$ required in Phase I of PDGT to find a set of first-order stationary points with high probability. Then, we establish an upper bound for $T_2$, the number of communication rounds required in the second phase. We further show that each time the algorithm finishes Phase II, a potential function decreases at least by $\tilde{\Theta}(\gamma^3)$. Finally, using these results, we characterize the overall communication rounds between nodes to find a second-order stationary point.

Before stating our result, we first discuss some conditions required for the averaging weights used in (5) and (6). Consider the mixing matrix $\mathbf{W} \in \mathbb{R}^{m \times m}$ where the element of its $i$-th row and $j$-th column is $w_{ij}$. We assume $\mathbf{W}$ satisfies the following conditions.

**Assumption 3.** *The mixing matrix $\mathbf{W} \in \mathbb{R}^{m \times m}$ satisfies the following:*

$$\mathbf{W} = \mathbf{W}^\top, \qquad \mathbf{W}\mathbf{1} = \mathbf{1}, \qquad \sigma := \max\{|\lambda_2(\mathbf{W})|, |\lambda_m(\mathbf{W})|\} < 1, \tag{7}$$

*where $\lambda_i(\mathbf{W})$ denotes the $i$-th largest eigenvalue of $\mathbf{W}$.*

The first condition in Assumption 3 implies that the weight node $i$ assigns to node $j$ equals the weight node $j$ assigns to node $i$. The second condition means $\mathbf{W}$ is row stochastic, and by symmetry, column stochastic. This condition ensures that the weights that each node $i$ assigns to its neighbors and itself sum up to 1. Further note that the eigenvalues of $\mathbf{W}$ are real and in the interval $[-1, 1]$; in fact they can be sorted in a non-increasing order as $1 = \lambda_1(\mathbf{W}) \geq \lambda_2(\mathbf{W}) \geq \cdots \geq \lambda_m(\mathbf{W}) \geq -1$. The last condition in Assumption 3 ensures that the maximum absolute value of all eigenvalues of $\mathbf{W}$

excluding $\lambda_1(\mathbf{W})$ is strictly smaller than 1. This is required since $\sigma := \max\{|\lambda_2(\mathbf{W})|, |\lambda_m(\mathbf{W})|\}$ indicates the rate of information propagation. For highly connected graphs $\sigma$ is close to zero, while for less connected graphs it is close to 1. A mixing matrix $W$ satisfying Assumption 3 can be chosen based on local degrees in a variety of ways (e.g., [36]).

**Remark 1.** *In the appendix we report explicit expressions. To simplify the presentation in the main body, we turn to asymptotic notation and consider sufficiently small $\eta$ and $\alpha$, thus hiding constants but preserving the scaling with respect to quantities that capture important elements of our analysis.*

Next, we present our first result, which formally characterizes the choice of parameters for PDGT to find an $(\epsilon, \rho)$-first-order stationary point, as defined in (1), with probability $1 - \delta_1$.

**Theorem 1.** *Consider Phase I of PDGT presented in Algorithm 2. If Assumptions 1 and 3 hold, and we set $\eta_1 = \Theta((1 - \sigma)\sqrt{\alpha})$ where $\alpha = \Theta((1 - \sigma)^2)$, and the number of iterations satisfies $T_1 \geq T = \Theta\left(\frac{f(\mathbf{x}^0) - f^*}{\eta_1 \epsilon^2}\right) = \Theta\left(\frac{f(\mathbf{x}^0) - f^*}{\sqrt{\alpha}(1 - \sigma)\epsilon^2}\right)$, then w.p. at least $1 - \delta_1$, the iterates $\tilde{\mathbf{x}}_1, \ldots, \tilde{\mathbf{x}}_m$ corresponding to one of the randomly selected time indices $\tilde{t}_1, .., \tilde{t}_{\log(\frac{1}{\delta_1})}$ from $[0 : T_1]$, satisfy*

$$\left\|\frac{1}{m}\sum_{i=1}^{m}\nabla f_i(\tilde{\mathbf{x}}_i)\right\|^2 + \frac{1}{m}\sum_{i=1}^{m}\left\|\tilde{\mathbf{x}}_i - \frac{1}{m}\sum_{j=1}^{m}\tilde{\mathbf{x}}_j\right\|^2 \leq \epsilon^2. \tag{8}$$

Theorem 1 shows that after $\Theta\left(\frac{f(\mathbf{x}^0) - f^*}{\sqrt{\alpha}(1 - \sigma)\epsilon^2} + \frac{1}{1 - \sigma}\log(\frac{1}{\delta_1})\log(\frac{1}{\epsilon})\right)$ rounds of exchanging information with neighboring nodes the goal of Phase I is achieved and we obtain a set of first-order stationary points with small gradient tracking disagreement. Note that the second term $\frac{1}{1 - \sigma}\log(\frac{1}{\delta_1})\log(\frac{1}{\epsilon})$ corresponds to the cost of running the average consensus protocol to choose the appropriate iterate among time steps $\tilde{t}_1, \tilde{t}_2, ..., \tilde{t}_{\log(\frac{1}{\delta_1})}$. This term is negligible compared to the first term.

Next we present our result for Phase II of PDGT. In particular, we show that if the input of Phase II, which satisfies (8), is a strict saddle meaning it has sufficient negative curvature, then PDGT will escape from it and as a result the following Lyapunov function decreases:

$$H(\underline{\mathbf{x}}, \underline{\mathbf{y}}) := \frac{1}{m}\sum_{i=1}^{m}f_i(\mathbf{x}_{avg}) + \frac{1}{m}\sum_{i=1}^{m}\|\mathbf{x}_i - \mathbf{x}_{avg}\|^2 + \frac{\alpha}{m}\sum_{i=1}^{m}\|\mathbf{y}_i - \mathbf{y}_{avg}\|^2, \tag{9}$$

where $\underline{\mathbf{x}} := [\mathbf{x}_1; \ldots; \mathbf{x}_m]$, $\underline{\mathbf{y}} := [\mathbf{y}_1; \ldots; \mathbf{y}_m]$, $\mathbf{x}_{avg} = \frac{1}{m}\sum_{j=1}^{m}\mathbf{x}_j$ and $\mathbf{y}_{avg} = \frac{1}{m}\sum_{j=1}^{m}\mathbf{y}_j$.

**Theorem 2.** *Consider Phase II of PDGT presented in Algorithm 3, and suppose Assumptions 1-3 hold. Further, suppose we set $\eta_2 = \tilde{\Theta}\left(\frac{\gamma^2}{d(1 - \sigma)}\right)$ and $\alpha = \tilde{\Theta}\left((1 - \sigma)^2\right)$, and the local perturbed iterates are computed according to $\mathbf{x}_i^0 = \tilde{\mathbf{x}}_i + \xi$, where $\xi$ is drawn from the uniform distribution over the ball of radius $R = \tilde{\Theta}(\gamma^{1.5})$. If the input of the second phase denoted by $\tilde{\mathbf{x}}_1, \ldots, \tilde{\mathbf{x}}_m$ satisfies*

$$\lambda_{\min}(\nabla^2 f(\tilde{\mathbf{x}}_{avg})) \leq -\gamma, \quad \left\|\frac{1}{m}\sum_{i=1}^{m}\nabla f_i(\tilde{\mathbf{x}}_i)\right\|^2 \leq \epsilon_1^2, \quad \frac{1}{m}\sum_{i=1}^{m}\left\|\tilde{\mathbf{x}}_i - \frac{1}{m}\sum_{j=1}^{m}\tilde{\mathbf{x}}_j\right\|^2 \leq \epsilon_2^2,$$

*where $\epsilon_1^2 = \tilde{\mathcal{O}}(\gamma^3)$ and $\epsilon_2^2 = \tilde{\mathcal{O}}(\frac{\gamma^5}{d})$, then after $T_2 \geq T = \tilde{\Theta}\left(\frac{d\log(1/\gamma\delta_2)}{\gamma^3}\right)$ iterations with probability at least $1 - \delta_2$ we have $H(\underline{\mathbf{x}}^{T_2}, \underline{\mathbf{y}}^{T_2}) - H(\underline{\tilde{\mathbf{x}}}, \underline{\tilde{\mathbf{y}}}) = -\tilde{\Omega}(\gamma^3)$.*

The result in Theorem 2 shows that if the input of Phase II of PDGT is a first-order stationary point with sufficient negative curvature, then by following the update of PDGT for $\tilde{\Theta}(\frac{d\log(1/\gamma\delta_2)}{\gamma^3})$ iterations with probability at least $1 - \delta_2$ the Lyapunov function $H$ decreases by $\tilde{\Omega}(\gamma^3)$. Further in order for the nodes to verify whether enough progress has been made we include two calls on the average consensus protocol on iterates $\underline{\tilde{\mathbf{x}}}$ and $\underline{\mathbf{x}}^{T_2}$ with overall communication complexity $\mathcal{O}(\frac{2}{1 - \sigma}\log(\frac{1}{\min\{\epsilon_1, \epsilon_2\}}))$, which is negligible compared to $\tilde{\Theta}(\frac{d\log(1/\gamma\delta_2)}{\gamma^3})$ iterations.

Combining the results of Theorems 1 and 2, and using the fact that the Lyapunov function $H$ is non-increasing in the first phase (proof is available in section 9) we obtain that if the outcome of the first phase has sufficient negative curvature (i.e, is a strict saddle), then the Lyapunov function $H$ after Phase I and Phase II decreases at least by $\tilde{\Theta}(\gamma^3)$. Hence, after at most $\tilde{\Theta}(\gamma^{-3})$ calls to the first and second phase of PDGT, we will find a second-order stationary point of Problem (2).

**Theorem 3.** *Consider the PDGT method in Algorithm 1, and suppose Assumptions 1-3 hold. If we set the stepsizes as $\eta_1 = \tilde{\Theta}\left((1-\sigma)^2\right), \eta_2 = \tilde{\Theta}\left(\frac{\gamma^2}{d(1-\sigma)}\right)$ and the number of iterations as $T_1 = \tilde{\Theta}\left(\frac{f(\mathbf{x}^0)-f^*}{(1-\sigma)^2 \min\{\epsilon^2,\rho^2\}}\right)$ and $T_2 = \tilde{\Theta}\left(\frac{d}{\gamma^3}\right)$, respectively, and we have $\epsilon^2 = \tilde{\mathcal{O}}\left(\gamma^3\right)$ and $\rho^2 = \tilde{\mathcal{O}}\left(\gamma^5/d\right)$, then after at most $\tilde{\Theta}\left(\max\left\{\frac{f(\mathbf{x}^0)-f^*}{(1-\sigma)^2 \min\{\epsilon^2,\rho^2\}\gamma^3}, \frac{d}{\gamma^6}\right\}\right)$ communication rounds PDGT finds an $(\epsilon, \gamma, \rho)$-second-order stationary point of Problem* (2)*, with high probability.*

A major difference between the analysis of PDGT and its centralized counterpart in [15] is that as the iterates move away from a first-order stationary point, the consensus error and the gradient tracking disagreement potentially increase exponentially fast blurring the escaping direction. Addressing this issue requires careful selection of the algorithm's parameters and setting appropriate stepsizes finetuning the tradeoff on the number of iterations between the first and the second phase. The aforementioned hurdles and the lack of knowledge regarding when the algorithm iterates lie close to a stationary point lead to an overall slower convergence rate than the one shown in the centralized case.

Recall that if the local solutions $[\hat{\mathbf{x}}_1, \ldots, \hat{\mathbf{x}}_m]$ form an $(\epsilon, \gamma, \rho)$-second-order stationary point of Problem (2), then their average $\hat{\mathbf{x}}_{avg} := \frac{1}{m}\sum_{i=1}^{m} \hat{\mathbf{x}}_i$ is an $(\epsilon + L_1\rho, \gamma + L_2\rho)$-second-order stationary point of Problem (1). Moreover, as discussed earlier, second order stationary points are of paramount importance because when all saddle points are strict, any second-order stationary point is a local minima. We formally state this condition in the following assumption and later show that under this assumption PDGT finds a local minima of Problem (1).

**Assumption 4.** *Function $f(\cdot)$ is $(\theta, \zeta, \nu)$- strict saddle, when for any point $\mathbf{x}$, if its gradient norm is smaller than $\theta$, then its Hessian satisfies the condition $\lambda_{\min}(\nabla^2 f(\mathbf{x})) \leq -\zeta$, unless $\mathbf{x}$ is $\nu-$close to the set of local minima.*

The strict saddle condition defined in Assumption 4 states that if a function is $(\theta, \zeta, \nu)$- strict saddle then each point in $\mathbb{R}^d$ belongs to one of these regions: 1) a region where the gradient is large and it is not close to any stationary point; 2) a region where the gradient is small but the Hessian has a significant negative eigenvalue; and 3) the region close to some local minimum. Indeed, under the extra assumption of strict saddle property on function $f$, PDGT is able to find a local minima in a finite number of iterations as we state in the following corollary.

**Corollary 1.** *Consider the PDGT method presented in Algorithm 3 and suppose the conditions in Theorem 3 are satisfied. If in addition Assumption 4 holds and the objective function $f$ is $(\theta, \zeta, \nu)$-strict saddle point, by setting $\epsilon + L_1\rho \leq \theta$ and $\gamma + L_2\rho \leq \zeta$, the PDGT will output a point $\nu-$close to the set of local minima after $\tilde{\Theta}\left(\max\left\{\frac{f(\mathbf{x}^0)-f^*}{(1-\sigma)^2 \min\{\epsilon^2,\rho^2\}\gamma^3}, \frac{d}{\gamma^6}\right\}\right)$ communication rounds.*

## 5 Numerical Experiments

In this section, we compare PDGT with a simple version of D-GET where each node has full knowledge of its local gradient. D-GET is a decentralized gradient tracking method that "does not use the perturbation idea" [36]. Our goal is to show that PDGT escapes quickly from saddle points. We focus on a matrix factorization problem for the MovieLens dataset, where the goal is to find a rank $r$ approximation of a matrix $\mathbf{M} \in \mathcal{M}^{l \times n}$, representing the ratings from 943 users to 1682 movies. Each user has rated at least 20 movies for a total of 9990 known ratings. This problem is given by:

$$(\mathbf{U}^*, \mathbf{V}^*) := \operatorname*{argmin}_{\mathbf{U} \in \mathcal{M}^{l \times r}, \mathbf{V} \in \mathcal{M}^{n \times r}} f(\mathbf{U}, \mathbf{V}) = \operatorname*{argmin}_{\mathbf{U} \in \mathcal{M}^{l \times r}, \mathbf{V} \in \mathcal{M}^{n \times r}} \|\mathbf{M} - \mathbf{U}\mathbf{V}^\top\|_F^2. \tag{10}$$

We consider different values of target rank and number of nodes. Both methods are given the same randomly generated connected graph, mixing matrix, and step size. The graph is created using the $G(n, p)$ model with $p = \frac{\log_2(n)}{n-1}$ enforcing the path $1 - 2 - \ldots - (n-1) - n$ to ensure the connectivity of the graph. Further we utilize the Maximum Degree Weight mixing matrix as is presented in (10) of [36]. The stepsize for D-GET and both phases of PDGT is 3. Finally both methods are initialized at the same point which lies in a carefully chosen neighborhood of a saddle point. Note that in this problem all saddles are escapable and each local min is a global min. Regarding the parameters of PDGT we set the number of rounds during phase I and II to be 1500 and 100, respectively. Further, we set the threshold before we add noise during phase I as presented in (8) to be $10^{-6}$ and the radius of the noise injected to be 4.

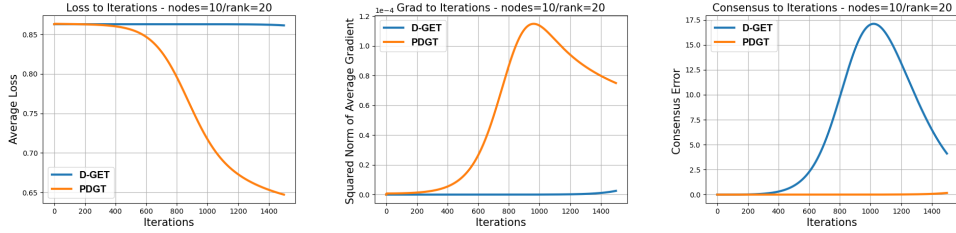

Figure 1: Average loss (left), squared norm of the average gradient (middle), consensus error (right) vs. iteration (10 nodes and target rank 20).

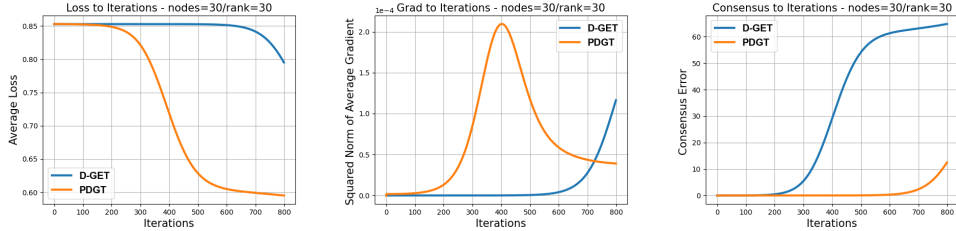

Figure 2: Average loss (left), squared norm of the average gradient (middle), consensus error (right) vs. iteration (30 nodes and target rank 30).

In Fig. 1 the experiment is run for 10 nodes, and the target rank is 20. Initially both algorithms are stuck close to a saddle point and make very little progress. However, since the theoretical criterion for PDGT is satisfied in the very first rounds (small average gradient and consensus error) we have injection of noise. This nudge is sufficient to accelerate substantially the escape of PDGT. As we see in the plot, D-GET remains close to the saddle point at least until iteration 1400 where we can see the gradient increasing somewhat faster. At the same time PDGT escapes the saddle point, decreases the loss and approaches a local minimum. In Fig. 2, the experiment is run for 30 nodes and the target rank is 30. Similarly, PDGT escapes from the saddle point much faster and decreases the loss substantially before it reaches the local minimum. We observe that D-GET also escapes the saddle point eventually following a similar trace to PDGT after spending a lot longer at the saddle. Interestingly, for this experiment, we observed that some parameters such as the stepsize of the first and the second phase, the injected noise and the threshold before we inject noise can afford to be substantially greater than the theoretical propositions casting PDGT useful for a series of practical applications.

## 6   Conclusion and Future Work

We proposed the Perturbed Decentralized Gradient Tracking (PDGT) algorithm that achieves second-order stationarity in a finite number of iterations, under the assumptions that the objective function gradient and Hessian are Lipschitz. We showed that PDGT finds an $(\epsilon, \gamma, \rho)$-second-order stationary point, where $\epsilon$ and $\gamma$ indicate the accuracy for first- and second-order optimality, respectively, and $\rho$ shows the consensus error, after $\tilde{\Theta}\left(\max\left\{\frac{f(\mathbf{x}^0)-f^*}{(1-\sigma)^2 \min\{\epsilon^2,\rho^2\}\gamma^3}, \frac{d}{\gamma^6}\right\}\right)$ communication rounds, where $d$ is dimension, $f(\mathbf{x}^0) - f^*$ is the initial error, and $1 - \sigma$ is related to graph connectivity.

This paper is the first step towards achieving second-order optimality in decentralized settings under standard smoothness assumptions, and several research problems are still unanswered in this area. First, our complexity scales linearly with dimension $d$, deviating from the poly-logarithmic dependence achieved for centralized perturbed gradient descent [15]. Closing this gap and developing an algorithm that obtains second-order optimality with communication rounds that scale sublinearly or even poly-logarithmically on the dimension is a promising research direction that requires further investigation. Second, in the centralized setting, it has been shown that by using gradient acceleration [16] it is possible to find a second-order stationary point faster than perturbed gradient descent. It would be interesting to see if the same conclusion also holds for decentralized settings. Last, extending the theory developed in this paper to the case that nodes only have access to a noisy estimate of their local gradients is another avenue of research that requires further study.

## 7 Broader Impact

Over the last couple of years we have witnessed an unprecedented increase in the amount of data collected and processed in order to tackle real life problems. Advances in numerous data-driven system such as the Internet of Things, health-care, multi-agent robotics wherein data are scattered across the agents (e.g., sensors, clouds, robots), and the sheer volume and spatial/temporal disparity of data render centralized processing and storage infeasible or inefficient. Compared to the typical parameter-server type distributed system with a fusion center, decentralized optimization has its unique advantages in preserving data privacy, enhancing network robustness, and improving the computation efficiency. Furthermore, in many emerging applications such as collaborative filtering, federated learning, distributed beamforming and dictionary learning, the data is naturally collected in a decentralized setting, and it is not possible to transfer the distributed data to a central location. Therefore, decentralized computation has sparked considerable interest in both academia and industry. At the same time convex formulations for training machine learning tasks have been replaced by nonconvex representations such as neural networks and a line of significant non convex problems are on the spotlight. Our paper contributes to this line of work and broadens the set of problems that can be successfully solved without the presence of a central coordinating authority in the aforementioned framework. The implications on the privacy of the agents are apparent while rendering the presence of an authority unnecessary has political and economical extensions. Furthermore, numerous applications are going to benefit from our result impacting society in many different ways.

## 8 Acknowledgments and Disclosure of Funding

The research of I. Tziotis and A. Mokhtari is supported by NSF Award CCF-2007668. C. Caramanis is supported by NSF Awards 1704778, 1646522, and 1609279.

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
