[Supplementary Material]

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

# Supplementary Material

The first two lemmas introduce a connection between first order and second order stationary points in the centralized and the decentralized regime.

**Lemma 1.** *Assume $[\hat{\mathbf{x}}_1, \ldots, \hat{\mathbf{x}}_m]$ is an $(\epsilon, \rho)$-first-order stationary point in the decentralized regime, that is*

$$\left\| \frac{1}{m} \sum_{i=1}^{m} \nabla f_i(\hat{\mathbf{x}}_i) \right\| \leq \epsilon, \qquad \frac{1}{m} \sum_{i=1}^{m} \left\| \hat{\mathbf{x}}_i - \frac{1}{m} \sum_{j=1}^{m} \hat{\mathbf{x}}_j \right\| \leq \rho. \tag{11}$$

*Then their average $\hat{\mathbf{x}}_{avg} := \frac{1}{m} \sum_{i=1}^{m} \hat{\mathbf{x}}_i$ is an $(\epsilon + L_1 \rho)$-first-order stationary point in the centralized regime i.e., $\|\frac{1}{m} \sum_{i=1}^{m} \nabla f_i(\hat{\mathbf{x}}_{avg})\| \leq \epsilon + L_1 \rho$.*

*Proof.*

$$\left\| \frac{1}{m} \sum_{i=1}^{m} \nabla f_i(\hat{\mathbf{x}}_{avg}) \right\| \leq \left\| \frac{1}{m} \sum_{i=1}^{m} \nabla f_i(\hat{\mathbf{x}}_{avg}) - \frac{1}{m} \sum_{i=1}^{m} \nabla f_i(\hat{\mathbf{x}}_i) \right\| + \left\| \frac{1}{m} \sum_{i=1}^{m} \nabla f_i(\hat{\mathbf{x}}_i) \right\| \tag{12}$$

$$\leq \frac{L_1}{m} \sum_{i=1}^{m} \|\hat{\mathbf{x}}_{avg} - \hat{\mathbf{x}}_i\| + \epsilon \tag{13}$$

$$\leq L_1 \rho + \epsilon \tag{14}$$

where in the first inequality we add and subtract the same term and in the second one we use smoothness of $f$. $\square$

**Lemma 2.** *Assume $[\hat{\mathbf{x}}_1, \ldots, \hat{\mathbf{x}}_m]$ is an $(\epsilon, \gamma, \rho)$-second-order stationary point in the decentralized regime, that is*

$$\left\| \frac{1}{m} \sum_{i=1}^{m} \nabla f_i(\hat{\mathbf{x}}_i) \right\| \leq \epsilon, \qquad \frac{1}{m} \sum_{i=1}^{m} \nabla^2 f_i(\hat{\mathbf{x}}_i) \succeq -\gamma \, \mathbf{I}, \qquad \frac{1}{m} \sum_{i=1}^{m} \left\| \hat{\mathbf{x}}_i - \frac{1}{m} \sum_{j=1}^{m} \hat{\mathbf{x}}_j \right\| \leq \rho. \tag{15}$$

*Then their average $\hat{\mathbf{x}}_{avg} := \frac{1}{m} \sum_{i=1}^{m} \hat{\mathbf{x}}_i$ is an $(\epsilon + L_1 \rho, \gamma + L_2 \rho)$-second-order stationary point in the centralized regime i.e., i.e., $\|\frac{1}{m} \sum_{i=1}^{m} \nabla f_i(\hat{\mathbf{x}}_{avg})\| \leq \epsilon + L_1 \rho$ and $\frac{1}{m} \sum_{i=1}^{m} \nabla^2 f_i(\hat{\mathbf{x}}_{avg}) \succeq -(\gamma + L_2 \rho) \, \mathbf{I}$.*

*Proof.* The first part is identical to Lemma 1. for the second part we work in a similar fashion.

$$\left\| \frac{1}{m} \sum_{i=1}^{m} \nabla^2 f_i(\hat{\mathbf{x}}_{avg}) \right\| \leq \left\| \frac{1}{m} \sum_{i=1}^{m} \nabla^2 f_i(\hat{\mathbf{x}}_{avg}) - \frac{1}{m} \sum_{i=1}^{m} \nabla^2 f_i(\hat{\mathbf{x}}_i) \right\| + \left\| \frac{1}{m} \sum_{i=1}^{m} \nabla^2 f_i(\hat{\mathbf{x}}_i) \right\| \tag{16}$$

$$\leq \frac{L_2}{m} \sum_{i=1}^{m} \|\hat{\mathbf{x}}_{avg} - \hat{\mathbf{x}}_i\| + \gamma \tag{17}$$

$$\leq L_2 \rho + \gamma \tag{18}$$

$\square$

where in the first inequality we add and subtract the same term and in the second one we use the Lipschitz continuous Hessian of $f$. The result follows.

## 9 Convergence to First Order Stationary Point with Consensus

**Initialization of Phase I**

$\underline{\mathbf{x}}^0 = \underline{\mathbf{x}}_{input}$

$\underline{\mathbf{y}}^0 = \underline{\mathbf{y}}_{input}$, with $\underline{\mathbf{y}}_{input}$ such that

$$\frac{1}{m} \sum_{i=1}^{m} \mathbf{y}_i^0 = \frac{1}{m} \sum_{i=1}^{m} f_i(\mathbf{x}_i^0)$$

Recall that the first time we initialize the algorithm the following also hold

$$\mathbf{x}_i^0 = \mathbf{x}_j^0, \quad \forall i, j$$

$$\mathbf{y}_i^0 = \frac{1}{m} \sum_{i=1}^{m} \nabla f_i(\mathbf{x}_i^0)$$

**Update rule of Gradient Tracking**

$$\mathbf{x}_i^r = \sum_{k \in N_i} \mathbf{W}_{ik} \mathbf{x}_k^{r-1} - \eta \mathbf{y}_i^{r-1}$$
$$\mathbf{y}_i^r = \sum_{k \in N_i} \mathbf{W}_{ik} \mathbf{y}_k^{r-1} + \nabla f_i(\mathbf{x}_i^r) - \nabla f_i(\mathbf{x}_i^{r-1})$$

**The Update rule of the average iterate**

$$\hat{\mathbf{x}}^r = \frac{1}{m} \sum_i \mathbf{x}_i^r$$
$$\hat{\mathbf{y}}^r = \frac{1}{m} \sum_i \mathbf{y}_i^r$$

$$\hat{\mathbf{x}}^r = \hat{\mathbf{x}}^{r-1} - \eta \hat{\mathbf{y}}^{r-1}$$

$$\hat{\mathbf{y}}^r = \hat{\mathbf{y}}^{r-1} + \frac{1}{m} \sum_i \nabla f_i(\mathbf{x}_i^r) - \frac{1}{m} \sum_i \nabla f_i(\mathbf{x}_i^{r-1})$$

$$\hat{\mathbf{y}}^r = \frac{1}{m} \sum_i \nabla f_i(\mathbf{x}_i^r)$$

In order to see why the last equality holds notice that $\hat{\mathbf{y}}^0 = \frac{1}{m} \sum_i \nabla f_i(\mathbf{x}_i^0)$ and an induction derives the result.

Also recall that $\sigma := \max\{|\lambda_2(\mathbf{W})|, |\lambda_m(\mathbf{W})|\} < 1$.

First we are going to provide bounds on the iterates proving contraction between consecutive rounds. Consequently, we are going to derive a similar bound on function $P_\alpha(\mathbf{x}^r)$ and combining the above we will show that the potential function $H(\underline{\mathbf{x}}^r, \underline{\mathbf{y}}^r)$ is decreasing between consecutive rounds.

**Lemma 3** (Bound on consecutive iterates)**.** *Assume the iterates $\mathbf{x}_i$ follow the Gradient Tracking Update with stepsize $\eta$ then we have*

$$\|\underline{\mathbf{x}}^r - \underline{\mathbf{x}}^{r-1}\|^2 \leq 8\|\underline{\mathbf{x}}^{r-1} - \underline{\hat{\mathbf{x}}}^{r-1}\|^2 + 4\eta^2 \|\underline{\mathbf{y}}^{r-1} - \underline{\hat{\mathbf{y}}}^{r-1}\|^2 + 4\eta^2 \|\underline{\hat{\mathbf{y}}}^{r-1}\|^2 \qquad (19)$$

*Proof.*

$$\begin{aligned}
\|\underline{\mathbf{x}}^r - \underline{\mathbf{x}}^{r-1}\|^2 &= \|\underline{\mathbf{W}}\mathbf{x}^{r-1} - \eta \underline{\mathbf{y}}^{r-1} - \underline{\mathbf{x}}^{r-1}\|^2 \\
&\leq 2\|\underline{\mathbf{W}}\mathbf{x}^{r-1} - \underline{\mathbf{x}}^{r-1}\|^2 + 2\eta^2 \|\underline{\mathbf{y}}^{r-1}\|^2 \\
&\leq 2\|\underline{\mathbf{W}}\mathbf{x}^{r-1} - \underline{\mathbf{x}}^{r-1} + \underline{\mathbf{W}}\hat{\mathbf{x}}^{r-1} - \underline{\hat{\mathbf{x}}}^{r-1}\|^2 + 2\eta^2 \|\underline{\mathbf{y}}^{r-1}\|^2 \\
&\leq 2\|(\underline{\mathbf{W}} - \mathbf{I})(\underline{\mathbf{x}}^{r-1} - \underline{\hat{\mathbf{x}}}^{r-1})\|^2 + 2\eta^2 \|\underline{\mathbf{y}}^{r-1} - \underline{\hat{\mathbf{y}}}^{r-1} + \underline{\hat{\mathbf{y}}}^{r-1}\|^2 \\
&\leq 2(\|\underline{\mathbf{W}}\| + \|\mathbf{I}\|)^2 \|\underline{\mathbf{x}}^{r-1} - \underline{\hat{\mathbf{x}}}^{r-1}\|^2 + 4\eta^2 \|\underline{\mathbf{y}}^{r-1} - \underline{\hat{\mathbf{y}}}^{r-1}\|^2 + 4\eta^2 \|\underline{\hat{\mathbf{y}}}^{r-1}\|^2 \\
&\leq 8\|\underline{\mathbf{x}}^{r-1} - \underline{\hat{\mathbf{x}}}^{r-1}\|^2 + 4\eta^2 \|\underline{\mathbf{y}}^{r-1} - \underline{\hat{\mathbf{y}}}^{r-1}\|^2 + 4\eta^2 \|\underline{\hat{\mathbf{y}}}^{r-1}\|^2
\end{aligned}$$

$\square$

**Lemma 4** (Iterate Contraction). *Assume the iterates $\mathbf{x}_i$ follow the Gradient Tracking Update with stepsize $\eta$ and let $\mathbf{v}_i^r = \nabla f_i(\mathbf{x}_i^r)$; then we have*

$$\|\underline{\mathbf{x}}^{r+1} - \hat{\underline{\mathbf{x}}}^{r+1}\|^2 \leq (1+\beta_1)\sigma^2\|\underline{\mathbf{x}}^r - \hat{\underline{\mathbf{x}}}^r\|^2 + (1+\frac{1}{\beta_1})\eta^2\|\underline{\mathbf{y}}^r - \hat{\underline{\mathbf{y}}}^r\|^2 \tag{20}$$

$$\|\underline{\mathbf{y}}^{r+1} - \hat{\underline{\mathbf{y}}}^{r+1}\|^2 \leq 8L_1^2(1+\frac{1}{\beta_2})\|\underline{\mathbf{x}}^r - \hat{\underline{\mathbf{x}}}^r\|^2 + ((1+\beta_2)\sigma^2 + \eta^2 4L_1^2(1+\frac{1}{\beta_2}))\|\underline{\mathbf{y}}^r - \hat{\underline{\mathbf{y}}}^r\|^2$$
$$+ \eta^2 4L_1^2(1+\frac{1}{\beta_2})\|\hat{\underline{\mathbf{y}}}^r\|^2 \tag{21}$$

*Proof.*

$$\|\underline{\mathbf{W}}\mathbf{x}^r - \hat{\mathbf{x}}^r\| = \|\underline{\mathbf{W}}(\mathbf{x}^r - \hat{\mathbf{x}}^r)\| \leq \sigma\|\mathbf{x}^r - \hat{\mathbf{x}}^r\| \tag{22}$$

To see why the inequality is true notice that $\mathbf{1}^T(\underline{\mathbf{x}}^r - \hat{\underline{\mathbf{x}}}^r) = 0$, i.e. $\underline{\mathbf{x}}^r - \hat{\underline{\mathbf{x}}}^r$ is orthogonal to $\mathbf{1}^T$, which is the eigenvector corresponding to $\lambda_{\max}(\mathbf{W})$. Similarly,

$$\|\underline{\mathbf{W}}\mathbf{y}^r - \hat{\mathbf{y}}^r\| \leq \sigma\|\mathbf{y}^r - \hat{\mathbf{y}}^r\| \tag{23}$$

$$\begin{aligned}
\|\underline{\mathbf{x}}^{r+1} - \hat{\underline{\mathbf{x}}}^{r+1}\|^2 &= \|\underline{\mathbf{W}}\mathbf{x}^r - \eta\underline{\mathbf{y}}^r - (\hat{\underline{\mathbf{x}}}^r - \eta\hat{\underline{\mathbf{y}}}^r)\|^2 \\
&\leq (1+\beta_1)\|\underline{\mathbf{W}}\mathbf{x}^r - \hat{\mathbf{x}}^r\|^2 + (1+\frac{1}{\beta_1})\eta^2\|\underline{\mathbf{y}}^r - \hat{\underline{\mathbf{y}}}^r\|^2 \\
&\leq (1+\beta_1)\sigma^2\|\underline{\mathbf{x}}^r - \hat{\underline{\mathbf{x}}}^r\|^2 + (1+\frac{1}{\beta_1})\eta^2\|\underline{\mathbf{y}}^r - \hat{\underline{\mathbf{y}}}^r\|^2
\end{aligned}$$

the last inequality comes from (22). Also

$$\begin{aligned}
&\|\underline{\mathbf{y}}^{r+1} - \hat{\underline{\mathbf{y}}}^{r+1}\|^2 \\
&= \|\underline{\mathbf{W}}\mathbf{y}^r + \underline{\mathbf{v}}^{r+1} - \underline{\mathbf{v}}^r - (\hat{\underline{\mathbf{y}}}^r + \hat{\underline{\mathbf{v}}}^{r+1} - \hat{\underline{\mathbf{v}}}^r)\|^2 \\
&\leq (1+\beta_2)\|\underline{\mathbf{W}}\mathbf{y}^r - \hat{\mathbf{y}}^r\|^2 + (1+\frac{1}{\beta_2})\|\underline{\mathbf{v}}^{r+1} - \underline{\mathbf{v}}^r - \hat{\underline{\mathbf{v}}}^{r+1} + \hat{\underline{\mathbf{v}}}^r\|^2 \\
&\leq (1+\beta_2)\sigma^2\|\underline{\mathbf{y}}^r - \hat{\underline{\mathbf{y}}}^r\|^2 + (1+\frac{1}{\beta_2})\|(\mathbf{I} - \frac{\mathbf{1}\mathbf{1}^T}{m})(\underline{\mathbf{v}}^{r+1} - \underline{\mathbf{v}}^r)\|^2 \\
&\leq (1+\beta_2)\sigma^2\|\underline{\mathbf{y}}^r - \hat{\underline{\mathbf{y}}}^r\|^2 + (1+\frac{1}{\beta_2})\sum_{i=1}^m\|\mathbf{v}_i^{r+1} - \mathbf{v}_i^r\|^2 \\
&= (1+\beta_2)\sigma^2\|\underline{\mathbf{y}}^r - \hat{\underline{\mathbf{y}}}^r\|^2 + (1+\frac{1}{\beta_2})\sum_{i=1}^m\|\nabla f_i(\mathbf{x}_i^{r+1}) - \nabla f_i(\mathbf{x}_i^r)\|^2 \\
&\leq (1+\beta_2)\sigma^2\|\underline{\mathbf{y}}^r - \hat{\underline{\mathbf{y}}}^r\|^2 + L_1^2(1+\frac{1}{\beta_2})\sum_{i=1}^m\|\mathbf{x}_i^{r+1} - \mathbf{x}_i^r\|^2 \\
&= (1+\beta_2)\sigma^2\|\underline{\mathbf{y}}^r - \hat{\underline{\mathbf{y}}}^r\|^2 + L_1^2(1+\frac{1}{\beta_2})\|\underline{\mathbf{x}}^{r+1} - \underline{\mathbf{x}}^r\|^2 \\
&\leq (1+\beta_2)\sigma^2\|\underline{\mathbf{y}}^r - \hat{\underline{\mathbf{y}}}^r\|^2 + L_1^2(1+\frac{1}{\beta_2})(8\|\underline{\mathbf{x}}^r - \hat{\underline{\mathbf{x}}}^r\|^2 + 4\eta^2\|\underline{\mathbf{y}}^r - \hat{\underline{\mathbf{y}}}^r\|^2 + 4\eta^2\|\hat{\underline{\mathbf{y}}}^r\|^2) \\
&= 8L_1^2(1+\frac{1}{\beta_2})\|\underline{\mathbf{x}}^r - \hat{\underline{\mathbf{x}}}^r\|^2 + ((1+\beta_2)\sigma^2 + \eta^2 4L_1^2(1+\frac{1}{\beta_2}))\|\underline{\mathbf{y}}^r - \hat{\underline{\mathbf{y}}}^r\|^2 + \eta^2 4L_1^2(1+\frac{1}{\beta_2})\|\hat{\underline{\mathbf{y}}}^r\|^2
\end{aligned}$$

Where the second inequality is from (23) and for the third we use the fact that $\|\mathbf{I} - \frac{\mathbf{1}\mathbf{1}^T}{m}\| \leq 1$. The last inequality is due to Lemma 3. $\qquad\square$

In the following lemma an upper bound on function $P_\alpha(\underline{\mathbf{x}}^r)$ is derived which we are going to combine with the iterate contraction lemma to show that a propery constructed potential function decreases between consecutive rounds.

**Lemma 5** (Intermediate function). *Assume the iterates $\mathbf{x}_i$ follow the Gradient Tracking Update with stepsize $\eta$ and let $\alpha > 0$. Also let $P_\alpha(\underline{\mathbf{x}}^r) := \frac{1}{m}(\|\underline{\mathbf{x}}^r - \hat{\underline{\mathbf{x}}}^r\|^2 + \alpha\|\underline{\mathbf{y}}^r - \hat{\underline{\mathbf{y}}}^r\|^2)$. It follows that*

$$
\begin{aligned}
P_\alpha(\underline{\mathbf{x}}^{r+1}) - P_\alpha(\underline{\mathbf{x}}^r) &\leq \left( (1+\beta_1)\sigma^2 - 1 + 8\alpha L_1^2(1+\frac{1}{\beta_2}) \right) \frac{1}{m}\|\underline{\mathbf{x}}^r - \hat{\underline{\mathbf{x}}}^r\|^2 \\
&+ \left( \alpha((1+\beta_2)\sigma^2 - 1) + \eta^2(1+\frac{1}{\beta_1}) + 4\alpha\eta^2 L_1^2(1+\frac{1}{\beta_2}) \right) \frac{1}{m}\|\underline{\mathbf{y}}^r - \hat{\underline{\mathbf{y}}}^r\|^2 \\
&+ 4\alpha\eta^2 L_1^2(1+\frac{1}{\beta_2})\|\hat{\mathbf{y}}^r\|^2 \quad (24)
\end{aligned}
$$

*Proof.*

$$
\begin{aligned}
&P_\alpha(\underline{\mathbf{x}}^{r+1}) - P_\alpha(\underline{\mathbf{x}}^r) \\
&\leq \frac{1}{m}\left[ \|\underline{\mathbf{x}}^{r+1} - \hat{\underline{\mathbf{x}}}^{r+1}\|^2 + \alpha\|\underline{\mathbf{y}}^{r+1} - \hat{\underline{\mathbf{y}}}^{r+1}\|^2 - \|\underline{\mathbf{x}}^r - \hat{\underline{\mathbf{x}}}^r\|^2 - \alpha\|\underline{\mathbf{y}}^r - \hat{\underline{\mathbf{y}}}^r\|^2 \right] \\
&\leq ((1+\beta_1)\sigma^2 - 1 + \alpha 8 L_1^2(1+\frac{1}{\beta_2}))\frac{1}{m}\|\underline{\mathbf{x}}^r - \hat{\underline{\mathbf{x}}}^r\|^2 \\
&\quad + (\alpha((1+\beta_2)\sigma^2 - 1) + \eta^2(1+\frac{1}{\beta_1}) + \alpha\eta^2 4 L_1^2(1+\frac{1}{\beta_2}))\frac{1}{m}\|\underline{\mathbf{y}}^r - \hat{\underline{\mathbf{y}}}^r\|^2 \\
&\quad + \alpha\eta^2 4 L_1^2(1+\frac{1}{\beta_2})\frac{1}{m}\|\hat{\underline{\mathbf{y}}}^r\|^2 \\
&= \left( (1+\beta_1)\sigma^2 - 1 + 8\alpha L_1^2(1+\frac{1}{\beta_2}) \right) \frac{1}{m}\|\underline{\mathbf{x}}^r - \hat{\underline{\mathbf{x}}}^r\|^2 \\
&\quad + \left( \alpha\left((1+\beta_2)\sigma^2 - 1\right) + \eta^2(1+\frac{1}{\beta_1}) + 4\alpha\eta^2 L_1^2(1+\frac{1}{\beta_2}) \right) \frac{1}{m}\|\underline{\mathbf{y}}^r - \hat{\underline{\mathbf{y}}}^r\|^2 \\
&\quad + 4\alpha\eta^2 L_1^2(1+\frac{1}{\beta_2})\|\hat{\mathbf{y}}^r\|^2
\end{aligned}
$$

where the second inequality comes from Lemma 4. □

Below we derive a bound on the function value of consecutive iterates. Notice that it is not strictly decreasing on every round and thus later we are going to focus on a suitable potential function.

**Lemma 6** (Function decrease). *Assume the iterates $\mathbf{x}_i$ follow the Gradient Tracking Update with stepsize $\eta$; we can show the following two bounds hold.*

$$
\langle \nabla f(\hat{\mathbf{x}}^r), \hat{\mathbf{x}}^{r+1} - \hat{\mathbf{x}}^r \rangle + \frac{L_1}{2}\|\hat{\mathbf{x}}^{r+1} - \hat{\mathbf{x}}^r\|^2 \leq \eta\frac{L_1^2}{2m}\|\underline{\mathbf{x}}^r - \hat{\underline{\mathbf{x}}}^r\|^2 - (\eta\frac{1}{2} - \eta^2\frac{L_1^2}{2})\|\hat{\mathbf{y}}^r\|^2 \quad (25)
$$

$$
f(\hat{\mathbf{x}}^{r+1}) - f(\hat{\mathbf{x}}^r) \leq \eta\frac{L_1^2}{2m}\|\underline{\mathbf{x}}^r - \hat{\underline{\mathbf{x}}}^r\|^2 - \eta\left(\frac{1}{2} - \eta\frac{L_1^2}{2}\right)\|\hat{\mathbf{y}}^r\|^2 \quad (26)
$$

*Proof.* For the first one we work as follows

$$\langle \nabla f(\hat{\mathbf{x}}^r), \hat{\mathbf{x}}^{r+1} - \hat{\mathbf{x}}^r \rangle + \frac{L_1}{2} \|\hat{\mathbf{x}}^{r+1} - \hat{\mathbf{x}}^r\|^2 \leq -\eta \langle \nabla f(\hat{\mathbf{x}}^r), \hat{\mathbf{y}}^r \rangle + \eta^2 \frac{L_1}{2} \|\hat{\mathbf{y}}^r\|^2$$

$$\leq -\eta \langle \nabla f(\hat{\mathbf{x}}^r) - \hat{\mathbf{y}}^r, \hat{\mathbf{y}}^r \rangle - \eta \|\hat{\mathbf{y}}^r\|^2 + \eta^2 \frac{L_1}{2} \|\hat{\mathbf{y}}^r\|^2$$

$$\leq \frac{\eta}{2} \|\nabla f(\hat{\mathbf{x}}^r) - \hat{\mathbf{y}}^r\|^2 + \frac{\eta}{2} \|\hat{\mathbf{y}}^r\|^2 - \eta \|\hat{\mathbf{y}}^r\|^2 + \eta^2 \frac{L_1}{2} \|\hat{\mathbf{y}}^r\|^2$$

$$= \frac{\eta}{2} \| \frac{1}{m} \sum_{i=1}^{m} \nabla f_i(\hat{\mathbf{x}}^r) - \frac{1}{m} \sum_{i=1}^{m} \nabla f_i(\mathbf{x}_i^r)\|^2 - (\frac{\eta}{2} - \frac{\eta^2 L_1}{2}) \|\hat{\mathbf{y}}^r\|^2$$

$$\leq \frac{\eta}{2} \frac{1}{m} \sum_{i=1}^{m} \|\nabla f_i(\hat{\mathbf{x}}^r) - \nabla f_i(\mathbf{x}_i^r)\|^2 - (\frac{\eta}{2} - \frac{\eta^2 L_1}{2}) \|\hat{\mathbf{y}}^r\|^2$$

$$\leq \eta \frac{L_1^2}{2m} \sum_{i=1}^{m} \|\mathbf{x}_i^r - \hat{\mathbf{x}}^r\|^2 - (\frac{\eta}{2} - \frac{\eta^2 L_1}{2}) \|\hat{\mathbf{y}}^r\|^2$$

$$= \eta \frac{L_1^2}{2m} \|\underline{\mathbf{x}}^r - \hat{\underline{\mathbf{x}}}^r\|^2 - (\frac{\eta}{2} - \frac{\eta^2 L_1}{2}) \|\hat{\mathbf{y}}^r\|^2$$

and thus for the second one we have

$$f(\hat{\mathbf{x}}^{r+1}) \leq f(\hat{\mathbf{x}}^r) + \langle \nabla f(\hat{\mathbf{x}}^r), \hat{\mathbf{x}}^{r+1} - \hat{\mathbf{x}}^r \rangle + \frac{L_1}{2} \|\hat{\mathbf{x}}^{r+1} - \hat{\mathbf{x}}^r\|^2$$

$$f(\hat{\mathbf{x}}^{r+1}) - f(\hat{\mathbf{x}}^r) \leq \langle \nabla f(\hat{\mathbf{x}}^r), \hat{\mathbf{x}}^{r+1} - \hat{\mathbf{x}}^r \rangle + \frac{L_1}{2} \|\hat{\mathbf{x}}^{r+1} - \hat{\mathbf{x}}^r\|^2$$

$$\leq \eta \frac{L_1^2}{2m} \|\underline{\mathbf{x}}^r - \hat{\underline{\mathbf{x}}}^r\|^2 + (\eta \frac{1}{2} - \eta^2 \frac{L_1^2}{2}) \|\hat{\mathbf{y}}^r\|^2$$

$\square$

**Lemma 7.** *Assume the iterates $\mathbf{x}_i$ follow the Gradient Tracking Update with stepsize $\eta$. Let us define the potential function $H(\underline{\mathbf{x}}^r, \underline{\mathbf{y}}^r) := \frac{1}{m} \sum_{i=1}^{m} f_i(\hat{\mathbf{x}}^r) + \frac{1}{m} \|\underline{\mathbf{x}}^r - \hat{\underline{\mathbf{x}}}^r\|^2 + \frac{\alpha}{m} \|\underline{\mathbf{y}}^r - \hat{\underline{\mathbf{y}}}^r\|^2$. Then for suitably chosen $\eta, \alpha$ the potential function is non-increasing over timesteps and specifically there exist positive constants $C_1, C_2, C_3$ such that*

$$H(\underline{\mathbf{x}}^{r+1}, \underline{\mathbf{y}}^{r+1}) - H(\underline{\mathbf{x}}^r, \underline{\mathbf{y}}^r) \leq -C_1 \|\hat{\mathbf{y}}^r\|^2 - C_2 \frac{1}{m} \|\underline{\mathbf{x}}^r - \hat{\underline{\mathbf{x}}}^r\|^2 - C_3 \frac{1}{m} \|\underline{\mathbf{y}}^r - \hat{\underline{\mathbf{y}}}^r\|^2 \qquad (27)$$

*Proof.*

$$H(\underline{\mathbf{x}}^{r+1}, \underline{\mathbf{y}}^{r+1}) - H(\underline{\mathbf{x}}^r, \underline{\mathbf{y}}^r) \tag{28}$$

$$= \frac{1}{m} \sum_{i=1}^{m} f_i(\hat{\mathbf{x}}^{r+1}) - \frac{1}{m} \sum_{i=1}^{m} f_i(\hat{\mathbf{x}}^r) + \frac{1}{m} (\|\underline{\mathbf{x}}^{r+1} - \hat{\underline{\mathbf{x}}}^{r+1}\|^2 - \|\underline{\mathbf{x}}^r - \hat{\underline{\mathbf{x}}}^r\|^2) \tag{29}$$

$$+ \frac{\alpha}{m} (\|\underline{\mathbf{y}}^{r+1} - \hat{\underline{\mathbf{y}}}^{r+1}\|^2 - \|\underline{\mathbf{y}}^r - \hat{\underline{\mathbf{y}}}^r\|^2) \tag{30}$$

$$= (\frac{1}{m} \sum_{i=1}^{m} f_i(\hat{\mathbf{x}}^{r+1}) - \frac{1}{m} \sum_{i=1}^{m} f_i(\hat{\mathbf{x}}^r)) + P_\alpha(\underline{\mathbf{x}}^{r+1}) - P_\alpha(\underline{\mathbf{x}}^r) \tag{31}$$

$$\leq \left( (1 + \beta_1)\sigma^2 - 1 + \alpha \left( 8L_1^2 (1 + \frac{1}{\beta_2}) \right) + \eta \frac{L_1^2}{2} \right) \frac{1}{m} \|\underline{\mathbf{x}}^r - \hat{\underline{\mathbf{x}}}^r\|^2 \tag{32}$$

$$+ \left( \alpha((1 + \beta_2)\sigma^2 - 1) + \eta^2 (1 + \frac{1}{\beta_1}) + \alpha\eta^2 (4L_1^2 (1 + \frac{1}{\beta_2})) \right) \frac{1}{m} \|\underline{\mathbf{y}}^r - \hat{\underline{\mathbf{y}}}^r\|^2 \tag{33}$$

$$+ \left( -\frac{\eta}{2} + \eta^2 \frac{L_1^2}{2} + \alpha\eta^2 \left( 4L_1^2 (1 + \frac{1}{\beta_2}) \right) \right) \|\hat{\mathbf{y}}^r\|^2 \tag{34}$$

The inequality follows from the results of Lemmas 5 and 6. By choosing $\eta$ and $\alpha$ sufficiently small we can ensure that the Lyapunov function $H$ is decreasing. In particular, we need to ensure that

$$(1 + \beta_1)\sigma^2 + \alpha\left(8L_1^2(1 + \frac{1}{\beta_2})\right) + \eta\frac{L_1^2}{2} < 1 \tag{35}$$

$$\alpha(1 + \beta_2)\sigma^2 + \eta^2\left(1 + \frac{1}{\beta_1}\right) + 4\alpha\eta^2 L_1^2\left(1 + \frac{1}{\beta_2}\right) < \alpha \tag{36}$$

$$\eta L_1^2 + 8\alpha\eta L_1^2\left(1 + \frac{1}{\beta_2}\right) < 1 \tag{37}$$

To simplify these conditions we set $\beta_1 = \beta_2 = \frac{1-\sigma}{\sigma} > 0$ which leads to the following conditions:

$$\sigma + \frac{\alpha 8L_1^2}{1 - \sigma} + \eta\frac{L_1^2}{2} < 1 \tag{38}$$

$$\alpha\sigma + \frac{\eta^2}{1 - \sigma} + \frac{4\alpha\eta^2 L_1^2}{1 - \sigma} < \alpha \tag{39}$$

$$\eta L_1^2 + \frac{8\alpha\eta L_1^2}{1 - \sigma} < 1 \tag{40}$$

Indeed, if $\alpha$ and $\eta$ are sufficiently small these conditions are satisfied. But, to obtain an explicit rate we assume that $\alpha$ satisfies the following inequality

$$\alpha \leq \frac{(1 - \sigma)^2}{16L_1^2}, \tag{41}$$

and $\eta$ as a result satisfies the following conditions

$$\eta \leq \frac{(1 - \sigma)}{2L_1^2} \tag{42}$$

$$\eta^2 \leq \frac{\alpha(1 - \sigma)^2}{2 + 8\alpha L_1^2} \tag{43}$$

$$\eta \leq \frac{1 - \sigma}{2L_1^2(1 - \sigma + 8\alpha)} \tag{44}$$

If we assume these four conditions hold and $\beta_1 = \beta_2 = \frac{1-\sigma}{\sigma}$, then we can obtain

$$H(\underline{\mathbf{x}}^{r+1}, \underline{\mathbf{y}}^{r+1}) - H(\underline{\mathbf{x}}^r, \underline{\mathbf{y}}^r) \leq -\frac{1 - \sigma}{4m}\|\underline{\mathbf{x}}^r - \hat{\underline{\mathbf{x}}}^r\|^2 - \frac{\alpha(1 - \sigma)}{2m}\|\underline{\mathbf{y}}^r - \hat{\underline{\mathbf{y}}}^r\|^2 - \frac{\eta}{4}\|\hat{\underline{\mathbf{y}}}^r\|^2 \tag{45}$$

Hence, $C_1 = \frac{\eta}{4}$, $C_2 = \frac{1-\sigma}{4}$, and $C_3 = \alpha\frac{1-\sigma}{2}$ $\qquad\square$

**Corollary 2.** . *Assume the conditions for Lemma 7 hold, then we immediately get*

$$H(\underline{\mathbf{x}}^0, \underline{\mathbf{y}}^0) - H(\underline{\mathbf{x}}^{r+1}, \underline{\mathbf{y}}^{r+1}) \geq C_1\sum_{t=0}^{r}\|\hat{\underline{\mathbf{y}}}^t\|^2 + C_2\frac{1}{m}\sum_{t=0}^{r}\|\underline{\mathbf{x}}^t - \hat{\underline{\mathbf{x}}}^t\|^2 + C_3\frac{1}{m}\sum_{t=0}^{r}\|\underline{\mathbf{y}}^t - \hat{\underline{\mathbf{y}}}^t\|^2 \tag{46}$$

**Theorem 4.** *Assume the iterates $\mathbf{x}_i$ follow the Gradient Tracking Update with stepsize $\eta_1$ such that $\eta_1, \alpha$ satisfy conditions (41) - (44). Also assume $\tilde{t}$ is sampled from the uniform distribution over $[0, T-1]$. Then we can bound the expectation of the sum of the square of the global gradient estimate and the square of the consensus error as follows:*

$$\mathbb{E}_{\tilde{t}}\left[\left\|\frac{1}{m}\sum_{i=1}^{m}\nabla f_i(\mathbf{x}_i^{\tilde{t}})\right\|^2 + \frac{1}{m}\left\|\underline{\mathbf{x}}^{\tilde{t}} - \hat{\underline{\mathbf{x}}}^{\tilde{t}}\right\|^2\right] \leq \frac{4}{\min\{\eta, 1 - \sigma\}}\frac{f(\mathbf{x}^0) - f^*}{T} \tag{47}$$

*Proof.*

$$\mathbb{E}_{\tilde{t}}\left[\left\|\frac{1}{m}\sum_{i=1}^{m}\nabla f_i(\mathbf{x}_i^{\tilde{t}})\right\|^2 + \frac{1}{m}\left\|\underline{\mathbf{x}}^{\tilde{t}} - \hat{\underline{\mathbf{x}}}^{\tilde{t}}\right\|^2\right]$$

$$\leq \frac{1}{T}\left(\sum_{t=0}^{T-1}\left\|\frac{1}{m}\sum_{i=1}^{m}\nabla f_i(\mathbf{x}_i^{t})\right\|^2 + \sum_{t=0}^{T-1}\frac{1}{m}\left\|\underline{\mathbf{x}}^{t} - \hat{\underline{\mathbf{x}}}^{t}\right\|^2 + \sum_{t=0}^{T-1}\frac{\alpha}{m}\left\|\underline{\mathbf{y}}^{t} - \hat{\underline{\mathbf{y}}}^{t}\right\|^2\right)$$

$$\leq \frac{1}{TC_0}\left(\frac{\eta_1}{4}\sum_{t=0}^{T-1}\left\|\hat{\mathbf{y}}^{t}\right\|^2 + \frac{1-\sigma}{4}\sum_{t=0}^{T-1}\frac{1}{m}\left\|\underline{\mathbf{x}}^{t} - \hat{\underline{\mathbf{x}}}^{t}\right\|^2 + \frac{1-\sigma}{2}\sum_{t=0}^{T-1}\frac{\alpha}{m}\left\|\underline{\mathbf{y}}^{t} - \hat{\underline{\mathbf{y}}}^{t}\right\|^2\right)$$

$$\leq \frac{1}{C_0}\frac{H(\underline{\mathbf{x}}^0,\underline{\mathbf{y}}^0) - H(\underline{\mathbf{x}}^T,\underline{\mathbf{y}}^T)}{T}$$

$$\leq \frac{1}{C_0}\frac{f(\mathbf{x}^0) - f^*}{T}$$

$$= \frac{1}{\min\{\frac{\eta_1}{4},\frac{1-\sigma}{4}\}}\frac{f(\mathbf{x}^0) - f^*}{T}$$

where $C_0 := \min\{\frac{\eta_1}{4},\frac{1-\sigma}{4}\}$. The last inequality holds because

$$H(\underline{\mathbf{x}}^0,\underline{\mathbf{y}}^0) := f(\hat{\mathbf{x}}^0) + \frac{1}{m}\left\|\underline{\mathbf{x}}^0 - \hat{\underline{\mathbf{x}}}^0\right\|^2 + \frac{\alpha}{m}\left\|\underline{\mathbf{y}}^0 - \hat{\underline{\mathbf{y}}}^0\right\|^2 = f(\hat{\mathbf{x}}^0) \tag{48}$$

$$H(\underline{\mathbf{x}}^r,\underline{\mathbf{y}}^r) := f(\hat{\mathbf{x}}^r) + \frac{1}{m}\left\|\underline{\mathbf{x}}^r - \hat{\underline{\mathbf{x}}}^r\right\|^2 + \frac{\alpha}{m}\left\|\underline{\mathbf{y}}^r - \hat{\underline{\mathbf{y}}}^r\right\|^2 \geq f(\hat{\mathbf{x}}^r) \geq f^* \tag{49}$$

$\square$

Thus we immediately have:

**Corollary 3.** *To achieve the following $\epsilon$-stationary solution for the separable version of our problem,*

$$\mathbb{E}_{\tilde{t}}\left[\left\|\frac{1}{m}\sum_{i=1}^{m}\nabla f_i(\mathbf{x}_i^{\tilde{t}})\right\|^2 + \frac{1}{m}\left\|\underline{\mathbf{x}}^{\tilde{t}} - \hat{\underline{\mathbf{x}}}^{\tilde{t}}\right\|^2\right] \leq \epsilon^2 \tag{50}$$

*using the Gradient Tracking Algorithm satisfying the conditions of Theorem 4 we require $T \geq \frac{4(f(\mathbf{x}^0)-f^*)}{\min\{\eta_1,1-\sigma\}\epsilon^2} + 1$ communication steps.*

Let us call timestep $\tilde{t}$ a **good choice** if $\left\|\frac{1}{m}\sum_{i=1}^{m}\nabla f_i(\mathbf{x}_i^{\tilde{t}})\right\|^2 + \frac{1}{m}\left\|\underline{\mathbf{x}}^{\tilde{t}} - \hat{\underline{\mathbf{x}}}^{\tilde{t}}\right\|^2 \leq \frac{\epsilon^2}{4}$ and a **bad choice** otherwise.

**Lemma 8.** *Assume the iterates $\mathbf{x}_i$ follow the Gradient Tracking Update with stepsize $\eta_1$ such that $\eta_1, \alpha$ satisfy conditions (41) - (44). Let $T \geq 4e\frac{4(f(\mathbf{x}^0)-f^*)}{\min\{\eta_1,1-\sigma\}\epsilon^2} + 1$ Also assume $\tilde{t}$ is sampled from the uniform distribution over $[0, T-1]$. Then*

$$\mathbf{P}\left(\left[\left\|\frac{1}{m}\sum_{i=1}^{m}\nabla f_i(\mathbf{x}_i^{\tilde{t}})\right\|^2 + \frac{1}{m}\left\|\underline{\mathbf{x}}^{\tilde{t}} - \hat{\underline{\mathbf{x}}}^{\tilde{t}}\right\|^2\right] \geq \frac{\epsilon^2}{4}\right) \leq \frac{1}{e} \tag{51}$$

*Proof.* From Corollary 3 we have

$$\mathbb{E}_{\tilde{t}}\left[\left\|\frac{1}{m}\sum_{i=1}^{m}\nabla f_i(\mathbf{x}_i^{\tilde{t}})\right\|^2 + \frac{1}{m}\left\|\underline{\mathbf{x}}^{\tilde{t}} - \hat{\underline{\mathbf{x}}}^{\tilde{t}}\right\|^2\right] \leq \frac{\epsilon^2}{4e} \tag{52}$$

Then we apply Markov's inequality and derive

$$\mathbf{P}\left(\left[\left\|\frac{1}{m}\sum_{i=1}^{m}\nabla f_i(\mathbf{x}_i^{\tilde{t}})\right\|^2 + \frac{1}{m}\left\|\underline{\mathbf{x}}^{\tilde{t}} - \hat{\underline{\mathbf{x}}}^{\tilde{t}}\right\|^2\right] \geq \frac{\epsilon^2}{4}\right) \leq \frac{\frac{\epsilon^2}{4e}}{\frac{\epsilon^2}{4}} = \frac{1}{e} \tag{53}$$

$\square$

**Theorem 5.** *Assume the iterates $\mathbf{x}_i$ follow the Gradient Tracking Update with stepsize $\eta_1$ such that $\eta_1, \alpha$ satisfy conditions (41) - (44). Let $T \geq 4e\frac{4(f(\mathbf{x}^0)-f^*)}{\min\{\eta_1, 1-\sigma\}\epsilon^2} + 1$ and assume we pick i.i.d. random variables $\tilde{t}_1, \tilde{t}_2, ..., \tilde{t}_{\log(\frac{1}{\delta_1})}$ sampled from the uniform distribution over $[0, T-1]$. Then the probability that at least one of them is a good choice is at least $1 - \delta_1$.*

*Proof.* From Lemma 8 we know that the probability of picking a bad choice it at most $\frac{1}{e}$. Thus the probability all $\log(\delta_1)$ of them are bad choices is at most $\frac{1}{e^{\log(\frac{1}{\delta_1})}} = \delta_1$. It follows that the probability that we pick at least one good choice is at least $1 - \delta_1$. $\square$

**Remark 2.** *In order to check each of these $\log\left(\frac{1}{\delta_1}\right)$ iterates we invoke the average consensus protocol with communication complexity at most $4\left(\frac{c\log(\frac{1}{\epsilon})+\log(\frac{1}{\alpha})+\log\left(4m^2(f(\mathbf{x}^0)-f^*+2\mathcal{F})\right)}{\log(\frac{1}{\sigma})} + 1\right)$ as reported in Corollary 9, with $\mathcal{F}$ defined in (60). Thus the overall number of rounds is $\log\left(\frac{1}{\delta_1}\right)\cdot 4\left(\frac{c\log(\frac{1}{\epsilon})+\log(\frac{1}{\alpha})+\log\left(4m^2(f(\mathbf{x}^0)-f^*+2\mathcal{F})\right)}{\log(\frac{1}{\sigma})} + 1\right)$ which is negligible compared to the number of rounds of phase I, which is $4e\frac{4(f(\mathbf{x}^0)-f^*)}{\min\{\eta_1, 1-\sigma\}\epsilon^2} + 1$.*

## 10 Escaping a first order stationary point with negative curvature

Let us denote with $\underline{\mathbf{x}}^{-1}$ the iterate that is returned by the first phase. From here on assume that we have the following bounds:

$$\left\|\frac{1}{m}\sum_{i=1}^{m}\nabla f_i(\mathbf{x}_i^{-1})\right\|^2 \leq \epsilon_1^2 \tag{54}$$

$$\frac{1}{m}\left\|\underline{\mathbf{x}}^{-1} - \hat{\underline{\mathbf{x}}}^{-1}\right\|^2 \leq \epsilon_2^2 \tag{55}$$

For the first phase to return w.p. $1 - \delta_1$ a point that satisfies the condition in (54) we need $4e\frac{4(f(\mathbf{x}^0)-f^*)}{\min\{\eta_1, 1-\sigma\}\epsilon_1^2} + 1 + \log(\frac{1}{\delta_1})\cdot 4\left(\frac{c\log(\frac{1}{\epsilon})+\log(\frac{1}{\alpha})+\log\left(4m^2(f(\mathbf{x}^0)-f^*+2\mathcal{F})\right)}{\log(\frac{1}{\sigma})} + 1\right)$ iterations.

For the first phase to return w.p. $1 - \delta_1$ a point that satisfies the condition in (55) we need $4e\frac{4(f(\mathbf{x}^0)-f^*)}{\min\{\eta_1, 1-\sigma\}\epsilon_2^2} + 1 + \log(\frac{1}{\delta_1})\cdot 4\left(\frac{c\log(\frac{1}{\epsilon})+\log(\frac{1}{\alpha})+\log\left(4m^2(f(\mathbf{x}^0)-f^*+2\mathcal{F})\right)}{\log(\frac{1}{\sigma})} + 1\right)$ iterations.

Hence

**Corollary 4.** *Assume the first phase runs Gradient Tracking with $\eta_1, \alpha$ such that they satisfy conditions (41) - (44). For the first phase to return a point that satisfies conditions (54) and (55) with probability $1 - \delta_1$ we need to run at most*

$$T_1 = 4e\frac{4(f(\mathbf{x}^0) - f^*)}{\min\{\eta_1, 1 - \sigma\}\min\{\epsilon_1^2, \epsilon_2^2\}} + 1$$
$$+ 4\log(\frac{1}{\delta_1})\cdot\left(\frac{c\log(\frac{1}{\epsilon}) + \log(\frac{1}{\alpha}) + \log\left(4m^2(f(\mathbf{x}^0) - f^* + 2\mathcal{F})\right)}{\log(\frac{1}{\sigma})} + 1\right) \tag{56}$$

*iterations . Thus*

$$T_1 = \tilde{\mathcal{O}}\left(\frac{1}{\eta_1 \min\{\epsilon_1^2, \epsilon_2^2\}}\right)$$

Notice that if $\underline{\mathbf{x}}^{-1}$ satisfies conditions (54) and (55) then $\hat{\mathbf{x}}$ is either an $(\epsilon_1 + L_1\epsilon_2, \gamma)$-second order stationary point or it has sufficient negative curvature i.e. $\lambda_{\min}\left(\nabla^2 f(\hat{\mathbf{x}}^{-1})\right) \leq -\gamma$. In the former case it suffices to report the iterate; we will now focus on the more involved latter case and specifically we are going to show that after injecting noise to the iterates $\mathbf{x}_i$'s and restarting the gradient tracking algorithm the potential function decreases substantially after a small number of iterations.

We start the second phase of our algorithm by injecting the same noise $\xi$, uniformly from the ball of radius $R$, to all local iterates.

$$\mathbf{x}_i^0 = \mathbf{x}_i^{-1} + \xi \tag{57}$$

Then we run the average consensus protocol on $\nabla f_i(\mathbf{x}_i^0)$'s for sufficiently large number of iterations in order to get $\mathbf{y}_i^0$'s such that

$$\hat{\mathbf{y}}^0 = \frac{1}{m}\sum_{i=1}^m \nabla f_i(\mathbf{x}_i^0) \quad and \quad \frac{1}{m}\left\|\underline{\mathbf{y}}^0 - \hat{\underline{\mathbf{y}}}^0\right\|^2 \leq \frac{L_1^2}{2(1-\sigma)}\eta_2\mathcal{F}, \tag{58}$$

$\eta_2$ is the stepsize of phase II and $\mathcal{F}$ is defined below. As presented in Lemma 20, the required number of iterations is $\left\lceil \frac{\log\left(\sqrt{\eta_2\mathcal{F}}\right) - \log\left(\left\|\underline{\mathbf{y}}^0 - \hat{\underline{\mathbf{y}}}^0\right\|\right) + \log\left(\sqrt{m}L_1\right) - \log\left(\sqrt{2(1-\sigma)}\right)}{\log(\sigma)} \right\rceil$ and is negligible compared to the number of iterations of phase I.

Consequently, the process follows the gradient tracking update for $T_{cap}$ iterations with stepsize $\eta_2$ such that $\eta_2, \alpha$ satisfy conditions (41) - (44). We also have the following useful quantities:

$$H := \nabla^2 f(\hat{\mathbf{x}}^{-1}) \tag{59}$$

$$\mathcal{F} := \frac{|\lambda_{\min}(H)|^3}{\log^3(d\kappa/\delta_2)}\frac{(\sqrt{2}-1)^2}{(24\sqrt{2}L_2\hat{c}^2)^2} \tag{60}$$

$$\mathcal{P} := \frac{|\lambda_{\min}(H)|}{\log(d\kappa/\delta_2)}\frac{\sqrt{2}-1}{(24\sqrt{2}L_2\hat{c}^2)} \tag{61}$$

$$\mathcal{J} := \frac{\log(d\kappa/\delta_2)}{\eta_2|\lambda_{\min}(H)|} \tag{62}$$

$$T_{cap} := \hat{c}\mathcal{J} \tag{63}$$

$$\mathcal{R} := \sqrt{\frac{\mathcal{F}}{L_1}} \tag{64}$$

$$\kappa := \frac{L_1}{\gamma} \tag{65}$$

$$\delta_2 \in \left(0, \frac{d\kappa}{e}\right] \tag{66}$$

Where $\mathcal{F}$ is the target decrease of the potential function, $P$ the bound on the norm of the iterates, $T_{cap}$ the number of iterations in the second phase, $R$ the radius of the ball, $\kappa$ the condition number, $d$ the dimension and $\delta_2$ the probability of failure; $\hat{c}$ is a positive constant to be defined later.

We proceed in the following lemma to show that if the norm of the global gradient and the consensus errors are small then the norm of the gradient of the average iterate returned by phase 1 is also small.

**Lemma 9.** *Suppose that conditions* (54), (55) *hold and* $\epsilon_1^2 \leq \mathcal{F}\frac{L_1}{2+2L_1^2}$ *and* $\epsilon_2^2 \leq \mathcal{F}\frac{L_1}{2+2L_1^2}$. *Then we can show that*

$$\left\|\nabla f(\hat{\mathbf{x}}^{-1})\right\| \leq \sqrt{L_1\mathcal{F}}. \tag{67}$$

.

*Proof.* Adding and subtracting the same term derives:

$$\left\|\nabla f(\hat{\mathbf{x}}^{-1})\right\|^2 \le 2\left\|\frac{1}{m}\sum_{i=1}^{m}\nabla f_i(\mathbf{x}_i^{-1}) - \frac{1}{m}\sum_{i=1}^{m}\nabla f_i(\hat{\mathbf{x}}^{-1})\right\|^2 + 2\left\|\frac{1}{m}\sum_{i=1}^{m}\nabla f_i(\mathbf{x}_i^{-1})\right\|^2 \tag{68}$$

$$\le 2\frac{L_1^2}{m}\sum_{i=1}^{m}\left\|\mathbf{x}_i^{-1} - \hat{\mathbf{x}}^{-1}\right\|^2 + 2\epsilon_1^2 \tag{69}$$

$$\le 2L_1^2\epsilon_2^2 + 2\epsilon_1^2 \tag{70}$$

$$\le L_1\mathcal{F} \tag{71}$$

Thus

$$\left\|\nabla f(\hat{\mathbf{x}}^{-1})\right\| \le \sqrt{L_1\mathcal{F}} \tag{72}$$

Where the second inequality comes from (54) and the third from (55). □

Utilizing the previous result we are going to show that by adding perturbation in the worst case we increase the function value at most by $\frac{3}{2}\mathcal{F}$.

**Lemma 10.** *Suppose that conditions (54), (55) hold and let $\epsilon_1^2 \le \mathcal{F}\frac{L_1}{2+2L_1^2}$ and $\epsilon_2^2 \le \mathcal{F}\frac{L_1}{2+2L_1^2}$ and for all $i$ let $\mathbf{x}_i^0 = \mathbf{x}_i^{-1} + \xi$ where $\xi$ comes from the uniform distribution over the ball of radius $R = \sqrt{\frac{\mathcal{F}}{L_1}}$. Then*

$$f(\hat{\mathbf{x}}^0) - f(\hat{\mathbf{x}}^{-1}) \le \frac{3}{2}\mathcal{F} \tag{73}$$

*Proof.* First notice that by Lemma 9 we have $\left\|\nabla f(\hat{\mathbf{x}}^{-1})\right\| \le \sqrt{L_1\mathcal{F}}$ and thus utilizing smoothness we obtain the bound

$$f(\hat{\mathbf{x}}^0) - f(\hat{\mathbf{x}}^{-1}) \le \langle \nabla f(\hat{\mathbf{x}}^{-1}), \xi \rangle + \frac{L_1}{2}\left\|\xi\right\|^2 \le \sqrt{L_1\mathcal{F}}\frac{\sqrt{\mathcal{F}}}{\sqrt{L_1}} + \frac{\mathcal{F}}{2} \le \frac{3}{2}\mathcal{F} \tag{74}$$

□

Below we show that the potential function increases at most by $\frac{7}{4}\mathcal{F}$ after the injection of noise.

**Lemma 11.** *Suppose that conditions (54), (55) and (58) hold and $\epsilon_1^2 \le \mathcal{F}\frac{L_1}{2+2L_1^2}$, $\epsilon_2^2 \le \mathcal{F}\frac{L_1}{2+2L_1^2}$. Further let $\alpha\eta_2 \le \frac{(1-\sigma)}{2L_1^2}$ and for all $i$ let $\mathbf{x}_i^0 = \mathbf{x}_i^{-1} + \xi$ where $\xi$ comes from the uniform distribution over the ball of radius $R = \sqrt{\frac{\mathcal{F}}{L_1}}$. Then we have the following*

$$H(\underline{\mathbf{x}}^0, \underline{\mathbf{y}}^0) - H(\underline{\mathbf{x}}^{-1}, \underline{\mathbf{y}}^{-1}) \le \frac{7}{4}\mathcal{F} \tag{75}$$

*Proof.* Recall that

$$H(\underline{\mathbf{x}}^r, \underline{\mathbf{y}}^r) := \frac{1}{m}\sum_{i=1}^{m}f_i(\hat{\mathbf{x}}^r) + \frac{1}{m}\left\|\underline{\mathbf{x}}^r - \hat{\underline{\mathbf{x}}}^r\right\|^2 + \frac{\alpha}{m}\left\|\underline{\mathbf{y}}^r - \hat{\underline{\mathbf{y}}}^r\right\|^2 \tag{76}$$

By the definition of the potential function we get

$$H(\underline{\mathbf{x}}^0, \underline{\mathbf{y}}^0) - H(\underline{\mathbf{x}}^{-1}, \underline{\mathbf{y}}^{-1})$$

$$= f(\hat{\mathbf{x}}^0) - f(\hat{\mathbf{x}}^{-1}) + \frac{1}{m}\left\|\underline{\mathbf{x}}^0 - \hat{\underline{\mathbf{x}}}^0\right\|^2 - \frac{1}{m}\left\|\underline{\mathbf{x}}^{-1} - \hat{\underline{\mathbf{x}}}^{-1}\right\|^2 + \frac{\alpha}{m}\left\|\underline{\mathbf{y}}^0 - \hat{\underline{\mathbf{y}}}^0\right\|^2 - \frac{\alpha}{m}\left\|\underline{\mathbf{y}}^{-1} - \hat{\underline{\mathbf{y}}}^{-1}\right\|^2$$

$$\le f(\hat{\mathbf{x}}^0) - f(\hat{\mathbf{x}}^{-1}) + \frac{\alpha}{m}\left\|\underline{\mathbf{y}}^0 - \hat{\underline{\mathbf{y}}}^0\right\|^2$$

$$\le \frac{3}{2}\mathcal{F} + \alpha\eta_2\frac{L_1^2}{2(1-\sigma)}\mathcal{F} \tag{77}$$

$$\le \frac{3}{2}\mathcal{F} + \frac{1}{4}\mathcal{F} \tag{78}$$

$$\le \frac{7}{4}\mathcal{F}, \tag{79}$$

where the first inequality comes from the fact that the same noise is injected to all local iterates and thus

$$\frac{1}{m}\|\underline{\mathbf{x}}^0 - \hat{\underline{\mathbf{x}}}^0\|^2 - \frac{1}{m}\|\underline{\mathbf{x}}^{-1} - \hat{\underline{\mathbf{x}}}^{-1}\|^2 = 0 \tag{80}$$

for the second inequality we use that $f(\hat{\mathbf{x}}^0) - f(\hat{\mathbf{x}}^{-1}) \leq \frac{3}{2}\mathcal{F}$ due to Lemma 10 and the bound from (58). $\qquad\square$

Having established the fact that the potential function is not increasing more than $\frac{7}{4}\mathcal{F}$ after the injection of noise, we can proceed to show that after we perturbed the iterates and apply Gradient Tracking Update for $T_{cap}$ iterations, the potential function will decrease substantially. Specifically, we will show that $H(\underline{\mathbf{x}}^{T_{cap}}, \underline{\mathbf{y}}^{T_{cap}}) - H(\underline{\mathbf{x}}^{-1}, \underline{\mathbf{y}}^{-1}) \leq -\mathcal{F}$. Towards proving this statement we consider two complementary cases.

In the first, more simple case we assume that at least one of the following sums is sufficiently large

$$\sum_{t=0}^{T_{cap}-1} \|\hat{\underline{\mathbf{y}}}^t\|^2 \geq \frac{12\mathcal{F}}{\eta_2}, \qquad \sum_{t=0}^{T_{cap}-1} \|\underline{\mathbf{x}}^t - \hat{\underline{\mathbf{x}}}^t\|^2 \geq \frac{12m}{(1-\sigma)}\mathcal{F} \tag{81}$$

and proceed to prove the potential function decrease.

**Lemma 12.** *Suppose that conditions (54), (55) and (58) hold and $\epsilon_1^2 \leq \mathcal{F}\frac{L_1}{2+2L_1^2}$, $\quad \epsilon_2^2 \leq \mathcal{F}\frac{L_1}{2+2L_1^2}$. Further let $\alpha\eta_2 \leq \frac{(1-\sigma)}{4L_1^2}$ and for all $i$ let $\mathbf{x}_i^0 = \mathbf{x}_i^{-1} + \xi$ where $\xi$ comes from the uniform distribution over the ball of radius $R = \sqrt{\frac{\mathcal{F}}{L_1}}$. Assume the iterates $\mathbf{x}_i$ follow the Gradient Tracking Update with stepsize $\eta_2$ such that $\eta_2, \alpha$ satisfy conditions (41) - (44). Finally, let at least one of the following sums be large enough*

$$\sum_{t=0}^{T_{cap}-1} \|\hat{\underline{\mathbf{y}}}^t\|^2 \geq \frac{12\mathcal{F}}{\eta_2}, \qquad \sum_{t=0}^{T_{cap}-1} \|\underline{\mathbf{x}}^t - \hat{\underline{\mathbf{x}}}^t\|^2 \geq \frac{12m}{1-\sigma}\mathcal{F} \tag{82}$$

*Then we can show that*

$$H(\underline{\mathbf{x}}^{T_{cap}}, \underline{\mathbf{y}}^{T_{cap}}) - H(\underline{\mathbf{x}}^{-1}, \underline{\mathbf{y}}^{-1}) \leq -\mathcal{F}$$

*Proof.* From (46) we get

$$H(\underline{\mathbf{x}}^{T_{cap}}, \underline{\mathbf{y}}^{T_{cap}}) - H(\underline{\mathbf{x}}^0, \underline{\mathbf{y}}^0) \leq -\frac{\eta_2}{4}\sum_{t=0}^{T_{cap}-1}\|\hat{\underline{\mathbf{y}}}^t\|^2 - \frac{1-\sigma}{4m}\sum_{t=0}^{T_{cap}-1}\|\underline{\mathbf{x}}^t - \hat{\underline{\mathbf{x}}}^t\|^2$$

$$- \alpha\frac{1-\sigma}{2m}\sum_{t=0}^{T_{cap}-1}\|\underline{\mathbf{y}}^t - \hat{\underline{\mathbf{y}}}\|^2 \tag{83}$$

$$\leq -\frac{\eta_2}{4}\sum_{t=0}^{T_{cap}-1}\|\hat{\underline{\mathbf{y}}}^t\|^2 - \frac{1-\sigma}{4m}\sum_{t=0}^{T_{cap}-1}\|\underline{\mathbf{x}}^t - \hat{\underline{\mathbf{x}}}^t\|^2$$

$$\leq -3\mathcal{F} \tag{84}$$

Thus immediately we get

$$H(\underline{\mathbf{x}}^{T_{cap}}, \underline{\mathbf{y}}^{T_{cap}}) - H(\underline{\mathbf{x}}^{-1}, \underline{\mathbf{y}}^{-1}) = H(\underline{\mathbf{x}}^{T_{cap}}, \underline{\mathbf{y}}^{T_{cap}}) - H(\underline{\mathbf{x}}^0, \underline{\mathbf{y}}^0) + H(\underline{\mathbf{x}}^0, \underline{\mathbf{y}}^0) - H(\underline{\mathbf{x}}^{-1}, \underline{\mathbf{y}}^{-1}) \leq -3\mathcal{F} + \frac{7}{4}\mathcal{F} \leq -\mathcal{F} \tag{85}$$

Where in the first inequality we use Lemma 11. $\qquad\square$

We are left to deal with the complementary case where both sums are sufficiently small:

$$\sum_{t=0}^{T_{cap}-1} \|\hat{\underline{\mathbf{y}}}^t\|^2 < \frac{12\mathcal{F}}{\eta_2} \quad and \quad \sum_{t=0}^{T_{cap}-1} \|\underline{\mathbf{x}}^t - \hat{\underline{\mathbf{x}}}^t\|^2 < \frac{12m}{1-\sigma}\mathcal{F} \tag{86}$$

The high level idea of the next lemma is the following. For the first $T_{cap}$ iteration either the $\hat{\mathbf{x}}^t$ iterates are going to decrease the function value by at least $3\mathcal{F}$ or the iterates are going to remain in a ball of radius $2\hat{c}\mathcal{P}$ around $\hat{\mathbf{x}}^{-1}$.

**Lemma 13.** *Assume that* (86) *holds and that we are given* $\underline{\mathbf{x}}^{-1}, \underline{\mathbf{x}}^{-0}$ *such that* $\|\hat{\mathbf{x}}^{-1} - \hat{\mathbf{x}}^0\| \leq 2\mathcal{R}$. *Assume that* $\underline{\mathbf{x}}^t$ *follows the Gradient Tracking Update with stepsize* $\eta_2 \leq \min\{1, \frac{1}{L_1}\}$ *and let* $\hat{c} \geq 36$. *Further, consider the definition of the stopping time* $T_{\hat{\mathbf{x}}}$

$$T_{\hat{\mathbf{x}}} = \min\{\inf_t\{t | f(\hat{\mathbf{x}}^t) - f(\hat{\mathbf{x}}^0) \leq -3\mathcal{F}\}, \hat{c}\mathcal{J}\}$$

*Then for all time indices* $t < T_{\hat{\mathbf{x}}}$ *we have* $\|\hat{\mathbf{x}}^t - \hat{\mathbf{x}}^0\| \leq \hat{c}\mathcal{P}$ *and as a result* $\|\hat{\mathbf{x}}^t - \hat{\mathbf{x}}^{-1}\| \leq 2\hat{c}\mathcal{P}$.

*Proof.* For all steps $t \geq 0$ we have

$$f(\hat{\mathbf{x}}^{t+1}) - f(\hat{\mathbf{x}}^t) \tag{87}$$

$$\leq \nabla f(\hat{\mathbf{x}}^t)^T(\hat{\mathbf{x}}^{t+1} - \hat{\mathbf{x}}^t) + \frac{L_1}{2}\|\hat{\mathbf{x}}^{t+1} - \hat{\mathbf{x}}^t\|^2 \tag{88}$$

$$\leq (\nabla f(\hat{\mathbf{x}}^t) - \hat{\mathbf{y}}^t)^T(\hat{\mathbf{x}}^{t+1} - \hat{\mathbf{x}}^t) + (\hat{\mathbf{y}}^t)^T(\hat{\mathbf{x}}^{t+1} - \hat{\mathbf{x}}^t) + \frac{L_1}{2}\|\hat{\mathbf{x}}^{t+1} - \hat{\mathbf{x}}^t\|^2 \tag{89}$$

$$\leq \left(\frac{1}{m}\sum_{i=1}^m \nabla f_i(\hat{\mathbf{x}}^t) - \frac{1}{m}\sum_{i=1}^m \nabla f_i(\mathbf{x}_i^t)\right)^T(\hat{\mathbf{x}}^{t+1} - \hat{\mathbf{x}}^t) - \frac{1}{\eta_2}\|\hat{\mathbf{x}}^{t+1} - \hat{\mathbf{x}}^t\|^2 + \frac{L_1}{2}\|\hat{\mathbf{x}}^{t+1} - \hat{\mathbf{x}}^t\|^2 \tag{90}$$

$$\leq \frac{\eta_2 L_1^2}{m}\|\underline{\mathbf{x}}^t - \hat{\underline{\mathbf{x}}}^t\|^2 + \frac{1}{4\eta_2}\|\hat{\mathbf{x}}^{t+1} - \hat{\mathbf{x}}^t\|^2 - \frac{1}{2\eta_2}\|\hat{\mathbf{x}}^{t+1} - \hat{\mathbf{x}}^t\|^2 \tag{91}$$

$$\leq \frac{\eta_2 L_1^2}{m}\|\underline{\mathbf{x}}^t - \hat{\underline{\mathbf{x}}}^t\|^2 - \frac{1}{4\eta_2}\|\hat{\mathbf{x}}^{t+1} - \hat{\mathbf{x}}^t\|^2 \tag{92}$$

In the second inequality we add and subtrack the same term and in the third inequality we utilize the update rule of gradient tracking. In the forth we utilize smoothness and the fact that $\eta_2 \leq \min\{1, \frac{1}{L_1}\}$. Summing up to any $k < T_{\hat{\mathbf{x}}}$ we obtain

$$f(\hat{\mathbf{x}}^k) - f(\hat{\mathbf{x}}^0) \leq \frac{\eta_2 L_1^2}{m}\sum_{t=0}^{k-1}\|\underline{\mathbf{x}}^t - \hat{\underline{\mathbf{x}}}^t\|^2 - \frac{1}{4\eta_2}\sum_{t=0}^{k-1}\|\hat{\mathbf{x}}^{t+1} - \hat{\mathbf{x}}^t\|^2 \tag{93}$$

$$\leq \frac{\eta_2 L_1^2}{m}\frac{12m}{1-\sigma}\mathcal{F} - \frac{1}{4\eta_2}\sum_{t=0}^{k-1}\|\hat{\mathbf{x}}^{t+1} - \hat{\mathbf{x}}^t\|^2 \tag{94}$$

$$\tag{95}$$

the last inequality holds because $T_{\hat{\mathbf{x}}} \leq T_{cap}$ and also due to the second condition in (86).

Utilizing the fact that $f(\hat{\mathbf{x}}^k) - f(\hat{\mathbf{x}}^0) > -3\mathcal{F}$ we derive

$$12\eta_2\mathcal{F} + \frac{48L_1^2}{1-\sigma}\eta_2^2\mathcal{F} \geq \sum_{t=0}^{k-1}\|\hat{\mathbf{x}}^{t+1} - \hat{\mathbf{x}}^t\|^2 \tag{96}$$

$$12\eta_2\mathcal{F}\left(1 + \frac{4L_1^2\eta_2}{1-\sigma}\right) \geq \sum_{t=0}^{k-1}\|\hat{\mathbf{x}}^{t+1} - \hat{\mathbf{x}}^t\|^2 \tag{97}$$

$$\tag{98}$$

By using the Cauchy-Schwartz inequality we can show that $\sum_{t=0}^{k-1}\|\hat{\mathbf{x}}^{t+1} - \hat{\mathbf{x}}^t\|$ is bounded above by

$$\sum_{t=0}^{k-1}\|\hat{\mathbf{x}}^{t+1} - \hat{\mathbf{x}}^t\| \leq \sqrt{\hat{c}\mathcal{J}\sum_{t=0}^{k-1}\|\hat{\mathbf{x}}^{t+1} - \hat{\mathbf{x}}^t\|^2} \tag{99}$$

$$\leq \sqrt{\hat{c}\frac{\log(d\kappa/\delta_2)}{\eta_2|\lambda_{\min}(H)|}12\eta_2\mathcal{F}\left(1 + \frac{4\eta_2 L_1^2}{1-\sigma}\right)} \tag{100}$$

$$= \sqrt{12\hat{c}\left(1 + \frac{4\eta_2 L_1^2}{1-\sigma}\right)}\sqrt{\frac{\log(d\kappa/\delta_2)\mathcal{F}}{|\lambda_{\min}(H)|}} \tag{101}$$

$$= \sqrt{12\hat{c}\left(1 + \frac{4\eta_2 L_1^2}{1-\sigma}\right)}\mathcal{P} \tag{102}$$

$$\leq \sqrt{36\hat{c}}\mathcal{P} \tag{103}$$

$$\leq \hat{c}\mathcal{P} \tag{104}$$

where the second to last inequality follows from condition (42) and the last from the fact that $\hat{c} \geq 36$.

Next we bound the different between $\hat{\mathbf{x}}^t$ and $\hat{\mathbf{x}}^0$ for all $t \leq k$. In this case we have

$$\left\|\hat{\mathbf{x}}^t - \hat{\mathbf{x}}^0\right\| \leq \sum_{i=1}^{t}\left\|\hat{\mathbf{x}}^i - \hat{\mathbf{x}}^{i-1}\right\| \leq \hat{c}\mathcal{P} \tag{105}$$

Hence, so far we have shown that for all $t \leq k$ we have $\left\|\hat{\mathbf{x}}^t - \hat{\mathbf{x}}^0\right\| \leq \hat{c}\mathcal{P}$.

Next, we proceed to characterize their distance to $\hat{\mathbf{x}}^{-1}$. To do so, first note that $\forall t \leq k-1$ we can derive the following upper bound

$$\left\|\hat{\mathbf{x}}^t - \hat{\mathbf{x}}^{-1}\right\| \leq \sum_{i=1}^{t}\left\|\hat{\mathbf{x}}^i - \hat{\mathbf{x}}^{i-1}\right\| + \left\|\hat{\mathbf{x}}^0 - \hat{\mathbf{x}}^{-1}\right\| \leq \hat{c}\mathcal{P} + 2\mathcal{R} \leq \hat{c}\mathcal{P} + 2\mathcal{P} \leq \frac{3}{2}\hat{c}\mathcal{P} \tag{106}$$

where the third inequality holds given that $L_1 \geq |\lambda_{\min}(H)| \geq \frac{|\lambda_{\min}(H)|}{\log(d\kappa/\delta_2)}$. Notice that since $\delta_2 \in \left(0, \frac{d\kappa}{e}\right]$ we have $\log(d\kappa/\delta_2) \geq 1$. The above implies

$$\mathcal{P} := \sqrt{\frac{\log(d\kappa/\delta_2)\mathcal{F}}{|\lambda_{\min}(H)|}} \geq \sqrt{\frac{\mathcal{F}}{L_1}} = \mathcal{R} \tag{107}$$

and the last inequality of (106) holds since $\hat{c} \geq 4$.
Hence, for all $t < T_{\hat{\mathbf{x}}}$ we have $\left\|\hat{\mathbf{x}}^t - \hat{\mathbf{x}}^{-1}\right\| \leq 2\hat{c}\mathcal{P}$. $\qquad\square$

The next lemma is going to be used in Lemma 15. Here we show that since the consensus error was small before the injection of noise it remains relative small with respect to the sequence $\underline{\mathbf{x}}^t$ and $\underline{\mathbf{w}}^t$ for the first $T_{cap}$ iterations.

**Lemma 14.** *Assume the major condition* (86) *and conditions* (55) *and* (58) *hold. Let* $\epsilon_2^2 \leq \frac{L_1^2}{2(1-\sigma)}\alpha\eta_2\mathcal{F}$ *and further* $\forall i$ *let* $\mathbf{x}_i^0 = \mathbf{x}_i^{-1} + \xi$ *where* $\xi$ *comes from the uniform distribution over the ball of radius* $R = \sqrt{\frac{\mathcal{F}}{L_1}}$. *Consider that the iterates* $\mathbf{x}_i$ *follow the Gradient Tracking Update with stepsize* $\eta_2$ *such that* $\eta_2, \alpha$ *satisfy conditions* (41) - (44). *Define the sequence of* $\mathbf{w}_i$'s *similarly to* $\mathbf{x}_i$'s *except* $\forall i \quad \mathbf{w}_i^0 = \mathbf{x}_i^0 + \mu\mathcal{R}\mathbf{e}_1$ *with* $\mathbf{e}_1$ *a unit eigenvector corresponding to the minimum eigenvalue of* $\nabla^2 f(\hat{\mathbf{x}}^{-1})$ *and* $\mu \in [\delta_2/2\sqrt{d}, 1]$. *Finally, consider a sufficiently large positive constant* $c_{new} \geq 14L_1\sqrt{L_1}$.

*Then for any* $t \leq T_{cap}$ *it holds that*

$$\eta_2\frac{L_1}{m}\sum_{i=1}^{m}(\left\|\hat{\mathbf{w}}^t - \mathbf{w}_i^t\right\| + \left\|\hat{\mathbf{x}}^t - \mathbf{x}_i^t\right\|) \leq c_{new}\sqrt{\frac{\alpha\eta_2}{1-\sigma}}\eta_2\mathcal{R}$$

*Proof.* We will derive a bound with respect to $\mathbf{x}^t$ since the proof for $\mathbf{w}^t$ is identical.
From the proof of Lemma 5 we know that

$$\frac{1}{m}\left\|\underline{\mathbf{x}}^t - \hat{\underline{\mathbf{x}}}^t\right\|^2 - \frac{1}{m}\left\|\underline{\mathbf{x}}^{t-1} - \hat{\underline{\mathbf{x}}}^{t-1}\right\|^2 + \frac{\alpha}{m}\left\|\underline{\mathbf{y}}^t - \hat{\underline{\mathbf{y}}}^t\right\|^2 - \frac{\alpha}{m}\left\|\underline{\mathbf{y}}^{t-1} - \hat{\underline{\mathbf{y}}}^{t-1}\right\|^2 \tag{108}$$

$$\leq \left((1+\beta_1)\sigma^2 - 1 + 8\alpha L_1^2(1 + \frac{1}{\beta_2})\right)\frac{1}{m}\left\|\underline{\mathbf{x}}^{t-1} - \hat{\underline{\mathbf{x}}}^{t-1}\right\|^2$$

$$+ \left(\alpha\left((1+\beta_2)\sigma^2 - 1\right) + \eta_2^2(1 + \frac{1}{\beta_1}) + 4\alpha\eta_2^2 L_1^2(1 + \frac{1}{\beta_2})\right)\frac{1}{m}\left\|\underline{\mathbf{y}}^{t-1} - \hat{\underline{\mathbf{y}}}^{t-1}\right\|^2$$

$$+ 4\alpha\eta_2^2 L_1^2(1 + \frac{1}{\beta_2})\left\|\hat{\mathbf{y}}^{t-1}\right\|^2 \tag{109}$$

$$\tag{110}$$

For $\beta_2 = \frac{1-\sigma}{\sigma}$ and choosing $\eta_2, \alpha$ to satisfy conditions (41) - (44) we guarantee that the coefficients of $\|\underline{\mathbf{x}}^{t-1} - \hat{\underline{\mathbf{x}}}^{t-1}\|^2$ and $\|\underline{\mathbf{y}}^{t-1} - \hat{\underline{\mathbf{y}}}^{t-1}\|^2$ are non-positive.

Thus, we obtain

$$\frac{1}{m}\left\|\underline{\mathbf{x}}^t - \hat{\underline{\mathbf{x}}}^t\right\|^2 - \frac{1}{m}\left\|\underline{\mathbf{x}}^{t-1} - \hat{\underline{\mathbf{x}}}^{t-1}\right\|^2 + \frac{\alpha}{m}\left\|\underline{\mathbf{y}}^t - \hat{\underline{\mathbf{y}}}^t\right\|^2 - \frac{\alpha}{m}\left\|\underline{\mathbf{y}}^{t-1} - \hat{\underline{\mathbf{y}}}^{t-1}\right\|^2 \tag{111}$$

$$\leq 4\alpha\eta_2^2 L_1^2\frac{1}{1-\sigma}\left\|\hat{\mathbf{y}}^{t-1}\right\|^2 \tag{112}$$

Summing up from $1$ to $t$ we get

$$\frac{1}{m}\left\|\underline{\mathbf{x}}^t - \hat{\underline{\mathbf{x}}}^t\right\|^2 - \frac{1}{m}\left\|\underline{\mathbf{x}}^0 - \hat{\underline{\mathbf{x}}}^0\right\|^2 + \frac{\alpha}{m}\left\|\underline{\mathbf{y}}^t - \hat{\underline{\mathbf{y}}}^t\right\|^2 - \frac{\alpha}{m}\left\|\underline{\mathbf{y}}^0 - \hat{\underline{\mathbf{y}}}^0\right\|^2 \tag{113}$$

$$\leq 4\alpha\eta_2^2 L_1^2\frac{1}{1-\sigma}\sum_{i=0}^{t-1}\left\|\hat{\mathbf{y}}^i\right\|^2 \tag{114}$$

$$\leq 4\alpha\eta_2^2 L_1^2\frac{1}{1-\sigma}\left(12\frac{\mathcal{F}}{\eta_2}\right) \tag{115}$$

$$= 48L_1^2\frac{1}{1-\sigma}\alpha\eta_2\mathcal{F} \tag{116}$$

Where the second inequality holds due to (86). The above implies a useful bound for $\frac{1}{m}\left\|\underline{\mathbf{x}}^t - \hat{\underline{\mathbf{x}}}^t\right\|^2$:

$$\frac{1}{m}\left\|\underline{\mathbf{x}}^t - \hat{\underline{\mathbf{x}}}^t\right\|^2 \leq \frac{1}{m}\left\|\underline{\mathbf{x}}^0 - \hat{\underline{\mathbf{x}}}^0\right\|^2 - \frac{\alpha}{m}\left\|\underline{\mathbf{y}}^t - \hat{\underline{\mathbf{y}}}^t\right\|^2 + \frac{\alpha}{m}\left\|\underline{\mathbf{y}}^0 - \hat{\underline{\mathbf{y}}}^0\right\|^2 + 48L_1^2\frac{1}{1-\sigma}\alpha\eta_2\mathcal{F} \tag{117}$$

$$\leq \frac{1}{m}\left\|\underline{\mathbf{x}}^0 - \hat{\underline{\mathbf{x}}}^0\right\|^2 + \frac{\alpha}{m}\left\|\underline{\mathbf{y}}^0 - \hat{\underline{\mathbf{y}}}^0\right\|^2 + 48L_1^2\frac{1}{1-\sigma}\alpha\eta_2\mathcal{F} \tag{118}$$

$$\leq \epsilon_2^2 + \frac{L_1^2}{2(1-\sigma)}\alpha\eta_2\mathcal{F} + 48L_1^2\frac{1}{1-\sigma}\alpha\eta_2\mathcal{F} \tag{119}$$

$$\leq 49L_1^2\frac{1}{1-\sigma}\alpha\eta_2\mathcal{F} \tag{120}$$

where the third inequality comes from (55) and (58) and the forth due to the assumption $\epsilon_2^2 \leq \frac{L_1^2}{2(1-\sigma)}\alpha\eta_2\mathcal{F}$. As immediate corollary we get

$$\frac{1}{\sqrt{m}}\left\|\underline{\mathbf{x}}^t - \hat{\underline{\mathbf{x}}}^t\right\| \leq 7L_1\frac{1}{\sqrt{1-\sigma}}\sqrt{\alpha\eta_2\mathcal{F}} = 7\sqrt{L_1}\frac{1}{\sqrt{1-\sigma}}\sqrt{\alpha\eta_2}\mathcal{R} \tag{121}$$

Finally, notice that

$$\eta_2 \frac{L_1}{m} \sum_{i=1}^{m} \left\| \hat{\mathbf{x}}^t - \mathbf{x}_i^t \right\| \leq \eta_2 \frac{L_1}{m} \sqrt{m \sum_{i=1}^{m} \left\| \hat{\mathbf{x}}^t - \mathbf{x}_i^t \right\|^2} \tag{122}$$

$$\leq \eta_2 \frac{L_1}{\sqrt{m}} \left\| \underline{\mathbf{x}}^t - \underline{\hat{\mathbf{x}}}^t \right\| \tag{123}$$

$$\leq \eta_2 L_1 7 \sqrt{L_1} \frac{1}{\sqrt{1-\sigma}} \sqrt{\alpha \eta_2} \mathcal{R} \tag{124}$$

$$= 7 L_1 \sqrt{L_1} \frac{1}{\sqrt{1-\sigma}} \eta_2 \sqrt{\alpha \eta_2} \mathcal{R} \tag{125}$$

$$\leq \frac{c_{new}}{2} \frac{1}{\sqrt{1-\sigma}} \eta_2 \sqrt{\alpha \eta_2} \mathcal{R} \tag{126}$$

and the last inequality holds for appropriate constant $c_{new} \geq 14 L_1 \sqrt{L_1}$. Since $\mathbf{w}$ is a sequence which develops also with the same parameters of Gradient Tracking the same bounds hold for $\eta_2 \frac{L_1}{m} \sum_{i=1}^{m} \left\| \hat{\mathbf{w}}^t - \mathbf{w}_i^t \right\|$ as well. The result follows.

$\square$

The next lemma shows that if a sequence $\mathbf{x}^t$ does not a escape from the saddle point after $T_{cap}$ iterations of Gradient Tracking then any other sequence $\mathbf{w}^t$ with the same starting point as $\mathbf{x}^t$ will escape if given a little bit of a nudge towards the direction of negative curvature.

**Lemma 15.** *Assume the major condition* (86) *and conditions* (55) *and* (58) *hold; Let* $\epsilon_2^2 \leq \frac{L_1^2}{2(1-\sigma)} \alpha \eta_2 \mathcal{F}$ *and assume* $\mathbf{x}^{-1}$ *such that* $\lambda_{\min}(\nabla^2 f(\hat{\mathbf{x}}^{-1})) \leq -\gamma$ *and further* $\forall i$ *let* $\mathbf{x}_i^0 = \mathbf{x}_i^{-1} + \xi$ *where* $\xi$ *comes from the uniform distribution over the ball of radius* $R = \sqrt{\frac{\mathcal{F}}{L_1}}$. *Consider that the iterates* $\mathbf{x}_i$ *follow the Gradient Tracking Update with stepsize* $\eta_2$ *such that* $\eta_2, \alpha$ *satisfy conditions* (41) - (44) *and further* $\eta_2 \leq \min\{1, \frac{1-\sigma}{L_1}\}$ *and* $\alpha \eta_2 \leq \left( \left( \frac{\delta_2(\sqrt{2}-1)}{8\sqrt{2}\hat{c}\cdot c_{new}} \right)^2 \frac{(1-\sigma)|\lambda_{\min}(H)|^2}{d \log^2\left( \frac{dL_1}{\gamma \delta_2} \right)} \right)$. *Define the sequence of* $\mathbf{w}_i$'s *similarly to* $\mathbf{x}_i$'s *except* $\forall i \quad \mathbf{w}_i^0 = \mathbf{x}_i^0 + \mu \mathcal{R} \mathbf{e}_1$ *with* $\mathbf{e}_1$ *a unit eigenvector corresponding to the minimum eigenvalue of* $\nabla^2 f(\hat{\mathbf{x}}^{-1})$ *and* $\mu \in [\delta_2/2\sqrt{d}, 1]$. *Let* $\mathbf{u}^t = \hat{\mathbf{w}}^t - \hat{\mathbf{x}}^t$ *and consider positive constants* $\hat{c}, c_{new}$ *such that* $\hat{c} \geq 36$ *and* $c_{new} \geq 14 L_1 \sqrt{L_1}$. *Further, consider the definition of the stopping time* $T_{\hat{\mathbf{w}}}$

$$T_{\hat{\mathbf{w}}} = \min\{\inf_t\{t | f(\hat{\mathbf{w}}^t) - f(\hat{\mathbf{w}}^0) \leq -3\mathcal{F}\}, \hat{c}\mathcal{J}\}$$

*If* $\left\| \hat{\mathbf{x}}_t - \hat{\mathbf{x}}^{-1} \right\| \leq 2\hat{c}\mathcal{P}$ *for all* $t < T_{\hat{\mathbf{w}}}$, *then it can be shown that* $T_{\hat{\mathbf{w}}} < \hat{c}\mathcal{J}$.

*Proof.* From the update rule of the iterates we have

$$\hat{\mathbf{x}}^{t+1} + \mathbf{u}^{t+1} = \hat{\mathbf{w}}^{t+1} \tag{127}$$

$$= \hat{\mathbf{w}}^t - \eta_2 \frac{1}{m} \sum_{i=1}^{m} \nabla f_i(\mathbf{w}_i^t) \tag{128}$$

$$= \hat{\mathbf{x}}^t + \mathbf{u}^t - \eta_2 \frac{1}{m} \sum_{i=1}^{m} \nabla f_i(\hat{\mathbf{x}}^t + \mathbf{u}^t) + \frac{\eta_2}{m} \sum_{i=1}^{m} (\nabla f_i(\hat{\mathbf{w}}^t) - \nabla f_i(\mathbf{w}_i^t)) \tag{129}$$

$$= \hat{\mathbf{x}}^t + \mathbf{u}^t - \frac{\eta_2}{m} \sum_{i=1}^{m} \nabla f_i(\hat{\mathbf{x}}^t) - \eta_2 \left( \int_0^1 \nabla^2 f(\hat{\mathbf{x}}^t + \theta \mathbf{u}^t) d\theta \right) \mathbf{u}^t + \frac{\eta_2}{m} \sum_{i=1}^{m} (\nabla f_i(\hat{\mathbf{w}}^t) - \nabla f_i(\mathbf{w}_i^t)) \tag{130}$$

where $\Delta^t = \int_0^1 \frac{1}{m} \sum_{i=1}^m \nabla f(\hat{\mathbf{x}}^t + \theta \mathbf{u}^t) d\theta - H$ and $H = \nabla^2 f(\hat{\mathbf{x}}^{-1})$. Hence, we obtain that

$$\mathbf{u}^{t+1} = \mathbf{u}^t - \eta_2 (H + \Delta^t) \mathbf{u}^t + \frac{\eta_2}{m} \sum_{i=1}^m (\nabla f_i(\hat{\mathbf{w}}^t) - \nabla f_i(\mathbf{w}_i^t)) - \frac{\eta_2}{m} \sum_{i=1}^m (\nabla f_i(\hat{\mathbf{x}}^t) - \nabla f_i(\mathbf{x}_i^t))$$

(131)

which can be simplified as

$$\mathbf{u}^{t+1} = (\mathbf{I} - \eta_2 H) \mathbf{u}^t - \eta_2 \Delta^t \mathbf{u}^t + \frac{\eta_2}{m} \sum_{i=1}^m (\nabla f_i(\hat{\mathbf{w}}^t) - \nabla f_i(\mathbf{w}_i^t)) - \frac{\eta_2}{m} \sum_{i=1}^m (\nabla f_i(\hat{\mathbf{x}}^t) - \nabla f_i(\mathbf{x}_i^t))$$

(132)

Consider the decomposition $\mathbf{u}^t = \mathbf{u}_{par}^t + \mathbf{u}_{per}^t$ where $\mathbf{u}_{par}^t$ is parallel with $\mathbf{e}_1$ and $\mathbf{u}_{per}^t$ is perpendicular to $\mathbf{e}_1$. Then, we can show that

$$\mathbf{e}_1^T (\mathbf{u}_{par}^{t+1} + \mathbf{u}_{per}^{t+1}) = \mathbf{e}_1^T (\mathbf{I} - \eta_2 H) (\mathbf{u}_{par}^t + \mathbf{u}_{per}^t)$$

$$+ \mathbf{e}_1^T \left( -\eta_2 \Delta^t \mathbf{u}^t + \frac{\eta_2}{m} \sum_{i=1}^m (\nabla f_i(\hat{\mathbf{w}}^t) - \nabla f_i(\mathbf{w}_i^t)) - \frac{\eta_2}{m} \sum_{i=1}^m (\nabla f_i(\hat{\mathbf{x}}^t) - \nabla f_i(\mathbf{x}_i^t)) \right)$$

(133)

which can be simplified as

$$\mathbf{e}_1^T \mathbf{u}_{par}^{t+1} = \mathbf{e}_1^T (\mathbf{u}_{par}^t + \mathbf{u}_{per}^t) - \eta_2 (\mathbf{e}_1^T H)(\mathbf{u}_{par}^t + \mathbf{u}_{per}^t)$$

$$+ \mathbf{e}_1^T \left( -\eta_2 \Delta^t \mathbf{u}^t + \frac{\eta_2}{m} \sum_{i=1}^m (\nabla f_i(\hat{\mathbf{w}}^t) - \nabla f_i(\mathbf{w}_i^t)) - \frac{\eta_2}{m} \sum_{i=1}^m (\nabla f_i(\hat{\mathbf{x}}^t) - \nabla f_i(\mathbf{x}_i^t)) \right)$$

$$= \mathbf{e}_1^T \mathbf{u}_{par}^t - \eta_2 (\lambda_{\min}(H) \mathbf{e}_1^T)(\mathbf{u}_{par}^t + \mathbf{u}_{per}^t)$$

$$+ \mathbf{e}_1^T \left( -\eta_2 \Delta^t \mathbf{u}^t + \frac{\eta_2}{m} \sum_{i=1}^m (\nabla f_i(\hat{\mathbf{w}}^t) - \nabla f_i(\mathbf{w}_i^t)) - \frac{\eta_2}{m} \sum_{i=1}^m (\nabla f_i(\hat{\mathbf{x}}^t) - \nabla f_i(\mathbf{x}_i^t)) \right)$$

$$= (1 - \eta_2 \lambda_{\min}(H)) \mathbf{e}_1^T \mathbf{u}_{par}^t$$

$$+ \mathbf{e}_1^T \left( -\eta_2 \Delta^t \mathbf{u}^t + \frac{\eta_2}{m} \sum_{i=1}^m (\nabla f_i(\hat{\mathbf{w}}^t) - \nabla f_i(\mathbf{w}_i^t)) - \frac{\eta_2}{m} \sum_{i=1}^m (\nabla f_i(\hat{\mathbf{x}}^t) - \nabla f_i(\mathbf{x}_i^t)) \right)$$

(134)

If we define $\psi_t$ as the norm of the projection of $\mathbf{u}^t$ onto $\mathbf{e}_1$ and define $\phi_t$ as the norm of the projection on the complementary subspace. Considering the above expression for the sequence $\mathbf{u}^t$, we can show that

$$\psi_{t+1} \geq (1 + \eta_2 |\lambda_{\min}(H)|) \psi_t - \eta_2 \|\Delta^t\| \|\mathbf{u}^t\| - \eta_2 \frac{L_1}{m} \sum_{i=1}^m (\|\hat{\mathbf{w}}^t - \mathbf{w}_i^t\| + \|\hat{\mathbf{x}}^t - \mathbf{x}_i^t\|) \quad (135)$$

Next, consider $\mathbf{e}_2, \ldots, \mathbf{e}_d$ as the remaining eigenvectors of $H$ which create the complementary space of $\mathbf{e}_1$. The projection of any vector $\mathbf{v}$ to this subspace is given by $\sum_{j=2}^d (\mathbf{v}^T \mathbf{e}_j) \mathbf{e}_j$. Therefore, norm of the projection of vector $\mathbf{u}^{t+1}$ onto this subspace is given by

$$\| \sum_{j=2}^d (\mathbf{e}_j^T \mathbf{u}^{t+1}) \mathbf{e}_j \| = \| \sum_{j=2}^d (\mathbf{e}_j^T (\mathbf{u}_{par}^{t+1} + \mathbf{u}_{per}^{t+1})) \mathbf{e}_j \| = \| \sum_{j=2}^d (\mathbf{e}_j^T \mathbf{u}_{per}^{t+1}) \mathbf{e}_j \| = \|\mathbf{u}_{per}^{t+1}\|$$

as expected by the definition. Using the same argument we can show that $\| \sum_{j=2}^d (\mathbf{e}_j^T \mathbf{u}^t) \mathbf{e}_j \| = \|\mathbf{u}_{per}^t\|$. In addition, we can show that

$$\| \sum_{j=2}^d (\mathbf{e}_j^T (I - \eta_2 H) \mathbf{u}^t) \mathbf{e}_j \| = \| \sum_{j=2}^d ((1 - \eta_2 \lambda_j(H)) \mathbf{e}_j^T \mathbf{u}^t) \mathbf{e}_j \|$$

$$\leq (1 - \eta_2 \lambda_{\min}(H)) \| \sum_{j=2}^d (\mathbf{e}_j^T \mathbf{u}^t) \mathbf{e}_j \|$$

$$= (1 - \eta_2 \lambda_{\min}(H)) \|\mathbf{u}_{per}^t\| \quad (136)$$

Then, according to (132) we can write

$$\|\mathbf{u}_{per}^{t+1}\| \tag{137}$$

$$= \|\sum_{j=2}^{d}(\mathbf{e}_j^T \mathbf{u}^{t+1})\mathbf{e}_j\| \tag{138}$$

$$= \|\sum_{j=2}^{d}(\mathbf{e}_j^T \left(\mathbf{I} - \eta_2 H\right) \mathbf{u}^t)\mathbf{e}_j \tag{139}$$

$$+ \sum_{j=2}^{d}(\mathbf{e}_j^T \left(-\eta_2 \Delta^t \mathbf{u}^t + \frac{\eta_2}{m}\sum_{i=1}^{m}(\nabla f_i(\hat{\mathbf{w}}^t) - \nabla f_i(\mathbf{w}_i^t)) - \frac{\eta_2}{m}\sum_{i=1}^{m}(\nabla f_i(\hat{\mathbf{x}}^t) - \nabla f_i(\mathbf{x}_i^t))\right))\mathbf{e}_j\| \tag{140}$$

$$\leq \left\|\sum_{j=2}^{d}(\mathbf{e}_j^T \left(\mathbf{I} - \eta_2 H\right) \mathbf{u}^t)\mathbf{e}_j\right\| \tag{141}$$

$$+ \left\|-\eta_2 \Delta^t \mathbf{u}^t + \frac{\eta_2}{m}\sum_{i=1}^{m}(\nabla f_i(\hat{\mathbf{w}}^t) - \nabla f_i(\mathbf{w}_i^t)) - \frac{\eta_2}{m}\sum_{i=1}^{m}(\nabla f_i(\hat{\mathbf{x}}^t) - \nabla f_i(\mathbf{x}_i^t))\right\| \tag{142}$$

$$\leq (1 - \eta_2 \lambda_{\min}(H))\|\mathbf{u}_{per}^t\|$$

$$+ \left\|-\eta_2 \Delta^t \mathbf{u}^t + \frac{\eta_2}{m}\sum_{i=1}^{m}(\nabla f_i(\hat{\mathbf{w}}^t) - \nabla f_i(\mathbf{w}_i^t)) - \frac{\eta_2}{m}\sum_{i=1}^{m}(\nabla f_i(\hat{\mathbf{x}}^t) - \nabla f_i(\mathbf{x}_i^t))\right\| \tag{143}$$

Note that $\phi_t$ is the norm of the projection onto the complementary subspace which is equal to $\|\mathbf{u}_{per}^t\|$. Further, since $\lambda_{\min}(H) \leq -\gamma$ then we have $|\lambda_{\min}(H)| = -\lambda_{\min}(H)$. Considering these points we can write

$$\phi_{t+1} \leq (1 + \eta_2|\lambda_{\min}(H)|)\phi_t + \eta_2 \|\Delta^t\| \|\mathbf{u}^t\| + \eta_2\frac{L_1}{m}\sum_{i=1}^{m}(\|\hat{\mathbf{w}}^t - \mathbf{w}_i^t\| + \|\hat{\mathbf{x}}^t - \mathbf{x}_i^t\|) \tag{144}$$

To bound the norm of $\Delta^t$ first note that for any $t < T_{\hat{\mathbf{w}}}$

$$\|\hat{\mathbf{x}}^t - \hat{\mathbf{x}}^{-1}\| \leq 2\hat{c}\mathcal{P} \tag{145}$$

and further since $\hat{\mathbf{w}}^0$ satisfy the condition of **Lemma 13** we have

$$\|\hat{\mathbf{w}}^0 - \hat{\mathbf{x}}^{-1}\| = \|\hat{\mathbf{x}}^0 - \hat{\mathbf{x}}^{-1}\| + \|\mathbf{u}^0\| \leq \mathcal{R} + \mu\mathcal{R} \leq 2\mathcal{R} \Rightarrow \|\hat{\mathbf{w}}^t - \hat{\mathbf{x}}^{-1}\| \leq 2\hat{c}\mathcal{P} \tag{146}$$

It follows that for any $t < T_{\hat{\mathbf{w}}}$

$$\|\mathbf{u}^t\| = \|\hat{\mathbf{x}}^t - \hat{\mathbf{w}}^t\| \leq \|\hat{\mathbf{x}}^t - \hat{\mathbf{x}}^{-1}\| + \|\hat{\mathbf{w}}^t - \hat{\mathbf{x}}^{-1}\| \leq 4\hat{c}\mathcal{P}. \tag{147}$$

Now we can show that

$$\|\Delta^t\| \leq \|\nabla^2 f(\hat{\mathbf{x}}^t + \mathbf{u}^t) - \nabla^2 f(\hat{\mathbf{x}}^{-1})\| \leq L_2(\|\hat{\mathbf{x}}^t - \hat{\mathbf{x}}^{-1}\| + \|\mathbf{u}^t\|) \leq 6L_2\hat{c}\mathcal{P} \tag{148}$$

Using this upper bound we can show that

$$\psi_{t+1} \geq (1 + \eta_2|\lambda_{\min}(H)|)\psi_t - \zeta\sqrt{\psi_t^2 + \phi_t^2} - \eta_2\frac{L_1}{m}\sum_{i=1}^{m}(\|\hat{\mathbf{w}}^t - \mathbf{w}_i^t\| + \|\hat{\mathbf{x}}^t - \mathbf{x}_i^t\|) \tag{149}$$

and

$$\phi_{t+1} \leq (1 + \eta_2|\lambda_{\min}(H)|)\phi_t + \zeta\sqrt{\psi_t^2 + \phi_t^2} + \eta_2\frac{L_1}{m}\sum_{i=1}^{m}(\|\hat{\mathbf{w}}^t - \mathbf{w}_i^t\| + \|\hat{\mathbf{x}}^t - \mathbf{x}_i^t\|) \tag{150}$$

where

$$\zeta = \eta_2 L_2 6\hat{c}\mathcal{P} = \eta_2 L_2 6\hat{c}\sqrt{\frac{\mathcal{F}\log(d\kappa/\delta_2)}{|\lambda_{\min}(H)|}}. \tag{151}$$

Using induction we are going to prove the following two statements $\forall t < T_{cap}$.

$$\eta_2 \frac{L_1}{m} \sum_{i=1}^{m} (\|\hat{\mathbf{w}}^t - \mathbf{w}_i^t\| + \|\hat{\mathbf{x}}^t - \mathbf{x}_i^t\|) \leq \zeta \psi_t \tag{152}$$

$$\phi_t \leq 4t\zeta\psi_t \tag{153}$$

Recall the values of the following variables

$$\mathcal{F} = \frac{|\lambda_{\min}(H)|^3}{\log^3(d\kappa/\delta_2)} \frac{(\sqrt{2}-1)^2}{(24\sqrt{2}L_2\hat{c}^2)^2} \tag{154}$$

$$\mathcal{P} = \frac{|\lambda_{\min}(H)|}{\log(d\kappa/\delta_2)} \frac{\sqrt{2}-1}{(24\sqrt{2}L_2\hat{c}^2)} \tag{155}$$

For the base of the induction, note that since $\mathbf{u}^0 = \mu \mathcal{R} \mathbf{e}_1$ then we can conclude that $\psi_0 = \mu \mathcal{R}$ and $\phi_0 = 0$. Then, we have

$$\zeta\psi_0 = \eta_2 L_2 6\hat{c} \sqrt{\frac{\log(d\kappa/\delta_2)\mathcal{F}}{|\lambda_{\min}(H)|}} \psi_0 \tag{156}$$

$$= \eta_2 L_2 6\hat{c} \sqrt{\frac{\log(d\kappa/\delta_2)\mathcal{F}}{|\lambda_{\min}(H)|}} \mathcal{R}\mu \tag{157}$$

$$\geq \eta_2 L_2 6\hat{c} \sqrt{\frac{\log(d\kappa/\delta_2)\mathcal{F}}{|\lambda_{\min}(H)|}} \mathcal{R} \frac{\delta_2}{2\sqrt{d}} \tag{158}$$

$$= 3\hat{c}L_2 \frac{\delta_2}{\sqrt{d}} \eta_2 \sqrt{\frac{\log(d\kappa/\delta_2)\mathcal{F}}{|\lambda_{\min}(H)|}} \mathcal{R} \tag{159}$$

where the inequality follows from the fact that $\mu \geq \frac{\delta_2}{2\sqrt{d}}$. Using this inequality and the result of Lemma 14 we can show that

$$\zeta\psi_0 - \eta_2 \frac{L_1}{m} \sum_{i=1}^{m} (\|\hat{\mathbf{w}}^t - \mathbf{w}_i^t\| + \|\hat{\mathbf{x}}^t - \mathbf{x}_i^t\|)$$

$$\geq 3\hat{c}L_2 \frac{\delta_2}{\sqrt{d}} \eta_2 \sqrt{\frac{\log(d\kappa/\delta_2)\mathcal{F}}{|\lambda_{\min}(H)|}} \mathcal{R} - c_{new} \sqrt{\frac{\alpha\eta_2}{1-\sigma}} \eta_2 \mathcal{R} \tag{160}$$

$$= \eta_2 \mathcal{R} \left( 3\hat{c}L_2 \frac{\delta_2}{\sqrt{d}} \sqrt{\frac{\mathcal{F}\log(d\kappa/\delta_2)}{|\lambda_{\min}(H)|}} - c_{new} \sqrt{\frac{\alpha\eta_2}{1-\sigma}} \right) \tag{161}$$

$$= \eta_2 \mathcal{R} \left( 3\hat{c}L_2 \frac{\delta_2(\sqrt{2}-1)}{24\sqrt{2}L_2\hat{c}^2} \frac{|\lambda_{\min}(H)|}{\sqrt{d}\log(d\kappa/\delta_2)} - c_{new} \sqrt{\frac{\alpha\eta_2}{1-\sigma}} \right) \tag{162}$$

$$= \eta_2 \mathcal{R} \left( \frac{\delta_2(\sqrt{2}-1)}{8\sqrt{2}\hat{c}} \frac{|\lambda_{\min}(H)|}{\sqrt{d}\log(d\kappa/\delta_2)} - c_{new} \sqrt{\frac{\alpha\eta_2}{1-\sigma}} \right) \tag{163}$$

$$\geq 0 \tag{164}$$

where the last inequality holds for $\alpha\eta_2 \leq \left( \left( \frac{\delta_2(\sqrt{2}-1)}{8\sqrt{2}\hat{c}\cdot c_{new}} \right)^2 \frac{(1-\sigma)|\lambda_{\min}(H)|^2}{d\log^2\left(\frac{dL_1}{\gamma\delta_2}\right)} \right)$. Hence, there are $\alpha, \eta_2$ properly chosen such that $a\eta_2 = \tilde{\mathcal{O}}\left(\frac{(1-\sigma)\gamma^2}{d}\right)$ that the base of induction for (152) holds. Further, since $\phi_0 = 0$ the second condition (153) is also satisfied for $t = 0$ and the base of the induction is complete.

Now let's assume that the conditions in (152) and (153) hold for time $t$. Our goal is to show that these conditions also hold for time $t+1$.

From the inductive hypothesis we have $\phi_t \leq 4t\zeta\psi_t$ and also $\zeta\sqrt{\psi_t^2 + \phi_t^2} \geq \zeta\psi_t \geq \eta_2 \frac{L_1}{m} \sum_{i=1}^{m} (\|\hat{\mathbf{w}}^t - \mathbf{w}_i^t\| + \|\hat{\mathbf{x}}^t - \mathbf{x}_i^t\|)$. Thus

$$\psi_{t+1} \geq (1 + \eta_2|\lambda_{\min}(H)|)\psi_t - \zeta\sqrt{\psi_t^2 + \phi_t^2} - \eta_2\frac{L_1}{m}\sum_{i=1}^{m}(\|\hat{\mathbf{w}}^t - \mathbf{w}_i^t\| + \|\hat{\mathbf{x}}^t - \mathbf{x}_i^t\|)$$

$$\geq (1 + \eta_2|\lambda_{\min}(H)|)\psi_t - 2\zeta\sqrt{\psi_t^2 + \phi_t^2} \tag{165}$$

And similarly we can show that

$$\phi_{t+1} \leq (1 + \eta_2|\lambda_{\min}(H)|)\phi_t + 2\zeta\sqrt{\psi_t^2 + \phi_t^2} \tag{166}$$

By multiplying both sides of (165) by $4(t+1)\zeta$ we obtain that

$$4(t+1)\zeta\psi_{t+1} \geq 4(t+1)\zeta\left((1 + \eta_2|\lambda_{\min}(H)|)\psi_t - 2\zeta\sqrt{\psi_t^2 + \phi_t^2}\right) \tag{167}$$

And if we replace $\phi_t$ in the right hand side (166) by its upper bound $4t\zeta\psi_t$ (given by the induction hypothesis), then we obtain

$$\phi_{t+1} \leq (1 + \eta_2|\lambda_{\min}(H)|)4t\zeta\psi_t + 2\zeta\sqrt{\psi_t^2 + \phi_t^2} \tag{168}$$

Considering the inequalities in (167) and (168), to prove that $4(t+1)\zeta\psi_{t+1} \geq \phi_{t+1}$ it suffices to show

$$4(t+1)\zeta\left((1 + \eta_2|\lambda_{\min}(H)|)\psi_t - 2\zeta\sqrt{\psi_t^2 + \phi_t^2}\right) \geq 4t\zeta(1 + \eta_2|\lambda_{\min}(H)|)\psi_t + 2\zeta\sqrt{\psi_t^2 + \phi_t^2} \tag{169}$$

which is equivalent to

$$4(t+1)\left((1 + \eta_2|\lambda_{\min}(H)|)\psi_t - 2\zeta\sqrt{\psi_t^2 + \phi_t^2}\right) \geq 4t(1 + \eta_2|\lambda_{\min}(H)|)\psi_t + 2\sqrt{\psi_t^2 + \phi_t^2} \tag{170}$$

Expanding the left hand side leads to

$$4t(1 + \eta_2|\lambda_{\min}(H)|)\psi_t + 4(1 + \eta_2|\lambda_{\min}(H)|)\psi_t - 8t\zeta\sqrt{\psi_t^2 + \phi_t^2} - 8\zeta\sqrt{\psi_t^2 + \phi_t^2}$$

$$\geq 4t(1 + \eta_2|\lambda_{\min}(H)|)\psi_t + 2\sqrt{\psi_t^2 + \phi_t^2} \tag{171}$$

Hence, the conditions in (169), (170), and (171) are equivalent. By regrouping the terms in (171) and dividing both sides by 2 we obtain the following condition

$$2(1 + \eta_2|\lambda_{\min}(H)|)\psi_t \geq (1 + 4(t+1)\zeta)\sqrt{\psi_t^2 + \phi_t^2} \tag{172}$$

Indeed, the condition in (172) holds if and only if (171) holds.

Finally to prove the last inequality in (172) notice that

$$\begin{aligned}
4(t+1)\zeta &\leq 4\zeta T_{cap} \\
&\leq 4\eta_2 6 L_2 \hat{c}\mathcal{P}\hat{c}\mathcal{J} \\
&\leq 24\eta_2 L_2 \hat{c}^2 \mathcal{P}\mathcal{J} \\
&\leq 24\eta_2 L_2 \hat{c}^2 \frac{|\lambda_{\min}(H)|}{\log(d\kappa/\delta_2)} \frac{\sqrt{2}-1}{(24\sqrt{2}L_2\hat{c}^2)} \frac{\log(d\kappa/\delta_2)}{\eta_2|\lambda_{\min}(H)|} \\
&\leq \sqrt{2} - 1 \tag{173}
\end{aligned}$$

Thus, we can show that

$$(1 + 4(t+1)\zeta)\sqrt{\psi_t^2 + \phi_t^2} \le \sqrt{2}\sqrt{\psi_t^2 + \psi_t^2} \le \sqrt{2}\sqrt{2\psi_t^2} \le 2(1 + \eta_2|\lambda_{\min}(H)|)\psi_t \qquad (174)$$

Hence, the condition in (172) holds, and as a result the condition in (169) holds. As we mentioned, (169) together with (167) and (168) implies that

$$\phi_{t+1} \le 4t\zeta\psi_{t+1}. \qquad (175)$$

Hence, the induction step for (152) is complete.

Next we show that if (153) holds for $t$ it also holds for $t + 1$. To do so, note that by considering the fact that $4t\zeta \le \sqrt{2} - 1$, from (173) and the result in (175), we can show that

$$\phi_{t+1} \le \psi_{t+1} \qquad (176)$$

Using the result in (176) as well as the inequality in (165) we can show that

$$
\begin{aligned}
\psi_{t+1} &\ge (1 + \eta_2|\lambda_{\min}(H)|)\psi_t - 2\sqrt{2}\zeta\psi_t \\
&\ge \psi_t + \eta_2|\lambda_{\min}(H)|\psi_t - 12\sqrt{2}\eta_2 L_2 \hat{c}\sqrt{\frac{\mathcal{F}\log(d\kappa/\delta_2)}{|\lambda_{\min}(H)|}}\psi_t \\
&\ge \psi_t + \eta_2|\lambda_{\min}(H)|\psi_t - 12\sqrt{2}\eta_2 L_2 \hat{c}\frac{|\lambda_{\min}(H)|}{\log(d\kappa/\delta_2)} \cdot \frac{\sqrt{2}-1}{24\sqrt{2}L_2\hat{c}^2}\psi_t \\
&\ge \psi_t(1 + \frac{\eta_2|\lambda_{\min}(H)|}{2})
\end{aligned}
\qquad (177)
$$

Since we showed that $\psi_{t+1} \ge \psi_t$ and thus $\zeta\psi_{t+1} \ge \zeta\psi_t$ it is straight forward to prove the second condition of the inductive step (153) simply by using the same bound for $\eta_2\frac{L_1}{m}\sum_{i=1}^{m}(\|\hat{\mathbf{w}}^t - \mathbf{w}_i^t\| + \|\hat{\mathbf{x}}^{t+1} - \mathbf{x}_i^{t+1}\|)$ from Lemma 14. To be more precise, based on the result of Lemma 14 we know that for any $t \le T_{cap}$

$$\eta_2\frac{L_1}{m}\sum_{i=1}^{m}(\|\hat{\mathbf{w}}^t - \mathbf{w}_i^t\| + \|\hat{\mathbf{x}}^t - \mathbf{x}_i^t\|) \le 3\hat{c}L_2\frac{\delta_2}{\sqrt{d}}\eta_2\sqrt{\frac{\log(d\kappa/\delta_2)\mathcal{F}}{|\lambda_{\min}(H)|}}\mathcal{R} \le \zeta\psi_0 \qquad (178)$$

Using this result and the fact that $\psi_t$ is increasing we can show that

$$\eta_2\frac{L_1}{m}\sum_{i=1}^{m}(\|\hat{\mathbf{w}}^{t+1} - \mathbf{w}_i^{t+1}\| + \|\hat{\mathbf{x}}^{t+1} - \mathbf{x}_i^{t+1}\|) \le 3\hat{c}L_2\frac{\delta_2}{\sqrt{d}}\eta_2\sqrt{\frac{\log(d\kappa/\delta_2)\mathcal{F}}{|\lambda_{\min}(H)|}}\mathcal{R} \le \zeta\psi_{t+1} \quad (179)$$

and the induction is complete.

Next, using the result of induction in (152) and (153) we show that $T_{\hat{\mathbf{w}}} < \hat{c}\mathcal{J}$. To do so, note that for all $t < T_{\hat{\mathbf{w}}}$ we have

$$4\hat{c}\mathcal{P} \geq \left\| \mathbf{u}^t \right\|$$
$$\geq \psi_t$$
$$\geq (1 + \frac{\eta_2|\lambda_{\min}(H)|}{2})^t \psi_0$$
$$= (1 + \frac{\eta_2|\lambda_{\min}(H)|}{2})^t \mu\mathcal{R}$$
$$\geq (1 + \frac{\eta_2|\lambda_{\min}(H)|}{2})^t \frac{\delta_2}{2\sqrt{d}}\sqrt{\frac{\mathcal{F}}{L_1}} \tag{180}$$
$$\geq (1 + \frac{\eta_2|\lambda_{\min}(H)|}{2})^t \frac{\delta_2}{2\sqrt{d}}\frac{|\lambda_{\min}|}{\log(d\kappa/\delta_2)}\frac{\sqrt{2}-1}{24\sqrt{2}L_2\hat{c}^2}\sqrt{\frac{|\lambda_{\min}|}{L_1\log(d\kappa/\delta_2)}} \tag{181}$$
$$\geq (1 + \frac{\eta_2|\lambda_{\min}(H)|}{2})^t \frac{\delta_2\mathcal{P}}{2\sqrt{d}}\sqrt{\frac{|\lambda_{\min}|}{L_1\log(d\kappa/\delta_2)}} \tag{182}$$
$$\geq (1 + \frac{\eta_2|\lambda_{\min}(H)|}{2})^t \frac{\delta_2\mathcal{P}}{2\sqrt{d}}\frac{|\lambda_{\min}|}{L_1\log(d\kappa/\delta_2)} \tag{183}$$
$$\geq (1 + \eta_2|\lambda_{\min}(H)|/2)^t \frac{\delta_2\mathcal{P}}{2\sqrt{d}\kappa\log(d\kappa/\delta_2)} \tag{184}$$

where the first inequality follows from (147), the second inequality holds since $\psi_t$ is the norm of projection of $\mathbf{u}^t$ onto a subspace, the third inequality holds because of the result in (177), and the second to last inequality holds since $\log(d\kappa/\delta_2) \geq 1$ and $\frac{|\lambda_{\min}|}{L_1} \leq 1$. Hence, we have for $t < T_{\hat{\mathbf{w}}}$

$$\frac{8\hat{c}\sqrt{d}\kappa\log(d\kappa/\delta_2)}{\delta_2} \geq (1 + \eta_2|\lambda_{\min}(H)|/2)^t \tag{185}$$

Therefore, this condition should also hold for $t = T_{\hat{\mathbf{w}}} - 1$ and therefore we have

$$T_{\hat{\mathbf{w}}} - 1 \leq \frac{\log(8\frac{\kappa\sqrt{d}}{\delta_2}\hat{c}\log(d\kappa/\delta_2))}{\log(1 + \frac{|\lambda_{\min}(H)|\eta_2}{2})} \tag{186}$$

$$\Rightarrow T_{\hat{\mathbf{w}}} < \frac{\log(8\frac{\kappa\sqrt{d}}{\delta_2}\hat{c}\log(d\kappa/\delta_2))}{\log(1 + \frac{|\lambda_{\min}(H)|\eta_2}{2})} + 1 \leq \frac{5}{2}\frac{\log(8\frac{\kappa\sqrt{d}}{\delta_2}\hat{c}\log(d\kappa/\delta_2))}{1 + \frac{|\lambda_{\min}(H)|\eta_2}{2}} + 1 \leq \frac{5}{2}(2 + \log(8\hat{c}))\mathcal{J} + 1 < \hat{c}\mathcal{J} \tag{187}$$

where the last inequality uses the facts that $\delta_2 \in (0, \frac{d\kappa}{e}]$ and $\log(d\kappa/\delta_2) \geq 1$ and $\hat{c}$ such that $\frac{5}{2}(2 + \log(8\hat{c})) \leq \hat{c}$. □

Now we are going to use Lemma 15 to show substantial function decrease in a small number of iterations after the noise injection with high probability. Specifically, we are going to show that $f(\hat{\mathbf{x}}^T) - f(\hat{\mathbf{x}}^{-1}) \leq -\mathcal{F}$ for some $T < \hat{c}\mathcal{J}$ which will be used subsequently consequently to show $H(\mathbf{x}^{\hat{c}\mathcal{J}}, \mathbf{y}^{\hat{c}\mathcal{J}}) - H(\mathbf{x}^{-1}, \mathbf{y}^{-1}) \leq -\mathcal{F}$.

**Lemma 16.** *Assume the major condition (86) and conditions (54), (55) and (58) hold; let $\epsilon_1^2 \leq \mathcal{F}\frac{L_1}{2+2L_1^2}$ and $\epsilon_2^2 \leq \min\{\frac{L_1^2}{2(1-\sigma)}\alpha\eta_2\mathcal{F}, \mathcal{F}\frac{L_1^2}{2+2L_1^2}\}$. Assume $\mathbf{x}^{-1}$ such that $\lambda_{\min}(\nabla^2 f(\hat{\mathbf{x}}^{-1})) \leq -\gamma$ and further $\forall i$ let $\mathbf{x}_i^0 = \mathbf{x}_i^{-1} + \xi$ where $\xi$ comes from the uniform distribution over the ball of radius $R = \sqrt{\frac{\mathcal{F}}{L_1}}$. Consider that the iterates $\mathbf{x}_i$ follow the Gradient Tracking Update with stepsize $\eta_2$ such that $\eta_2, \alpha$ satisfy conditions (41) - (44) and further $\eta_2 \leq \min\{1, \frac{1-\sigma}{L_1}\}$ and $\alpha\eta_2 \leq \left( \left(\frac{\delta_2(\sqrt{2}-1)}{8\sqrt{2}\hat{c}\cdot c_{new}}\right)^2 \frac{(1-\sigma)|\lambda_{\min}(H)|^2}{d\log^2\left(\frac{dL_1}{\gamma\delta_2}\right)} \right)$. Then with probability at least $1-\delta_2$ we have the following for some $T < \hat{c}\mathcal{J}$*

$$f(\hat{\mathbf{x}}^T) - f(\hat{\mathbf{x}}^0) \leq -3\mathcal{F} \tag{188}$$

*which implies*

$$f(\hat{\mathbf{x}}^T) - f(\hat{\mathbf{x}}^{-1}) \leq -\mathcal{F} \tag{189}$$

*Proof.* In Lemma 10 by adding perturbation we proved that the function value increases at most by $\frac{3}{2}\mathcal{F}$. Thus we have

$$f(\hat{\mathbf{x}}^0) - f(\hat{\mathbf{x}}^{-1}) \leq \frac{3}{2}\mathcal{F} \tag{190}$$

We know that $\hat{\mathbf{x}}^0$ comes from the uniform distribution over $\mathcal{B}_{\hat{\mathbf{x}}^{-1}}(\mathcal{R})$. Let us denote with $\mathcal{X}_{stuck} \subset \mathcal{B}_{\hat{\mathbf{x}}^{-1}}(\mathcal{R})$ the set of bad starting points so that if $\hat{\mathbf{x}}^0 \in \mathcal{X}_{stuck}$, then the iterates are not going to make substantial progress after at most $\hat{c}\mathcal{J}$ steps i.e. $f(\hat{\mathbf{x}}^T) - f(\hat{\mathbf{x}}^0) > -3\mathcal{F}, \quad \forall T < \hat{c}\mathcal{J}$. On the contrary when $\hat{\mathbf{x}}^0 \in (\mathcal{B}_{\hat{\mathbf{x}}^{-1}}(R) - \mathcal{X}_{stuck})$ there exists a $T$ such that $f(\hat{\mathbf{x}}^T) - f(\hat{\mathbf{x}}^0) < -3\mathcal{F}$.

By the Lemma 15 we know that when $\hat{\mathbf{x}}^0 \in \mathcal{X}_{stuck}$ it is guaranteed that $(\hat{\mathbf{x}}^0 \pm \mu\mathcal{R}\mathbf{e}_1) \notin \mathcal{X}_{stuck}$ where $\mu \in \left[\frac{\delta_2}{2\sqrt{d}}, 1\right]$. Denote with $\mathcal{I}_{\mathcal{X}_{stuck}}(\cdot)$ the indicator function of being inside set $\mathcal{X}_{stuck}$ and vector $\mathbf{v} = \left(v^{(1)}, \mathbf{v}^{(-1)}\right)$, where $v^1$ is the component along the direction of $\mathbf{e}_1$ and $\mathbf{v}^{(-1)}$ the remaining vector. We are going to derive an upper bound on the volume of $\mathcal{X}_{stuck}$.

$$Vol(\mathcal{X}_{stuck}) = \int_{\mathcal{B}_{\hat{\mathbf{x}}^{-1}}^d(\mathcal{R})} d\mathbf{x} \cdot \mathcal{I}_{\mathcal{X}_{stuck}}(\mathbf{x}) \tag{191}$$

$$\leq \int_{\mathcal{B}_{\hat{\mathbf{x}}^{-1}}^{d-1}(\mathcal{R})} d\mathbf{x}^{(-1)} \cdot 2\frac{\delta_2}{2\sqrt{d}}\mathcal{R} \tag{192}$$

$$= Vol(\mathcal{B}_{\hat{\mathbf{x}}^{-1}}^{d-1}(\mathcal{R})) \times \frac{\delta_2 \mathcal{R}}{\sqrt{d}} \tag{193}$$

Then we immediately have the ratio:

$$\frac{Vol(\mathcal{X}_{stuck})}{Vol\left(\mathcal{B}_{\hat{\mathbf{x}}^{-1}}^d(\mathcal{R})\right)} \leq \frac{\frac{\delta_2 \mathcal{R}}{\sqrt{d}} Vol(\mathcal{B}_{\hat{\mathbf{x}}^{-1}}^{d-1}(\mathcal{R}))}{Vol\left(\mathcal{B}_{\hat{\mathbf{x}}^{-1}}^d(\mathcal{R})\right)} = \frac{\delta_2}{\sqrt{\pi d}}\frac{\Gamma(\frac{d}{2}+1)}{\Gamma(\frac{d}{2}+\frac{1}{2})} \leq \frac{\delta_2}{\sqrt{\pi d}}\sqrt{\frac{d}{2}+\frac{1}{2}} \leq \delta_2 \tag{194}$$

The second to last inequality is by the property of Gamma function that $\frac{\Gamma(x+1)}{\Gamma(x+\frac{1}{2})} < \sqrt{x+\frac{1}{2}}$ as long as $x \geq 0$. Therefore, with at least probability $1 - \delta_2$, $\hat{\mathbf{x}}^0 \notin \mathcal{X}_{stuck}$ i.e.

$$f(\hat{\mathbf{x}}^T) - f(\hat{\mathbf{x}}^0) \leq -3\mathcal{F} \tag{195}$$

In this case we have:

$$f(\hat{\mathbf{x}}^T) - f(\hat{\mathbf{x}}^{-1}) = f(\hat{\mathbf{x}}^T) - f(\hat{\mathbf{x}}^0) + f(\hat{\mathbf{x}}^0) - f(\hat{\mathbf{x}}^{-1}) \tag{196}$$

$$\leq -3\mathcal{F} + \frac{3}{2}\mathcal{F} \tag{197}$$

$$\leq -\mathcal{F} \tag{198}$$

$\square$

**Lemma 17.** *Assume the major condition (86) and conditions (54), (55) and (58) hold; let $\epsilon_1^2 \leq \mathcal{F}\frac{L_1}{2+2L_1^2}$ and $\epsilon_2^2 \leq \min\{\frac{L_1^2}{2(1-\sigma)}\alpha\eta_2\mathcal{F}, \mathcal{F}\frac{L_1^2}{2+2L_1^2}\}$. Assume $\mathbf{x}^{-1}$ such that $\lambda_{\min}(\nabla^2 f(\hat{\mathbf{x}}^{-1})) \leq -\gamma$ and further $\forall i$ let $\mathbf{x}_i^0 = \mathbf{x}_i^{-1} + \xi$ where $\xi$ comes from the uniform distribution over the ball of radius $R = \sqrt{\frac{\mathcal{F}}{L_1}}$. Consider that the iterates $\mathbf{x}_i$ follow the Gradient Tracking Update with stepsize $\eta_2$ such that $\eta_2, \alpha$ satisfy conditions (41) - (44) and further $\eta_2 \leq \min\{1, \frac{1-\sigma}{L_1}\}$ and $\alpha\eta_2 \leq \min\left\{\left(\frac{\delta_2(\sqrt{2}-1)}{8\sqrt{2}\hat{c}\cdot c_{new}}\right)^2\frac{(1-\sigma)|\lambda_{\min}(H)|^2}{d\log^2\left(\frac{dL_1}{\gamma\delta_2}\right)}, \frac{1-\sigma}{192L_1^2}\right\}$. Then with probability at least $1 - \delta_2$ we have the following*

$$H(\underline{\mathbf{x}}^{T_{cap}}, \underline{\mathbf{y}}^{T_{cap}}) - H(\underline{\mathbf{x}}^{-1}, \underline{\mathbf{y}}^{-1}) \leq -\mathcal{F} \tag{199}$$

*Proof.* Recall that

$$H(\underline{\mathbf{x}}^r, \underline{\mathbf{y}}^r) := \frac{1}{m}\sum_{i=1}^m f_i(\hat{\mathbf{x}}^r) + \frac{1}{m}\|\underline{\mathbf{x}}^r - \hat{\underline{\mathbf{x}}}^r\|^2 + \frac{\alpha}{m}\|\underline{\mathbf{y}}^r - \hat{\underline{\mathbf{y}}}^r\|^2 \tag{200}$$

$$H(\underline{\mathbf{x}}^0, \underline{\mathbf{y}}^0) - H(\underline{\mathbf{x}}^{-1}, \underline{\mathbf{y}}^{-1})$$
$$= f(\hat{\mathbf{x}}^0) - f(\hat{\mathbf{x}}^{-1}) + \frac{1}{m}\|\underline{\mathbf{x}}^0 - \hat{\underline{\mathbf{x}}}^0\|^2 - \frac{1}{m}\|\underline{\mathbf{x}}^{-1} - \hat{\underline{\mathbf{x}}}^{-1}\|^2 + \frac{\alpha}{m}\|\underline{\mathbf{y}}^0 - \hat{\underline{\mathbf{y}}}^0\|^2 - \frac{\alpha}{m}\|\underline{\mathbf{y}}^{-1} - \hat{\underline{\mathbf{y}}}^{-1}\|^2$$
$$\leq f(\hat{\mathbf{x}}^0) - f(\hat{\mathbf{x}}^{-1}) + \frac{\alpha}{m}\|\underline{\mathbf{y}}^0 - \hat{\underline{\mathbf{y}}}^0\|^2$$
$$\leq \frac{3}{2}\mathcal{F} + \alpha\eta_2 \frac{L_1^2}{2(1-\sigma)}\mathcal{F} \tag{201}$$
$$\leq \frac{3}{2}\mathcal{F} + \frac{1}{4}\mathcal{F} \tag{202}$$
$$\leq \frac{7}{4}\mathcal{F}, \tag{203}$$

where the first inequality comes from the fact that the same noise is injected to all local iterates and thus

$$\frac{1}{m}\|\underline{\mathbf{x}}^0 - \hat{\underline{\mathbf{x}}}^0\|^2 - \frac{1}{m}\|\underline{\mathbf{x}}^{-1} - \hat{\underline{\mathbf{x}}}^{-1}\|^2 = 0 \tag{204}$$

for the second inequality we use that $f(\hat{\mathbf{x}}^0) - f(\hat{\mathbf{x}}^{-1}) \leq \frac{3}{2}\mathcal{F}$ due to Lemma 10 and the bound from (58). □

Further, we have for the same $T < \hat{c}\mathcal{J}$ from Lemma 16

$$H(\underline{\mathbf{x}}^T, \underline{\mathbf{y}}^T) - H(\underline{\mathbf{x}}^0, \underline{\mathbf{y}}^0) = \frac{1}{m}\sum_{i=1}^m f_i(\hat{\mathbf{x}}^T) - \frac{1}{m}\sum_{i=1}^m f_i(\hat{\mathbf{x}}^0) + \frac{1}{m}\|\underline{\mathbf{x}}^T - \hat{\underline{\mathbf{x}}}^T\|^2 - \frac{1}{m}\|\underline{\mathbf{x}}^0 - \hat{\underline{\mathbf{x}}}^0\|^2$$
$$+ \frac{\alpha}{m}\|\underline{\mathbf{y}}^T - \hat{\underline{\mathbf{y}}}^T\|^2 - \frac{\alpha}{m}\|\underline{\mathbf{y}}^0 - \hat{\underline{\mathbf{y}}}^0\|^2$$
$$= f(\hat{\mathbf{x}}^T) - f(\hat{\mathbf{x}}^0) + P(\underline{\mathbf{x}}^T) - P(\underline{\mathbf{x}}^0)$$
$$\leq f(\hat{\mathbf{x}}^T) - f(\hat{\mathbf{x}}^0) + \alpha\eta_2^2 4L_1^2(1 + \frac{1}{\beta_2})\sum_{t=0}^T \|\hat{\mathbf{y}}^t\|^2$$
$$\leq f(\hat{\mathbf{x}}^T) - f(\hat{\mathbf{x}}^0) + \alpha\eta_2^2 4L_1^2 \frac{1}{1-\sigma}12\frac{\mathcal{F}}{\eta_2}$$
$$= f(\hat{\mathbf{x}}^T) - f(\hat{\mathbf{x}}^0) + \frac{1}{1-\sigma}48L_1^2\alpha\eta_2\mathcal{F}$$
$$\leq -3\mathcal{F} + \frac{1}{1-\sigma}48L_1^2\alpha\eta_2\mathcal{F}$$
$$\leq -\frac{11}{4}\mathcal{F} \tag{205}$$

where the first inequality comes from Lemma 5, the second by condition (86) and the third by Lemma 16. Finally we get

$$H(\underline{\mathbf{x}}^T, \underline{\mathbf{y}}^T) - H(\underline{\mathbf{x}}^{-1}, \underline{\mathbf{y}}^{-1}) = H(\underline{\mathbf{x}}^T, \underline{\mathbf{y}}^T) - H(\underline{\mathbf{x}}^0, \underline{\mathbf{y}}^0) + H(\underline{\mathbf{x}}^0, \underline{\mathbf{y}}^0) - H(\underline{\mathbf{x}}^{-1}, \underline{\mathbf{y}}^{-1}) \leq \frac{7}{4}\mathcal{F} - \frac{11}{4}\mathcal{F} \leq -\mathcal{F} \tag{206}$$

Since the potential function is non increasing $H(\underline{\mathbf{x}}^{T_{cap}}, \underline{\mathbf{y}}^{T_{cap}}) \leq H(\underline{\mathbf{x}}^T, \underline{\mathbf{y}}^T)$ and the result follows.

Combining Lemma 12 and Lemma 17 we derive the following corollary stating that during the second phase, the Gradient Tracking sequence is going to escape with high probability from an initial point of sufficient negative curvature.

**Corollary 5.** *Assume conditions (54), (55) and (58) hold; let $\epsilon_1^2 \leq \mathcal{F}\frac{L_1}{2+2L_1^2}$ and $\epsilon_2^2 \leq \min\{\frac{L_1^2}{2(1-\sigma)}\alpha\eta_2\mathcal{F}, \mathcal{F}\frac{L_1}{2+2L_1^2}\}$. Assume $\mathbf{x}^{-1}$ such that $\lambda_{\min}(\nabla^2 f(\hat{\mathbf{x}}^{-1})) \leq -\gamma$ and further $\forall i$ let $\mathbf{x}_i^0 = \mathbf{x}_i^{-1} + \xi$ where $\xi$ comes from the uniform distribution over the ball of radius $R = \sqrt{\frac{\mathcal{F}}{L_1}}$. Consider that the iterates $\mathbf{x}_i$ follow the Gradient Tracking Update with stepsize $\eta_2$ such that $\eta_2, \alpha$ satisfy conditions (41) - (44) and further $\eta_2 \leq \min\{1, \frac{1-\sigma}{L_1}\}$ and $\alpha\eta_2 \leq \min\left\{ \left(\frac{\delta_2(\sqrt{2}-1)}{8\sqrt{2}\hat{c}\cdot c_{new}}\right)^2 \frac{(1-\sigma)|\lambda_{\min}(H)|^2}{d\log^2\left(\frac{dL_1}{\gamma\delta_2}\right)}, \frac{1-\sigma}{192L_1^2}\right\}$. Then with probability at least $1-\delta_2$ we have the following*

$$H(\underline{\mathbf{x}}^{T_{cap}}, \underline{\mathbf{y}}^{T_{cap}}) - H(\underline{\mathbf{x}}^{-1}, \underline{\mathbf{y}}^{-1}) \leq -\mathcal{F} \tag{207}$$

Finally notice that as shown is Theorem 7, we can track whether the second phase succeeded in substantially decreasing the potential function by two runs of the average consensus protocol. One on iterate $\underline{\mathbf{x}}^{-1}$ and one on $\underline{\mathbf{x}}^{T_{cap}}$. If there is substantial decrease then $\underline{\mathbf{x}}^{T_{cap}}$ and $\underline{\mathbf{y}}^{T_{cap}}$ are provided as a starting point for the first phase. If the decrease is not substantial then with probability $1-\delta_2$ the point $\hat{\underline{\mathbf{x}}}^{-1}$ is a $(\epsilon_1 + L_1\epsilon_2, \gamma)-$approximate second order stationary point and we terminate.

## 10.1 Convergence Rates

**Theorem 6.** *Let $\epsilon, \rho$ be the target gradient and consensus error accuracy. Assume condition 58 and let $\hat{\epsilon} = \min\left\{\epsilon, \rho, \frac{L_1}{\sqrt{2(1-\sigma)}}\sqrt{\alpha\eta_2\mathcal{F}}, \sqrt{\mathcal{F}}\frac{L_1}{\sqrt{2+2L_1^2}}\right\}$ and assume that in the first phase the iterates $\mathbf{x}_i$ follow the Gradient Tracking Update with stepsize $\eta_1$ such that $\eta_1, \alpha$ satisfy conditions (41) - (44). Let $T_1 = 4e\frac{4(f(\mathbf{x}^0)-f^*)}{\min\{\eta_1, 1-\sigma\}\hat{\epsilon}^2} + 1$. Let $\underline{\mathbf{x}}^{-1}$ the point the first phase outputs and assume $\lambda_{\min}(\nabla^2 f(\hat{\mathbf{x}}^{-1})) \leq -\gamma$. Further $\forall i$ let $\mathbf{x}_i^0 = \mathbf{x}_i^{-1} + \xi$ where $\xi$ comes from the uniform distribution over the ball of radius $R = \sqrt{\frac{\mathcal{F}}{L_1}}$. Consider that the iterates $\mathbf{x}_i$ follow the Gradient Tracking Update with stepsize $\eta_2$ such that $\eta_2, \alpha$ satisfy conditions (41) - (44) and further $\eta_2 \leq \min\{1, \frac{1-\sigma}{L_1}\}$ and $\alpha\eta_2 \leq \min\left\{ \left(\frac{\delta_2(\sqrt{2}-1)}{8\sqrt{2}\hat{c}\cdot c_{new}}\right)^2 \frac{(1-\sigma)|\lambda_{\min}(H)|^2}{d\log^2\left(\frac{dL_1}{\gamma\delta_2}\right)}, \frac{1-\sigma}{192L_1^2}\right\}$. Then with probability at least $(1-\delta_1)(1-\delta_2)$ we have the*

$$H(\underline{\mathbf{x}}^{T_{cap}}, \underline{\mathbf{y}}^{T_{cap}}) - H(\underline{\mathbf{x}}^0, \underline{\mathbf{y}}^0) \leq -\mathcal{F} \tag{208}$$

*where $\underline{\mathbf{x}}^0$ is the first iterate of the first phase and $\underline{\mathbf{x}}^{T_{cap}}$ is the last iterate of the second phase. Further let us denote the average consensus iterations for phase I and II with $T_{con}$ and the total number of communication rounds throughout both phases with $T_{1,2}$. Then it holds:*

$$T_{1,2} = 4e\frac{4(f(\mathbf{x}^0)-f^*)}{\min\{\eta_1, 1-\sigma\}\hat{\epsilon}^2} + \hat{c}\frac{\log(d\kappa/\delta)}{\eta_2|\lambda_{\min}(H)|} + T_{con} = \tilde{\mathcal{O}}\left(\min\left\{\frac{1}{\eta_1\hat{\epsilon}^2}, \frac{1}{\eta_2\gamma}\right\}\right) \tag{209}$$

*Proof.* From Theorem 5 we have $\left\|\frac{1}{m}\sum_{i=1}^{m}\nabla f_i(\mathbf{x}_i^{-1})\right\|^2 + \frac{1}{m}\left\|\underline{\mathbf{x}}^{-1} - \hat{\underline{\mathbf{x}}}^{-1}\right\|^2 \leq \hat{\epsilon}^2$ with probability $1-\delta_1$. Since the conditions of Corollary 5 hold we also have that with probability at least $1-\delta_2$

$$H(\underline{\mathbf{x}}^{T_{cap}}, \underline{\mathbf{y}}^{T_{cap}}) - H(\underline{\mathbf{x}}^0, \underline{\mathbf{y}}^0) \leq H(\underline{\mathbf{x}}^{T_{cap}}, \underline{\mathbf{y}}^{T_{cap}}) - H(\underline{\mathbf{x}}^{-1}, \underline{\mathbf{y}}^{-1}) + H(\underline{\mathbf{x}}^{-1}, \underline{\mathbf{y}}^{-1}) - H(\underline{\mathbf{x}}^0, \underline{\mathbf{y}}^0) \tag{210}$$

$$\leq -\mathcal{F} + H(\underline{\mathbf{x}}^{-1}, \underline{\mathbf{y}}^{-1}) - H(\underline{\mathbf{x}}^0, \underline{\mathbf{y}}^0) \tag{211}$$

$$\leq -\mathcal{F} \tag{212}$$

where the last inequality comes from the monotonicity of the potential function through the first phase. The total number of communication rounds include the first phase iterations and consensus rounds as well as the second phase iterations and consensus rounds. $\qquad\square$

**Lemma 18.** *Let $\epsilon, \rho$ be the target gradient and consensus error accuracy. Further let $\hat{\epsilon} = \min\left\{ \epsilon, \rho, \frac{L_1}{\sqrt{2(1-\sigma)}}\sqrt{\alpha \eta_2 \mathcal{F}}, \sqrt{\mathcal{F}}\frac{L_1}{\sqrt{2+2L_1^2}} \right\}$. There exist $\alpha = \mathcal{O}\left((1-\sigma)^2\right), \eta_1 = \mathcal{O}\left((1-\sigma)^2\right)$ and $\eta_2 = \tilde{\mathcal{O}}\left(\frac{\gamma^2}{d(1-\sigma)}\right)$ such that the conditions of Theorem 6 hold and the communication rounds throughout the first and the second phases is*

$$T_{1,2} = \tilde{\mathcal{O}}\left( \min\left\{ \frac{1}{(1-\sigma)^2 \hat{\epsilon}^2}, \frac{(1-\sigma)d}{\gamma^3} \right\} \right) \tag{213}$$

*Proof.* From theorem 6 we have

$$T_{1,2} = \tilde{\mathcal{O}}\left( \min\left\{ \frac{1}{\eta_1 \hat{\epsilon}^2}, \frac{1}{\eta_2 \gamma} \right\} \right) = \tilde{\mathcal{O}}\left( \min\left\{ \frac{1}{(1-\sigma)^2 \hat{\epsilon}^2}, \frac{d}{\gamma^3} \right\} \right) \tag{214}$$

$\square$

Recall that after each pass of phase II the potential function is decreased at least by $\tilde{\mathcal{O}}(\gamma^3)$ and thus the following corollary captures the overall communication complexity of our algorithm before it reaches some approximate second order stationary point.

**Corollary 6.** *Assume the conditions of Lemma 18 hold. Then the overall communication rounds performed by our algorithm is at most*

$$T_{total} = \tilde{\mathcal{O}}\left( \min\left\{ \frac{1}{(1-\sigma)^2 \gamma^3 \hat{\epsilon}^2}, \frac{d}{\gamma^6} \right\} \right) \tag{215}$$

If we further assume that the strict saddle property holds then by setting $\hat{\epsilon} \le \frac{\theta}{1+L_1}$ our algorithm converges to local minima. This claim is an immediate corollary of the following lemma.

**Lemma 19.** *Assume that conditions (54), (55) hold and further $\epsilon_1 + L_1 \epsilon_2 < \theta$. Then either $\hat{\mathbf{x}}^{-1}$ is $\nu-$close to some local minimum or $\lambda_{\min}(\nabla^2 f(\hat{\mathbf{x}}^{-1})) \le -\zeta$.*

*Proof.* We can bound $\left\| \nabla f(\hat{\mathbf{x}}^{-1}) \right\|$ as follows :

$$\left\| \nabla f(\hat{\mathbf{x}}^{-1}) \right\| \le \left\| \frac{1}{m}\sum_{i=1}^m \nabla f_i(\mathbf{x}_i^{-1}) - \frac{1}{m}\sum_{i=1}^m \nabla f_i(\hat{\mathbf{x}}^{-1}) \right\| + \left\| \frac{1}{m}\sum_{i=1}^m \nabla f_i(\mathbf{x}_i^{-1}) \right\| \tag{216}$$

$$\le \frac{L_1}{m}\sum_{i=1}^m \left\| \mathbf{x}_i^{-1} - \hat{\mathbf{x}}^{-1} \right\| + \epsilon_1 \tag{217}$$

$$\le \frac{L_1}{m}\sqrt{m\sum_{i=1}^m \left\| \mathbf{x}_i^{-1} - \hat{\mathbf{x}}^{-1} \right\|^2} + \epsilon_1 \tag{218}$$

$$\le L_1\sqrt{\frac{1}{m}\left\| \underline{\mathbf{x}}^{-1} - \underline{\hat{\mathbf{x}}}^{-1} \right\|^2} + \epsilon_1 \tag{219}$$

$$\le L_1 \epsilon_2 + \epsilon_1 \tag{220}$$

$$< \theta \tag{221}$$

Where the second inequality comes from (54) and the last in equality comes from (55). The result follows from Assumption 4 . $\square$

## 11 Average Consensus Protocol

We will now present how to utilize the average consensus protocol to achieve the following objectives:

1. Initialize $\underline{\mathbf{y}}^0$ at the beginning of phase II such that

$$\hat{\mathbf{y}}^0 = \frac{1}{m}\sum_{i=1}^m \nabla f_i(\mathbf{x}_i^0) \quad and \quad \frac{1}{m}\left\| \underline{\mathbf{y}}^0 - \underline{\hat{\mathbf{y}}}^0 \right\|^2 \le \frac{L_1^2}{2(1-\sigma)}\eta_2 \mathcal{F}.$$

2. Coordinate the nodes to pick a phase I iteration $r$ such that
$$\left\| \frac{1}{m} \sum_{i=1}^{m} \nabla f_i(\mathbf{x}_i^r) \right\|^2 + \frac{1}{m} \left\| \underline{\mathbf{x}}^r - \underline{\hat{\mathbf{x}}}^r \right\|^2 \leq \epsilon^2.$$

3. Track the potential function decrease before and at the end of phase II,
$H(\underline{\mathbf{x}}^{T_{cap}}, \underline{\mathbf{y}}^{T_{cap}}) - H(\underline{\mathbf{x}}^{-1}, \underline{\mathbf{y}}^{-1})$.

**Average Consensus Update Rule for some vector $\underline{\mathbf{x}}^r$**

$$\underline{\mathbf{x}}^{r,0} = \underline{\mathbf{x}}^r \tag{222}$$
$$\underline{\mathbf{x}}^{r,t+1} = \underline{\mathbf{W}}\underline{\mathbf{x}}^{r,t} \tag{223}$$

## 11.1  Initializing the second phase

Towards proving the first of our objectives we present the following lemma where we show that the consensus error diminishes exponentially fast in the number of iterations. Notice that since $\eta_2 \mathcal{F} = \tilde{\mathcal{O}}\left(\frac{\gamma^5}{d}\right)$ the number of iterations of the average consensus protocol have a logarithmic dependence on $\gamma, d$ and the initial error $\sum_{i=1}^{m} \left\| \nabla f_i(\mathbf{x}_i^0) - \sum_{j=1}^{m} \nabla f_j(\mathbf{x}_j^0) \right\|^2$.

**Lemma 20.** *Consider the iterates $\mathbf{x}_i^0$'s at the beginning of phase II and let each node, $i$, set $\mathbf{y}_i^0 = \nabla f_i(\mathbf{x}_i^0)$. Let $\underline{\mathbf{y}}^{0,0} = \underline{\mathbf{y}}^0$, $\quad \underline{\mathbf{y}}^{0,t+1} = \underline{\mathbf{W}}\underline{\mathbf{y}}^{0,t}$ and $\hat{\mathbf{y}}^0 = \frac{1}{m}\sum_{i=1}^{m}\mathbf{y}_i^0 = \frac{1}{m}\sum_{i=1}^{m}\nabla f_i(\mathbf{x}_i^0)$. After $t_{\mathbf{y}} + 1$ rounds of the average consensus protocol on $\mathbf{y}_i^0$'s we have the following guarantee*

$$\frac{1}{m}\left\|\underline{\mathbf{y}}^{0,t+1} - \underline{\hat{\mathbf{y}}}^0\right\|^2 \leq \frac{L_1^2}{2(1-\sigma)}\eta_2\mathcal{F} \tag{224}$$

*for $t_{\mathbf{y}} = \left\lfloor \frac{\log\left(\sqrt{\eta_2\mathcal{F}}\right) - \log\left(\|\underline{\mathbf{y}}^0 - \underline{\hat{\mathbf{y}}}^0\|\right) + \log\left(\sqrt{m}L_1\right) - \log\left(\sqrt{2(1-\sigma)}\right)}{\log(\sigma)} \right\rfloor.$*

*Proof.* From the consensus update rule we know
$$\left\|\underline{\mathbf{y}}^{0,t+1} - \underline{\hat{\mathbf{y}}}^0\right\| \leq \left\|\underline{\mathbf{W}}\left(\underline{\mathbf{y}}^{0,t} - \underline{\hat{\mathbf{y}}}^0\right)\right\| \leq \sigma\left\|\underline{\mathbf{y}}^{0,t} - \underline{\hat{\mathbf{y}}}^0\right\| \tag{225}$$
Thus we derive the following
$$\left\|\underline{\mathbf{y}}^{0,t+1} - \underline{\hat{\mathbf{y}}}^0\right\| \leq \sigma^{t+1}\left\|\underline{\mathbf{y}}^0 - \underline{\hat{\mathbf{y}}}^0\right\| \tag{226}$$
Solving for $t$ that guarantees $\sigma^{t+1}\left\|\underline{\mathbf{y}}^0 - \underline{\hat{\mathbf{y}}}^0\right\| \leq \frac{L_1\sqrt{m}}{\sqrt{2(1-\sigma)}}\sqrt{\eta_2\mathcal{F}}$ we get

$$\sigma^{t+1} \leq \frac{L_1\sqrt{m}}{\left\|\underline{\mathbf{y}}^0 - \underline{\hat{\mathbf{y}}}^0\right\|\sqrt{2(1-\sigma)}}\sqrt{\eta_2\mathcal{F}} \tag{227}$$

$$(t+1)\log(\sigma) \leq \log\left(\sqrt{m}L_1\right) + \log\left(\sqrt{\eta_2\mathcal{F}}\right) - \log\left(\left\|\underline{\mathbf{y}}^0 - \underline{\hat{\mathbf{y}}}^0\right\|\right) - \log\left(\sqrt{2(1-\sigma)}\right) \tag{228}$$

$$t \geq \frac{\log\left(\sqrt{m}L_1\right) + \log\left(\sqrt{\eta_2\mathcal{F}}\right) - \log\left(\left\|\underline{\mathbf{y}}^0 - \underline{\hat{\mathbf{y}}}^0\right\|\right) - \log\left(\sqrt{2(1-\sigma)}\right)}{\log(\sigma)} - 1 \tag{229}$$

Thus for $t = \left\lfloor \frac{\log\left(\sqrt{\eta_2\mathcal{F}}\right) - \log\left(\|\underline{\mathbf{y}}^0 - \underline{\hat{\mathbf{y}}}^0\|\right) + \log\left(\sqrt{m}L_1\right) - \log\left(\sqrt{2(1-\sigma)}\right)}{\log(\sigma)} \right\rfloor$ we have

$$\left\|\underline{\mathbf{y}}^{0,t+1} - \underline{\hat{\mathbf{y}}}^0\right\| \leq \frac{L_1\sqrt{m}}{\sqrt{2(1-\sigma)}}\sqrt{\eta_2\mathcal{F}} \tag{230}$$

which implies

$$\frac{1}{m}\left\|\underline{\mathbf{y}}^{0,t+1} - \underline{\hat{\mathbf{y}}}^0\right\|^2 \leq \frac{L_1^2}{2(1-\sigma)}\eta_2\mathcal{F} \tag{231}$$

$\square$

Initializing $\mathbf{y}_i^0 = \mathbf{y}_i^{0,t_{\mathbf{y}}+1}$ we achieve our first objective.

## 11.2 Choosing a good iterate

In order to achieve our second objective first we provide upper bounds for $\|\mathbf{x}^r - \hat{\mathbf{x}}^r\|$ and $\|\mathbf{y}^r - \hat{\mathbf{y}}^r\|$ for any iteration $r$ of our lagorithm.

**Lemma 21.** *Consider any iterates $\underline{\mathbf{x}}^r, \underline{\mathbf{y}}^r$ following Gradient Tracking Update with $\eta_1, \alpha$ that satisfy conditions (41) - (44). Also assume that the potential function decreases between consecutive first phases. Then*

$$\|\underline{\mathbf{x}}^r - \hat{\underline{\mathbf{x}}}^r\| \leq \sqrt{m(f(\underline{\mathbf{x}}^0) - \underline{f}) + 2m\mathcal{F}} \tag{232}$$

$$\|\underline{\mathbf{y}}^r - \hat{\underline{\mathbf{y}}}^r\| \leq \sqrt{\frac{m}{\alpha}(f(\underline{\mathbf{x}}^0) - \underline{f}) + \frac{2m}{\alpha}\mathcal{F}} \tag{233}$$

*Proof.*

$$H(\underline{\mathbf{x}}^0, \underline{\mathbf{y}}^0) - H(\underline{\mathbf{x}}^r, \underline{\mathbf{y}}^r) = f(\hat{\mathbf{x}}^0) + 0 + 0 - f(\hat{\mathbf{x}}^r) - \frac{1}{m}\|\underline{\mathbf{x}}^r - \hat{\underline{\mathbf{x}}}^r\|^2 - \frac{\alpha}{m}\|\underline{\mathbf{y}}^r - \hat{\underline{\mathbf{y}}}^r\|^2 \geq -2\mathcal{F} \tag{234}$$

where the last inequality holds because the potential function is non-increasing throughout a single phase and due to Lemma 11. Thus we get

$$\frac{1}{m}\|\underline{\mathbf{x}}^r - \hat{\underline{\mathbf{x}}}^r\|^2 + \frac{\alpha}{m}\|\underline{\mathbf{y}}^r - \hat{\underline{\mathbf{y}}}^r\|^2 \leq f(\hat{\mathbf{x}}^0) - f(\hat{\mathbf{x}}^r) + 2\mathcal{F} \leq f(\hat{\mathbf{x}}^0) - f^* + 2\mathcal{F} \tag{235}$$

which derives

$$\|\underline{\mathbf{x}}^r - \hat{\underline{\mathbf{x}}}^r\|^2 \leq m(f(\hat{\mathbf{x}}^0) - f^*) + 2m\mathcal{F} \tag{236}$$

$$\|\underline{\mathbf{y}}^r - \hat{\underline{\mathbf{y}}}^r\|^2 \leq \frac{m}{\alpha}(f(\hat{\mathbf{x}}^0) - f^*) + \frac{2m}{\alpha}\mathcal{F} \tag{237}$$

The result follows after taking the square roots of both bounds. $\square$

In the following lemma we are going to show that the consensus error diminishes exponentially fast in the number of iterations.

**Lemma 22.** *Consider any iterates $\underline{\mathbf{x}}^r, \underline{\mathbf{y}}^r$ following Gradient Tracking Update with $\eta_1, \alpha$ that satisfy conditions (41) - (44). Also assume that the potential function decreases between consecutive first phases. Let $\underline{\mathbf{y}}^{r,0} = \underline{\mathbf{y}}^r$, $\underline{\mathbf{y}}^{r,t+1} = \underline{\mathbf{W}}\underline{\mathbf{y}}^{r,t}$ and $\hat{\mathbf{y}}^r = \frac{1}{m}\sum_{i=1}^{m} \mathbf{y}_i^r$. After $t_\mathbf{y} + 1$ rounds of the average consensus protocol on $\mathbf{y}_i^r$'s we have the following guarantee*

$$\|\underline{\mathbf{y}}^{r,t_\mathbf{y}+1} - \hat{\underline{\mathbf{y}}}^r\| \leq \epsilon^c \tag{238}$$

*for $t_\mathbf{y} = \left\lfloor \frac{c\log(\epsilon) + \log(\sqrt{\alpha}) - \log\left(\sqrt{m(f(\mathbf{x}^0) - f^*) + 2m\mathcal{F}}\right)}{\log(\sigma)} \right\rfloor$ and any positive constant c.*

*Similarly let $\underline{\mathbf{x}}^{r,0} = \underline{\mathbf{x}}^r$, $\underline{\mathbf{x}}^{r,t+1} = \underline{\mathbf{W}}\underline{\mathbf{x}}^{r,t}$ and $\hat{\mathbf{x}}^r = \frac{1}{m}\sum_{i=1}^{m} \mathbf{x}_i^r$. After $t_\mathbf{x} + 1$ rounds of the average consensus protocol on on $\mathbf{x}_i^r$'s we have the following guarantee*

$$\|\underline{\mathbf{x}}^{r,t_\mathbf{x}+1} - \hat{\underline{\mathbf{x}}}^r\| \leq \epsilon^c \tag{239}$$

*for $t_\mathbf{x} = \left\lfloor \frac{c\log(\epsilon) - \log\left(\sqrt{m(f(\mathbf{x}^0) - f^*) + 2m\mathcal{F}}\right)}{\log(\sigma)} \right\rfloor$ and any positive constant c.*

*Proof.* From the consensus update rule we know

$$\|\underline{\mathbf{y}}^{r,t+1} - \hat{\underline{\mathbf{y}}}^r\| \leq \|\underline{\mathbf{W}}\left(\underline{\mathbf{y}}^{r,t} - \hat{\underline{\mathbf{y}}}^r\right)\| \leq \sigma \|\underline{\mathbf{y}}^{r,t} - \hat{\underline{\mathbf{y}}}^r\| \tag{240}$$

Thus we derive the following

$$\|\underline{\mathbf{y}}^{r,t+1} - \hat{\underline{\mathbf{y}}}^r\| \leq \sigma^{t+1} \|\underline{\mathbf{y}}^{r,0} - \hat{\underline{\mathbf{y}}}^r\| \leq \sigma^{t+1}\sqrt{\frac{m}{\alpha}(f(\mathbf{x}^0) - f^*) + \frac{2m}{\alpha}\mathcal{F}} \tag{241}$$

where the second inequality comes from Lemma 21. Solving for $t$ that guarantees $\sigma^{t+1}\sqrt{\frac{m}{\alpha}(f(\mathbf{x}^0)-f^*)+\frac{2m}{\alpha}\mathcal{F}}\leq\epsilon^c$ we get

$$\sigma^{t+1}\leq\frac{\sqrt{\alpha}\epsilon^c}{\sqrt{m(f(\mathbf{x}^0)-f^*)+2m\mathcal{F}}} \tag{242}$$

$$(t+1)\log(\sigma)\leq\log(\sqrt{\alpha}\epsilon^c)-\log\left(\sqrt{m(f(\mathbf{x}^0)-f^*)+2m\mathcal{F}}\right) \tag{243}$$

$$t\geq\frac{c\log(\epsilon)+\log(\sqrt{\alpha})-\log\left(\sqrt{m(f(\mathbf{x}^0)-f^*)+2m\mathcal{F}}\right)}{\log(\sigma)}-1 \tag{244}$$

Thus for $t=\left\lfloor\frac{c\log(\epsilon)+\log(\sqrt{\alpha})-\log\left(\sqrt{m(f(\mathbf{x}^0)-f^*)+2m\mathcal{F}}\right)}{\log(\sigma)}\right\rfloor$ we have

$$\left\|\underline{\mathbf{y}}^{r,t+1}-\hat{\underline{\mathbf{y}}}^r\right\|\leq\epsilon^c \tag{245}$$

Similarly from the consensus update rule we know

$$\left\|\underline{\mathbf{x}}^{r,t+1}-\hat{\underline{\mathbf{x}}}^r\right\|\leq\left\|\mathbf{W}\left(\underline{\mathbf{x}}^{r,t}-\hat{\underline{\mathbf{x}}}^r\right)\right\|\leq\sigma\left\|\underline{\mathbf{x}}^{r,t}-\hat{\underline{\mathbf{x}}}^r\right\| \tag{246}$$

Thus we derive the following

$$\left\|\underline{\mathbf{x}}^{r,t+1}-\hat{\underline{\mathbf{x}}}^r\right\|\leq\sigma^{t+1}\left\|\underline{\mathbf{x}}^{r,0}-\hat{\underline{\mathbf{x}}}^r\right\|\leq\sigma^{t+1}\sqrt{m(f(\mathbf{x}^0)-f^*)+2m\mathcal{F}} \tag{247}$$

where the second inequality comes from Lemma 21. Solving for $t$ that guarantees $\sigma^{t+1}\sqrt{m(f(\mathbf{x}^0)-f^*)+2m\mathcal{F}}\leq\epsilon^c$ we get

$$\sigma^{t+1}\leq\frac{\epsilon^c}{\sqrt{m(f(\mathbf{x}^0)-f^*)+2m\mathcal{F}}} \tag{248}$$

$$(t+1)\log(\sigma)\leq c\log(\epsilon)-\log\left(\sqrt{m(f(\mathbf{x}^0)-f^*)+2m\mathcal{F}}\right) \tag{249}$$

$$t\geq\frac{c\log(\epsilon)-\log\left(\sqrt{m(f(\mathbf{x}^0)-f^*)+2m\mathcal{F}}\right)}{\log(\sigma)}-1 \tag{250}$$

Thus for $t=\left\lfloor\frac{c\log(\epsilon)-\log\left(\sqrt{m(f(\mathbf{x}^0)-f^*)+2m\mathcal{F}}\right)}{\log(\sigma)}\right\rfloor$ we have

$$\left\|\underline{\mathbf{x}}^{r,t+1}-\hat{\underline{\mathbf{x}}}^r\right\|\leq\epsilon^c \tag{251}$$

$\square$

The following corollary suggest that after a logarithmic number of iterations with respect to $\epsilon$, every node is going to have an accurate estimate of the average vector of interest.

**Corollary 7.** *After* $\left\lfloor\frac{c\log(\epsilon)+\log(\sqrt{\alpha})-\log\left(\sqrt{m(f(\mathbf{x}^0)-f^*)+2m\mathcal{F}}\right)}{\log(\sigma)}\right\rfloor+1$ *rounds of the average consensus protocol on* $\mathbf{y}_i^r$*'s we have the following guarantee*

$$\left\|\mathbf{y}_i^{r,t}-\hat{\mathbf{y}}^r\right\|\leq\epsilon^c,\quad\forall i\in[m] \tag{252}$$

*After* $\left\lfloor\frac{c\log(\epsilon)-\log\left(\sqrt{m(f(\mathbf{x}^0)-f^*)+2m\mathcal{F}}\right)}{\log(\sigma)}\right\rfloor+1$ *rounds of the average consensus protocol on* $\mathbf{x}_i^r$*'s we have the following guarantee*

$$\left\|\mathbf{x}_i^{r,t}-\hat{\mathbf{x}}^r\right\|\leq\epsilon^c,\quad\forall i\in[m] \tag{253}$$

The next lemma provides bounds that we will use in order to argue about the number of iterations required when we run the average consensus protocol on $\left\|\mathbf{y}_i^{r,t_{\mathbf{y}}+1}-\mathbf{y}_i^r\right\|^2$'s and on $\left\|\mathbf{x}_i^{r,t_{\mathbf{x}}+1}-\mathbf{x}_i^r\right\|^2$'s.

**Lemma 23.** *Consider any iterates $\underline{\mathbf{x}}^r, \underline{\mathbf{y}}^r$ following Gradient Tracking Update with $\eta_1, \alpha$ that satisfy conditions (41) - (44). Also assume that the potential function decreases between consecutive first phases. Let $\underline{\mathbf{y}}^{r,0} = \underline{\mathbf{y}}^r$ and $\hat{\mathbf{y}}^r = \frac{1}{m}\sum_{i=1}^{m}\mathbf{y}_i^r$, $\quad \underline{\mathbf{x}}^{r,0} = \underline{\mathbf{x}}^r$ and $\hat{\mathbf{x}}^r = \frac{1}{m}\sum_{i=1}^{m}\mathbf{x}_i^r$. Further let*

$$t_{\mathbf{y}} = \left\lfloor \frac{c\log(\epsilon)+\log(\sqrt{\alpha})-\log\left(\sqrt{m(f(\mathbf{x}^0)-f^*)+2m\mathcal{F}}\right)}{\log(\sigma)} \right\rfloor \text{ and } t_{\mathbf{x}} = \left\lfloor \frac{c\log(\epsilon)-\log\left(\sqrt{m(f(\mathbf{x}^0)-f^*)+2m\mathcal{F}}\right)}{\log(\sigma)} \right\rfloor.$$

*The following bounds hold for $\epsilon \leq \min\{1, \sqrt{\frac{m}{\alpha 16}}(f(\mathbf{x}^0) - f^* + 2\mathcal{F})\}$ and $c \geq 1$*

$$\sum_{i=1}^{m}\left(\left\|\mathbf{y}_i^{r,t_{\mathbf{y}}+1} - \mathbf{y}_i^r\right\|^2 - \frac{1}{m}\sum_{j=1}^{m}\left\|\mathbf{y}_j^{r,t_{\mathbf{y}}+1} - \mathbf{y}_j^r\right\|^2\right)^2 \leq \frac{16m^3}{\alpha^2}(f(\mathbf{x}^0) - f^* + \mathcal{F})^2 \quad (254)$$

$$\sum_{i=1}^{m}\left(\left\|\mathbf{x}_i^{r,t_{\mathbf{x}}+1} - \mathbf{x}_i^r\right\|^2 - \frac{1}{m}\sum_{j=1}^{m}\left\|\mathbf{x}_j^{r,t_{\mathbf{x}}+1} - \mathbf{x}_j^r\right\|^2\right)^2 \leq 16m^3(f(\mathbf{x}^0) - f^* + \mathcal{F})^2 \quad (255)$$

*Proof.* First notice that

$$\left\|\mathbf{y}_i^{r,t_{\mathbf{y}}+1} - \mathbf{y}_i^r\right\|^2 \leq \left\|\mathbf{y}_i^{r,t_{\mathbf{y}}+1} - \hat{\mathbf{y}}^r\right\|^2 + \|\hat{\mathbf{y}}^r - \mathbf{y}_i^r\|^2 + 2\langle\left\|\mathbf{y}_i^{r,t_{\mathbf{y}}+1} - \hat{\mathbf{y}}^r\right\|, \|\hat{\mathbf{y}}^r - \mathbf{y}_i^r\|\rangle \quad (256)$$

$$\leq \epsilon^{2c} + \frac{m}{\alpha}(f(\mathbf{x}^0) - f^* + 2\mathcal{F}) + 2\epsilon^c\sqrt{\frac{m}{\alpha}(f(\mathbf{x}^0) - f^* + 2\mathcal{F})} \quad (257)$$

$$\leq \frac{2m}{\alpha}(f(\mathbf{x}^0) - f^* + 2\mathcal{F}) \quad (258)$$

In the second inequality we use the results from Lemma 21 and Corollary 7. The third inequality holds for sufficiently small $\epsilon \leq \min\{1, \sqrt{\frac{m}{16\alpha}(f(\mathbf{x}^0) - f^* + 2\mathcal{F})}\}$. Thus we have

$$\sum_{i=1}^{m}\left(\left\|\mathbf{y}_i^{r,t_{\mathbf{y}}+1} - \mathbf{y}_i^r\right\|^2 - \frac{1}{m}\sum_{j=1}^{m}\left\|\mathbf{y}_j^{r,t_{\mathbf{y}}+1} - \mathbf{y}_j^r\right\|^2\right)^2$$

$$\leq \sum_{i=1}^{m}\left(\left\|\mathbf{y}_i^{r,t_{\mathbf{y}}+1} - \mathbf{y}_i^r\right\|^2 + \frac{1}{m}\sum_{j=1}^{m}\left\|\mathbf{y}_j^{r,t_{\mathbf{y}}+1} - \mathbf{y}_j^r\right\|^2\right)^2 \quad (259)$$

$$\leq \sum_{i=1}^{m}\left(\frac{2m}{\alpha}(f(\mathbf{x}^0) - f^* + 2\mathcal{F}) + \frac{1}{m}\sum_{j=1}^{m}\frac{2m}{\alpha}(f(\mathbf{x}^0) - f^* + 2\mathcal{F})^2\right) \quad (260)$$

$$\leq \sum_{i=1}^{m}\frac{16m^2}{\alpha^2}(f(\mathbf{x}^0) - f^* + 2\mathcal{F})^2 \quad (261)$$

$$\leq \frac{16m^3}{\alpha^2}(f(\mathbf{x}^0) - f^* + 2\mathcal{F})^2 \quad (262)$$

Notice that

$$\left\|\mathbf{x}_i^{r,t_{\mathbf{x}}+1} - \mathbf{x}_i^r\right\|^2 \leq \left\|\mathbf{x}_i^{r,t_{\mathbf{x}}+1} - \hat{\mathbf{x}}^r\right\|^2 + \|\hat{\mathbf{x}}^r - \mathbf{x}_i^r\|^2 + 2\langle\left\|\mathbf{x}_i^{r,t_{\mathbf{x}}+1} - \hat{\mathbf{x}}^r\right\|, \|\hat{\mathbf{x}}^r - \mathbf{x}_i^r\|\rangle \quad (263)$$

$$\leq \epsilon^{2c} + m(f(\mathbf{x}^0) - f^* + 2\mathcal{F}) + 2\epsilon^c\sqrt{m(f(\mathbf{x}^0) - f^* + 2\mathcal{F})} \quad (264)$$

$$\leq 2m(f(\mathbf{x}^0) - f^* + 2\mathcal{F}) \quad (265)$$

In the second inequality we use the results from Lemma 21 and Corollary 7. The third inequality holds for sufficiently small $\epsilon \leq \min\{1, \sqrt{\frac{m}{16}(f(\underline{\mathbf{x}}^0) - \underline{f})}\}$. Thus we have

$$\sum_{i=1}^{m} \left( \left\| \mathbf{x}_i^{r,t_{\mathbf{x}}+1} - \mathbf{x}_i^r \right\|^2 - \frac{1}{m} \sum_{j=1}^{m} \left\| \mathbf{x}_j^{r,t_{\mathbf{x}}+1} - \mathbf{x}_j^r \right\|^2 \right)^2 \tag{266}$$

$$\leq \sum_{i=1}^{m} \left( \left\| \mathbf{x}_i^{r,t_{\mathbf{x}}+1} - \mathbf{x}_i^r \right\|^2 + \frac{1}{m} \sum_{j=1}^{m} \left\| \mathbf{x}_j^{r,t_{\mathbf{x}}+1} - \mathbf{x}_j^r \right\|^2 \right)^2 \tag{267}$$

$$\leq \sum_{i=1}^{m} \left( 2m(f(\mathbf{x}^0) - f^* + 2\mathcal{F}) + \frac{1}{m} \sum_{j=1}^{m} 2m(f(\mathbf{x}^0) - f^* + 2\mathcal{F}) \right)^2 \tag{268}$$

$$\leq \sum_{i=1}^{m} 16m^2(f(\mathbf{x}^0) - f^* + 2\mathcal{F})^2 \tag{269}$$

$$\leq 16m^3(f(\mathbf{x}^0) - f^* + 2\mathcal{F})^2 \tag{270}$$

$$\square$$

The next lemma provides an upper bound on the number of iterations required when we run the average consensus protocol on $\left\| \mathbf{y}_i^{r,t_{\mathbf{y}}+1} - \mathbf{y}_i^r \right\|^2$'s and on $\left\| \mathbf{x}_i^{r,t_{\mathbf{x}}+1} - \mathbf{x}_i^r \right\|^2$'s in order to achieve accuracy $\epsilon^c$.

**Lemma 24.** *Consider any iterates $\underline{\mathbf{x}}^r, \underline{\mathbf{y}}^r$ following Gradient Tracking Update with $\eta_1, \alpha$ that satisfy conditions (41) - (44) and $\epsilon \leq \min\{1, \sqrt{\frac{m}{16\alpha}(f(\mathbf{x}^0) - f^* + 2\mathcal{F})}\}$, $c \geq 1$. Also assume that the potential function decreases between consecutive first phases. Let $\underline{\mathbf{y}}^{r,0} = \underline{\mathbf{y}}^r$ and $\hat{\mathbf{y}}^r = \frac{1}{m} \sum_{i=1}^{m} \mathbf{y}_i^r$, $\underline{\mathbf{x}}^{r,0} = \underline{\mathbf{x}}^r$ and $\hat{\mathbf{x}}^r = \frac{1}{m} \sum_{i=1}^{m} \mathbf{x}_i^r$. Further let $t_{\mathbf{y}} = \left\lfloor \frac{c\log(\epsilon) + \log(\sqrt{\alpha}) - \log\left(\sqrt{m(f(\mathbf{x}^0) - f^*) + 2m\mathcal{F}}\right)}{\log(\sigma)} \right\rfloor$ and $t_{\mathbf{x}} = \left\lfloor \frac{c\log(\epsilon) - \log\left(\sqrt{m(f(\mathbf{x}^0) - f^*) + 2m\mathcal{F}}\right)}{\log(\sigma)} \right\rfloor$.*
*Define $\mathbf{z}_i^{r,0} := \left\| \mathbf{y}_i^{r,t_{\mathbf{y}}+1} - \mathbf{y}_i^r \right\|^2$, $\hat{\mathbf{z}}^r := \frac{1}{m} \sum_{i=1}^{m} \left\| \mathbf{y}_i^{r,t_{\mathbf{y}}+1} - \mathbf{y}_i^r \right\|^2$ and $t_{\mathbf{z}} = \left\lfloor \frac{c\log(\epsilon) + \log\alpha - \log\left(4m^2(f(\mathbf{x}^0) - f^* + 2\mathcal{F})\right)}{\log(\sigma)} \right\rfloor$.*

*After $t_{\mathbf{z}} + 1$ rounds of the average consensus protocol on $\left\| \mathbf{y}_i^{r,t_{\mathbf{y}}+1} - \mathbf{y}_i^r \right\|^2$'s we have the following guarantee*

$$\left\| \underline{\mathbf{z}}^{r,t_{\mathbf{z}}+1} - \hat{\underline{\mathbf{z}}}^r \right\| \leq \epsilon^c \tag{271}$$

*Define $\mathbf{w}_i^{r,0} := \left\| \mathbf{x}_i^{r,t_{\mathbf{x}}+1} - \mathbf{x}_i^r \right\|^2$, $\hat{\mathbf{w}}^r := \frac{1}{m} \sum_{i=1}^{m} \left\| \mathbf{x}_i^{r,t_{\mathbf{x}}+1} - \mathbf{x}_i^r \right\|^2$ and $t_{\mathbf{w}} = \left\lfloor \frac{c\log(\epsilon) - \log\left(4m^2(f(\mathbf{x}^0) - f^* + 2\mathcal{F})\right)}{\log(\sigma)} \right\rfloor$.*

*After $t_{\mathbf{w}} + 1$ rounds of the average consensus protocol on $\left\| \mathbf{x}_i^{r,t_{\mathbf{x}}+1} - \mathbf{x}_i^r \right\|^2$'s we have the following guarantee*

$$\left\| \underline{\mathbf{w}}^{r,t_{\mathbf{w}}+1} - \hat{\underline{\mathbf{w}}}^r \right\| \leq \epsilon^c \tag{272}$$

*Proof.* From the consensus update rule we know

$$\left\| \underline{\mathbf{z}}^{r,t+1} - \hat{\underline{\mathbf{z}}}^r \right\| \leq \left\| \mathbf{W} \left( \underline{\mathbf{z}}^{r,t} - \hat{\underline{\mathbf{z}}}^r \right) \right\| \leq \sigma \left\| \underline{\mathbf{z}}^{r,t} - \hat{\underline{\mathbf{z}}}^r \right\| \tag{273}$$

Thus we derive the following

$$\left\| \underline{\mathbf{z}}^{r,t+1} - \hat{\underline{\mathbf{z}}}^r \right\| \le \sigma^{t+1} \left\| \underline{\mathbf{z}}^{r,0} - \hat{\underline{\mathbf{z}}}^r \right\| \tag{274}$$

$$\le \sigma^{t+1} \sqrt{ \sum_{i=1}^{m} \left( \left\| \mathbf{y}_i^{r,t_\mathbf{y}+1} - \mathbf{y}_i^r \right\|^2 - \frac{1}{m} \sum_{j=1}^{m} \left\| \mathbf{y}_j^{r,t_\mathbf{y}+1} - \mathbf{y}_j^r \right\|^2 \right)^2 } \tag{275}$$

$$\le \sigma^{t+1} \sqrt{ \frac{16m^3}{\alpha^2} (f(\mathbf{x}^0) - f^* + 2\mathcal{F})^2 } \tag{276}$$

$$\le \sigma^{t+1} \frac{4m^2}{\alpha} (f(\mathbf{x}^0) - f^* + 2\mathcal{F}) \tag{277}$$

where the third inequality comes from Lemma 23. Solving for $t$ that guarantees $\sigma^{t+1} \frac{4m^2}{\alpha} (f(\mathbf{x}^0) - f^* + 2\mathcal{F}) \le \epsilon^c$ we get

$$\sigma^{t+1} \le \frac{\alpha \epsilon^c}{4m^2(f(\mathbf{x}^0) - f^* + 2\mathcal{F})} \tag{278}$$

$$(t+1)\log(\sigma) \le c\log(\epsilon) + \log\alpha - \log\left(4m^2(f(\mathbf{x}^0) - f^* + 2\mathcal{F})\right) \tag{279}$$

$$t \ge \frac{c\log(\epsilon) + \log\alpha - \log\left(4m^2(f(\mathbf{x}^0) - f^* + 2\mathcal{F})\right)}{\log(\sigma)} - 1 \tag{280}$$

Thus for $t = \left\lfloor \frac{c\log(\epsilon) + \log\alpha - \log\left(4m^2(f(\mathbf{x}^0) - f^* + 2\mathcal{F})\right)}{\log(\sigma)} \right\rfloor$ we have

$$\left\| \underline{\mathbf{z}}^{r,t+1} - \hat{\underline{\mathbf{z}}}^r \right\| \le \epsilon^c \tag{281}$$

Similarly from the consensus update rule we know

$$\left\| \underline{\mathbf{w}}^{r,t+1} - \hat{\underline{\mathbf{w}}}^r \right\| \le \left\| \mathbf{W} \left( \underline{\mathbf{w}}^{r,t} - \hat{\underline{\mathbf{w}}}^r \right) \right\| \le \sigma \left\| \underline{\mathbf{w}}^{r,t} - \hat{\underline{\mathbf{w}}}^r \right\| \tag{282}$$

Thus we derive the following

$$\left\| \underline{\mathbf{w}}^{r,t+1} - \hat{\underline{\mathbf{w}}}^r \right\| \le \sigma^{t+1} \left\| \underline{\mathbf{w}}^{r,0} - \hat{\underline{\mathbf{w}}}^r \right\| \tag{283}$$

$$\le \sigma^{t+1} \sqrt{ \sum_{i=1}^{m} \left( \left\| \mathbf{x}_i^{r,t_\mathbf{x}+1} - \mathbf{x}_i^r \right\|^2 - \frac{1}{m} \sum_{j=1}^{m} \left\| \mathbf{x}_j^{r,t_\mathbf{x}+1} - \mathbf{x}_j^r \right\|^2 \right)^2 } \tag{284}$$

$$\le \sigma^{t+1} \sqrt{ 16m^3 (f(\mathbf{x}^0) - f^* + 2\mathcal{F})^2 } \tag{285}$$

$$\le \sigma^{t+1} 4m^2 (f(\mathbf{x}^0) - f^* + 2\mathcal{F}) \tag{286}$$

where the third inequality comes from Lemma 23. Solving for $t$ that guarantees $\sigma^{t+1} 4m^2 (f(\mathbf{x}^0) - f^* + 2\mathcal{F}) \le \epsilon^c$ we get

$$\sigma^{t+1} \le \frac{\epsilon^c}{4m^2(f(\mathbf{x}^0) - f^* + 2\mathcal{F})} \tag{287}$$

$$(t+1)\log(\sigma) \le c\log(\epsilon) - \log\left(4m^2(f(\mathbf{x}^0) - f^* + 2\mathcal{F})\right) \tag{288}$$

$$t \ge \frac{c\log(\epsilon) - \log\left(4m^2(f(\mathbf{x}^0) - f^* + 2\mathcal{F})\right)}{\log(\sigma)} - 1 \tag{289}$$

Thus for $t = \left\lfloor \frac{c\log(\epsilon) - \log\left(4m^2(f(\mathbf{x}^0) - f^* + 2\mathcal{F})\right)}{\log(\sigma)} \right\rfloor$ we have

$$\left\| \underline{\mathbf{z}}^{r,t+1} - \hat{\underline{\mathbf{z}}}^r \right\| \le \epsilon^c \tag{290}$$

$\square$

**Corollary 8.** *Let* $\epsilon \leq \min\{1, \sqrt{\frac{m}{16\alpha}(f(\mathbf{x}^0) - f^* + 2\mathcal{F})}\}, c \geq 1$. *The total number of average consensus iterations to achieve sufficient accuracy captured in the following four bounds*

$$\left\|\underline{\mathbf{y}}^{r,t_{\mathbf{y}}} - \underline{\hat{\mathbf{y}}}^r\right\| \leq \epsilon^c \tag{291}$$

$$\left\|\underline{\mathbf{x}}^{r,t_{\mathbf{x}}} - \underline{\hat{\mathbf{x}}}^r\right\| \leq \epsilon^c \tag{292}$$

$$\left\|\underline{\mathbf{z}}^{r,t_{\mathbf{z}}} - \underline{\hat{\mathbf{z}}}^r\right\| \leq \epsilon^c \tag{293}$$

$$\left\|\underline{\mathbf{w}}^{r,t_{\mathbf{w}}} - \underline{\hat{\mathbf{w}}}^r\right\| \leq \epsilon^c \tag{294}$$

*is at most*

$$4\left(\frac{c\log(\frac{1}{\epsilon}) + \log(\frac{1}{\alpha}) + \log\left(4m^2(f(\mathbf{x}^0) - f^* + 2\mathcal{F})\right)}{\log(\frac{1}{\sigma})} + 1\right) \tag{295}$$

*Proof.* The total number of iterations is at most

$$t_{\mathbf{y}} + t_{\mathbf{x}} + t_{\mathbf{z}} + t_{\mathbf{w}} + 4 \leq 4t_{\mathbf{z}} + 4 \tag{296}$$

$$\leq 4\left(\frac{c\log(\epsilon) + \log\alpha - \log\left(4m^2(f(\mathbf{x}^0) - f^* + 2\mathcal{F})\right)}{\log(\sigma)} + 1\right) \tag{297}$$

$$\leq 4\left(\frac{c\log(\frac{1}{\epsilon}) + \log(\frac{1}{\alpha}) + \log\left(4m^2(f(\mathbf{x}^0) - f^* + 2\mathcal{F})\right)}{\log(\frac{1}{\sigma})} + 1\right) \tag{298}$$

$\square$

The next lemma shows how far off is the square of the estimated difference $\left\|\underline{\mathbf{y}}^{r,t_{\mathbf{y}}+1} - \underline{\mathbf{y}}^r\right\|^2$ (and respectively $\left\|\underline{\mathbf{x}}^{r,t_{\mathbf{x}}+1} - \underline{\mathbf{x}}^r\right\|^2$) from the square of the true difference $\left\|\underline{\hat{\mathbf{y}}}^r - \underline{\mathbf{y}}^r\right\|^2$ (and respectively $\left\|\underline{\hat{\mathbf{x}}}^r - \underline{\mathbf{x}}^r\right\|^2$). Similar result for the square of the average gradient estimate is also provided.

**Lemma 25.** *Consider any iterates* $\underline{\mathbf{x}}^r, \mathbf{y}^r$ *following Gradient Tracking Update with* $\eta_1, \alpha$ *that satisfy conditions* (41) - (44). *Also assume that the potential function decreases between consecutive first phases. Let* $\underline{\mathbf{y}}^{r,t_{\mathbf{y}}+1}$ *and* $\underline{\mathbf{x}}^{r,t_{\mathbf{x}}+1}$ *as defined in Lemma 22. The following bounds hold:*

$$\frac{1}{m}\left\|\underline{\hat{\mathbf{y}}}^r - \underline{\mathbf{y}}^r\right\|^2 + \frac{1}{m}\epsilon^{2c} + \frac{2}{m}\epsilon^c\left\|\underline{\hat{\mathbf{y}}}^r - \underline{\mathbf{y}}^r\right\| \geq \frac{1}{m}\left\|\underline{\mathbf{y}}^{r,t_{\mathbf{y}}+1} - \underline{\mathbf{y}}^r\right\|^2 \geq \frac{1}{m}\left\|\underline{\hat{\mathbf{y}}}^r - \underline{\mathbf{y}}^r\right\|^2 - \frac{2}{m}\epsilon^c\left\|\underline{\hat{\mathbf{y}}}^r - \underline{\mathbf{y}}^r\right\| \tag{299}$$

$$\frac{1}{m}\left\|\underline{\hat{\mathbf{x}}}^r - \underline{\mathbf{x}}^r\right\|^2 + \frac{1}{m}\epsilon^{2c} + \frac{2}{m}\epsilon^c\left\|\underline{\hat{\mathbf{x}}}^r - \underline{\mathbf{x}}^r\right\| \geq \frac{1}{m}\left\|\underline{\mathbf{x}}^{r,t_{\mathbf{x}}+1} - \underline{\mathbf{x}}^r\right\|^2 \geq \frac{1}{m}\left\|\underline{\hat{\mathbf{x}}}^r - \underline{\mathbf{x}}^r\right\|^2 - \frac{2}{m}\epsilon^c\left\|\underline{\hat{\mathbf{x}}}^r - \underline{\mathbf{x}}^r\right\| \tag{300}$$

$$\forall i \quad \|\hat{\mathbf{y}}^r\|^2 + \epsilon^{2c} + 2\epsilon^c\|\hat{\mathbf{y}}^r\| \geq \left\|\mathbf{y}_i^{r,t_{\mathbf{y}}+1}\right\|^2 \geq \|\hat{\mathbf{y}}^r\|^2 - 2\epsilon^c\|\hat{\mathbf{y}}^r\| \tag{301}$$

*Proof.*

$$\frac{1}{m}\left\|\underline{\mathbf{y}}^{r,t_{\mathbf{y}}+1} - \underline{\mathbf{y}}^r\right\|^2 = \frac{1}{m}\left\|\underline{\mathbf{y}}^{r,t_{\mathbf{y}}+1} - \underline{\hat{\mathbf{y}}}^r + \underline{\hat{\mathbf{y}}}^r - \underline{\mathbf{y}}^r\right\|^2 \tag{302}$$

$$= \frac{1}{m}\left(\left\|\underline{\mathbf{y}}^{r,t_{\mathbf{y}}+1} - \underline{\hat{\mathbf{y}}}^r\right\|^2 + \left\|\underline{\hat{\mathbf{y}}}^r - \underline{\mathbf{y}}^r\right\|^2 + 2\langle\underline{\mathbf{y}}^{r,t_{\mathbf{y}}+1} - \underline{\hat{\mathbf{y}}}^r, \underline{\hat{\mathbf{y}}}^r - \underline{\mathbf{y}}^r\rangle\right) \tag{303}$$

and from here we can derive both the upper and the lower bound

$$\frac{1}{m}\left\|\underline{\mathbf{y}}^{r,t_{\mathbf{y}}+1} - \underline{\mathbf{y}}^r\right\|^2 \geq \frac{1}{m}\left\|\underline{\hat{\mathbf{y}}}^r - \underline{\mathbf{y}}^r\right\|^2 - \frac{2}{m}\left\|\underline{\mathbf{y}}^{r,t_{\mathbf{y}}+1} - \underline{\hat{\mathbf{y}}}^r\right\|\left\|\underline{\hat{\mathbf{y}}}^r - \underline{\mathbf{y}}^r\right\| \tag{304}$$

$$\geq \frac{1}{m}\left\|\underline{\hat{\mathbf{y}}}^r - \underline{\mathbf{y}}^r\right\|^2 - \frac{2}{m}\epsilon^c\left\|\underline{\hat{\mathbf{y}}}^r - \underline{\mathbf{y}}^r\right\| \tag{305}$$

where the last inequality is due to Lemma 22. Also

$$\frac{1}{m} \left\| \underline{\mathbf{y}}^{r,t_{\mathbf{y}}+1} - \underline{\mathbf{y}}^r \right\|^2 \leq \frac{1}{m} \left( \left\| \underline{\mathbf{y}}^{r,t_{\mathbf{y}}+1} - \underline{\hat{\mathbf{y}}}^r \right\|^2 + \left\| \underline{\hat{\mathbf{y}}}^r - \underline{\mathbf{y}}^r \right\|^2 + 2 \left\| \underline{\mathbf{y}}^{r,t_{\mathbf{y}}+1} - \underline{\hat{\mathbf{y}}}^r \right\| \left\| \underline{\hat{\mathbf{y}}}^r - \underline{\mathbf{y}}^r \right\| \right) \tag{306}$$

$$\leq \frac{1}{m} \left\| \underline{\hat{\mathbf{y}}}^r - \underline{\mathbf{y}}^r \right\|^2 + \frac{1}{m} \epsilon^{2c} + \frac{2}{m} \epsilon^c \left\| \underline{\hat{\mathbf{y}}}^r - \underline{\mathbf{y}}^r \right\| \tag{307}$$

where again the second inequality comes from Lemma 22.

The proof deriving the bounds for $\frac{1}{m} \left\| \underline{\mathbf{x}}^{r,t_{\mathbf{x}}+1} - \underline{\hat{\mathbf{x}}}^r \right\|^2$ is identical. For the third bound we work as follows

$$\left\| \mathbf{y}_i^{r,t+1} \right\|^2 = \left\| \mathbf{y}_i^{r,t+1} - \hat{\mathbf{y}}^r + \hat{\mathbf{y}}^r \right\|^2 = \left\| \mathbf{y}_i^{r,t+1} - \hat{\mathbf{y}}^r \right\|^2 + \|\hat{\mathbf{y}}^r\|^2 + 2\langle \mathbf{y}_i^{r,t+1} - \hat{\mathbf{y}}^r, \hat{\mathbf{y}}^r \rangle \tag{308}$$

and thus we derive the bounds

$$\left\| \mathbf{y}_i^{r,t+1} \right\|^2 \geq \|\hat{\mathbf{y}}^r\|^2 - 2 \left\| \mathbf{y}_i^{r,t+1} - \hat{\mathbf{y}}^r \right\| \|\hat{\mathbf{y}}^r\| \tag{309}$$

$$\geq \|\hat{\mathbf{y}}^r\|^2 - 2\epsilon^c \|\hat{\mathbf{y}}^r\| \tag{310}$$

$$\tag{311}$$

the last inequality follows from Corollary 7. Also

$$\left\| \mathbf{y}_i^{r,t+1} \right\|^2 \leq \|\hat{\mathbf{y}}^r\|^2 + \left\| \mathbf{y}_i^{r,t+1} - \hat{\mathbf{y}}^r \right\|^2 + 2 \left\| \mathbf{y}_i^{r,t+1} - \hat{\mathbf{y}}^r \right\| \|\hat{\mathbf{y}}^r\| \tag{312}$$

$$\leq \|\hat{\mathbf{y}}^r\|^2 + \epsilon^{2c} + 2\epsilon^c \|\hat{\mathbf{y}}^r\| \tag{313}$$

$\square$

The result derived in the following lemma is utilized as an intermediate step to towards proving second objective.

**Lemma 26.** *Consider any iterates $\underline{\mathbf{x}}^r, \underline{\mathbf{y}}^r$ following Gradient Tracking Update with $\eta_1, \alpha$ that satisfy conditions* (41) - (44). *Also assume that the potential function decreases between consecutive first phases. Let $\mathbf{y}^{r,t_{\mathbf{y}}+1}$ and $\mathbf{x}^{r,t_{\mathbf{x}}+1}$ as defined in Lemma 22. Also let $\epsilon \leq \min\{\frac{1}{8}, \sqrt{\frac{m}{16\alpha}(f(\mathbf{x}^0) - f^* + 2\mathcal{F})}\}$ and $c \geq 2$. Assume that*

$$\|\hat{\mathbf{y}}^r\|^2 + \frac{1}{m} \|\underline{\hat{\mathbf{x}}}^r - \underline{\mathbf{x}}^r\|^2 \leq \frac{\epsilon^2}{4} \tag{314}$$

*then*

$$\left\| \mathbf{y}_i^{r,t_{\mathbf{y}}+1} \right\|^2 + \frac{1}{m} \left\| \underline{\mathbf{x}}^{r,t_{\mathbf{x}}+1} - \underline{\mathbf{x}}^r \right\|^2 \leq \frac{\epsilon^2}{3} \tag{315}$$

*Further assume that*

$$\|\hat{\mathbf{y}}^r\|^2 + \frac{1}{m} \|\underline{\hat{\mathbf{x}}}^r - \underline{\mathbf{x}}^r\|^2 > \epsilon^2 \tag{316}$$

*then*

$$\left\| \mathbf{y}_i^{r,t_{\mathbf{y}}+1} \right\|^2 + \frac{1}{m} \left\| \underline{\mathbf{x}}^{r,t_{\mathbf{x}}+1} - \underline{\mathbf{x}}^r \right\|^2 > \frac{7}{8}\epsilon^2 \tag{317}$$

*Proof.* For the first part of the proof we utilize the upper bounds from Lemma 25:

$$\left\| \mathbf{y}_i^{r,t_{\mathbf{y}}+1} \right\|^2 + \frac{1}{m} \left\| \underline{\mathbf{x}}^{r,t_{\mathbf{x}}+1} - \underline{\mathbf{x}}^r \right\|^2 \tag{318}$$

$$\leq \|\hat{\mathbf{y}}^r\|^2 + \frac{1}{m} \|\underline{\hat{\mathbf{x}}}^r - \underline{\mathbf{x}}^r\|^2 + \left(1 + \frac{1}{m}\right) \epsilon^{2c} + 2\epsilon^c \left( \|\hat{\mathbf{y}}^r\|^2 + \frac{1}{m} \|\underline{\hat{\mathbf{x}}}^r - \underline{\mathbf{x}}^r\|^2 \right) \tag{319}$$

$$\leq \frac{\epsilon^2}{4} + \left(1 + \frac{2}{m}\right) \epsilon^{2c} + 2\epsilon^c \frac{\epsilon^2}{4} \tag{320}$$

$$\leq \epsilon^2 \left( \frac{1}{4} + \left(1 + \frac{2}{m}\right) \epsilon^{2c-2} + \frac{1}{2}\epsilon^c \right) \tag{321}$$

$$\leq \frac{\epsilon^2}{3} \tag{322}$$

where the last inequality holds for $\epsilon \leq \frac{1}{6}$ and $c \geq 2$.

For the second part of the proof we utilize the lower bounds from Lemma 25:

$$\left\|\mathbf{y}_i^{r,t_\mathbf{y}+1}\right\|^2 + \frac{1}{m}\left\|\underline{\mathbf{x}}^{r,t_\mathbf{x}+1} - \underline{\mathbf{x}}^r\right\|^2 \tag{323}$$

$$\geq \|\hat{\mathbf{y}}^r\|^2 + \frac{1}{m}\|\underline{\hat{\mathbf{x}}}^r - \underline{\mathbf{x}}^r\|^2 - 2\epsilon^c\left(\|\hat{\mathbf{y}}^r\|^2 + \frac{1}{m}\|\underline{\hat{\mathbf{x}}}^r - \underline{\mathbf{x}}^r\|^2\right) \tag{324}$$

$$> \epsilon^2 - 2\epsilon^{c+2} \tag{325}$$

$$\geq \frac{7}{8}\epsilon^2 \tag{326}$$

where the last inequality holds for $\epsilon \leq \frac{1}{6}$ and $c \geq 2$.

$\square$

The following lemma states that if $r$ is a good iteration then for each node the estimation after running the consensus protocol is at most $\frac{\epsilon^2}{2}$. On the other hand if $r$ is an iterate with $\|\hat{\mathbf{y}}^r\|^2 + \frac{1}{m}\|\underline{\hat{\mathbf{x}}}^r - \underline{\mathbf{x}}^r\|^2 + \frac{1}{m}\|\underline{\hat{\mathbf{y}}}^r - \underline{\mathbf{y}}^r\|^2 > \epsilon^2$ then each node has an estimation of value at least $\frac{3}{4}\epsilon^2$.

**Lemma 27.** *Consider any iterates $\mathbf{x}^r, \mathbf{y}^r$ following Gradient Tracking Update with $\eta_1, \alpha$ that satisfy conditions (41) - (44). Also assume that the potential function decreases between consecutive first phases. Let $\underline{\mathbf{y}}^{r,t_\mathbf{y}+1}$ and $\underline{\mathbf{x}}^{r,t_\mathbf{x}+1}$ , $\underline{\mathbf{z}}^{r,t_\mathbf{z}+1}$ and $\underline{\mathbf{w}}^{r,t_\mathbf{w}+1}$ as defined in Lemma 22 and Lemma 24. Also let $\epsilon \leq \min\{\frac{1}{8}, \sqrt{\frac{m}{16\alpha}(f(\mathbf{x}^0) - f^* + 2\mathcal{F})}\}$ and $c \geq 3$. If it holds that*

$$\|\hat{\mathbf{y}}^r\|^2 + \frac{1}{m}\|\underline{\hat{\mathbf{x}}}^r - \underline{\mathbf{x}}^r\|^2 \leq \frac{\epsilon^2}{4} \tag{327}$$

*then*

$$\left\|\mathbf{y}_i^{r,t_\mathbf{y}+1}\right\|^2 + \mathbf{w}_i^{r,t_\mathbf{w}+1} \leq \frac{\epsilon^2}{2} \quad \forall i \tag{328}$$

*Further if it holds that*

$$\|\hat{\mathbf{y}}^r\|^2 + \frac{1}{m}\|\underline{\hat{\mathbf{x}}}^r - \underline{\mathbf{x}}^r\|^2 > \epsilon^2 \tag{329}$$

*then*

$$\left\|\mathbf{y}_i^{r,t_\mathbf{y}+1}\right\|^2 + \mathbf{w}_i^{r,t_\mathbf{w}+1} > \frac{3}{4}\epsilon^2 \tag{330}$$

*Proof.* From Lemma 24 we know the following

$$\left\|\underline{\mathbf{w}}^{r,t_\mathbf{w}+1} - \underline{\hat{\mathbf{w}}}^r\right\| \leq \epsilon^c \tag{331}$$

$$|\mathbf{w}_i^{r,t_\mathbf{w}+1} - \hat{\mathbf{w}}^r| \leq \epsilon^c \tag{332}$$

$$\left|\mathbf{w}_i^{r,t_\mathbf{w}+1} - \frac{1}{m}\sum_{i=1}^m\left\|\mathbf{x}_i^{r,t_\mathbf{x}+1} - \mathbf{x}_i^r\right\|^2\right| \leq \epsilon^c \tag{333}$$

$$\left|\mathbf{w}_i^{r,t_\mathbf{w}+1} - \frac{1}{m}\left\|\underline{\mathbf{x}}^{r,t_\mathbf{x}+1} - \underline{\mathbf{x}}^r\right\|^2\right| \leq \epsilon^c \tag{334}$$

$$\tag{335}$$

And thus we have

$$\frac{1}{m}\left\|\underline{\mathbf{x}}^{r,t_\mathbf{x}+1} - \underline{\mathbf{x}}^r\right\|^2 - \epsilon^c \leq \mathbf{w}_i^{r,t_\mathbf{w}+1} \leq \frac{1}{m}\left\|\underline{\mathbf{x}}^{r,t_\mathbf{x}+1} - \underline{\mathbf{x}}^r\right\|^2 + \epsilon^c \tag{336}$$

Utilizing the above bounds and Lemma 26 we can show the first claim

$$\left\|\mathbf{y}_i^{r,t_\mathbf{y}+1}\right\|^2 + \mathbf{w}_i^{r,t_\mathbf{w}+1} \leq \left\|\mathbf{y}_i^{r,t_\mathbf{y}+1}\right\|^2 + \frac{1}{m}\left\|\underline{\mathbf{x}}^{r,t_\mathbf{x}+1} - \underline{\mathbf{x}}^r\right\|^2 + \epsilon^c \tag{337}$$

$$\leq \frac{\epsilon^2}{3} + \epsilon^c \tag{338}$$

$$\leq \frac{\epsilon^2}{2} \tag{339}$$

where the last inequality holds for $\epsilon \leq \frac{1}{6}$ and $c \geq 3$. The second claim is derived along the same lines:

$$\left\|\mathbf{y}_i^{r,t_\mathbf{y}+1}\right\|^2 + \mathbf{w}_i^{r,t_\mathbf{w}+1} \geq \left\|\mathbf{y}_i^{r,t_\mathbf{y}+1}\right\|^2 + \frac{1}{m}\left\|\underline{\mathbf{x}}^{r,t_\mathbf{x}+1} - \underline{\mathbf{x}}^r\right\|^2 - \epsilon^c \tag{340}$$

$$> \frac{7}{8}\epsilon^2 - \epsilon^c \tag{341}$$

$$\geq \frac{3}{4}\epsilon^2 \tag{342}$$

where the last inequality holds for $\epsilon \leq \frac{1}{8}$ and $c \geq 3$. □

The next lemma shows that after a small number of iterations all the nodes can coordinate to either approve or disapprove iteration $r$.

**Lemma 28.** *Consider any first phase iterates $\underline{\mathbf{x}}^r, \mathbf{y}^r$ following Gradient Tracking Update with $\eta_1, \alpha$ that satisfy conditions (41) - (44). Also assume that the potential function decreases between consecutive first phases. Let $\mathbf{y}^{r,t_\mathbf{y}+1}$, $\underline{\mathbf{x}}^{r,t_\mathbf{x}+1}$ and $\underline{\mathbf{w}}^{r,t_\mathbf{w}+1}$ as defined in Lemma 22 and Lemma 24. Let $\epsilon \leq \min\{\frac{1}{8}, \sqrt{\frac{m}{16\alpha}(f(\mathbf{x}^0) - f^* + 2\mathcal{F})}\}$ and $c \geq 3$.*

*Define $ind_i^{r,0} := \mathbb{1}_{\left\{\left\|\mathbf{y}_i^{r,t_\mathbf{y}+1}\right\|^2 + \mathbf{w}_i^{r,t_\mathbf{w}+1} \leq \frac{\epsilon^2}{2}\right\}}$ and also $i\hat{n}d^r := \frac{1}{m}\sum_{i=1}^m ind_i^{r,0}$*

*Also define $t_{\mathbf{ind}} := \left\lfloor \frac{\log(2m^{\frac{3}{2}})}{\log(\frac{1}{\sigma})} \right\rfloor$*

*If we run the average consensus protocol on $ind_i^{r,0}$'s for $t_{\mathbf{ind}}$ iterations we are going to achieve the following bound*

$$\left\|\mathbf{ind}^{r,t_{\mathbf{ind}}+1} - i\hat{\mathbf{n}}\mathbf{d}^r\right\| \leq \frac{1}{2m} \tag{343}$$

*Proof.* From the update of the average consensus protocol we have

$$\left\|\mathbf{ind}^{r,t+1} - i\hat{\mathbf{n}}\mathbf{d}^r\right\| \leq \left\|\mathbf{W}\left(\mathbf{ind}^{r,t} - i\hat{\mathbf{n}}\mathbf{d}^r\right)\right\| \leq \sigma\left\|\mathbf{ind}^{r,t} - i\hat{\mathbf{n}}\mathbf{d}^r\right\| \tag{344}$$

$$\tag{345}$$

Thus we have

$$\left\|\mathbf{ind}^{r,t+1} - i\hat{\mathbf{n}}\mathbf{d}^r\right\| \leq \sigma^{t+1}\left\|\mathbf{ind}^{r,0} - i\hat{\mathbf{n}}\mathbf{d}^r\right\| \tag{346}$$

Choosing $t$ such that $\sigma^{t+1}\left\|\mathbf{ind}^{r,0} - i\hat{\mathbf{n}}\mathbf{d}^r\right\| \leq \frac{1}{2m}$ and since $\left\|\mathbf{ind}^{r,0} - i\hat{\mathbf{n}}\mathbf{d}^r\right\| \leq \sqrt{m}$ we get

$$\sigma^{t+1} \leq \frac{1}{2m^{\frac{3}{2}}} \tag{347}$$

$$(t+1)\log(\sigma) \leq -\log(2m^{\frac{3}{2}}) \tag{348}$$

$$t \leq \frac{\log(2m^{\frac{3}{2}})}{\log(\frac{1}{\sigma})} - 1 \tag{349}$$

Thus choosing $t_{\mathbf{ind}} = \left\lfloor \frac{\log(2m^{\frac{3}{2}})}{\log(\frac{1}{\sigma})} \right\rfloor$ we derive the result. □

**Corollary 9.** *After running the average consensus protocol on $\mathbf{y}_i^r$'s, $\mathbf{x}_i^r$'s, $\left\|\mathbf{x}_i^{r,t_\mathbf{x}+1} - \mathbf{x}_i^r\right\|^2$'s and $ind_i^r$'s each node approves iteration $r$ if $1 - \frac{1}{2m} \leq ind_i^{r,t_{\mathbf{ind}}+1}$. The total number of iterations are at most $t_{tot} = 4\left(\frac{c\log(\frac{1}{\epsilon}) + \log(\frac{1}{\alpha}) + \log\left(4m^2(f(\mathbf{x}^0) - f^* + 2\mathcal{F})\right)}{\log(\frac{1}{\sigma})} + 1\right)$. Further if*

$$\|\hat{\mathbf{y}}^r\|^2 + \frac{1}{m}\|\hat{\underline{\mathbf{x}}}^r - \underline{\mathbf{x}}^r\|^2 \leq \frac{\epsilon^2}{4} \tag{350}$$

*then all nodes will approve whereas if*

$$\|\hat{\mathbf{y}}^r\|^2 + \frac{1}{m}\|\hat{\underline{\mathbf{x}}}^r - \underline{\mathbf{x}}^r\|^2 \geq \epsilon^2 \tag{351}$$

*then all nodes will disapprove.*

*Proof.* The second part of the corollary is immediate from Lemma 28. For the number of iterations we have the following:

$$t_{tot} \leq t_{\mathbf{y}} + t_{\mathbf{x}} + t_{\mathbf{w}} + 3 + t_{\mathbf{ind}} + 1 \tag{352}$$

$$\leq 3t_{\mathbf{z}} + 3 + t_{\mathbf{ind}} + 1 \tag{353}$$

$$\leq 3\left(\frac{c\log(\frac{1}{\epsilon}) + \log(\frac{1}{\alpha}) + \log\left(4m^2(f(\mathbf{x}^0) - f^* + 2\mathcal{F})\right)}{\log(\frac{1}{\sigma})} + 1\right) + \frac{\log(2m^{\frac{3}{2}})}{\log(\frac{1}{\sigma})} + 1 \tag{354}$$

$$\leq 4\left(\frac{c\log(\frac{1}{\epsilon}) + \log(\frac{1}{\alpha}) + \log\left(4m^2(f(\mathbf{x}^0) - f^* + 2\mathcal{F})\right)}{\log(\frac{1}{\sigma})} + 1\right) \tag{355}$$

$$\square$$

## 11.3 Tracking the Potential Function

Finally, working towards our third objective recall that $H(\underline{\mathbf{x}}, \underline{\mathbf{y}}) = f(\hat{\mathbf{x}}) + \frac{1}{m}\left\|\underline{\mathbf{x}}^r - \hat{\underline{\mathbf{x}}}^r\right\|^2 + \frac{\alpha}{m}\left\|\underline{\mathbf{y}}^r - \hat{\underline{\mathbf{y}}}^r\right\|^2$. We start by utilizing Corollary 8 and thus for sufficiently small $\tilde{\epsilon}$ after $4\left(\frac{c\log(\frac{1}{\tilde{\epsilon}}) + \log(\frac{1}{\alpha}) + \log\left(4m^2(f(\mathbf{x}^0) - f^* + 2\mathcal{F})\right)}{\log(\frac{1}{\sigma})} + 1\right)$ communication rounds we achieve the following accuracy bounds for some iteration $r$:

$$\left\|\underline{\mathbf{y}}^{r,t_{\mathbf{y}}} - \hat{\underline{\mathbf{y}}}^r\right\| \leq \tilde{\epsilon}^c \tag{356}$$

$$\left\|\underline{\mathbf{x}}^{r,t_{\mathbf{x}}} - \hat{\underline{\mathbf{x}}}^r\right\| \leq \tilde{\epsilon}^c \tag{357}$$

$$\left\|\underline{\mathbf{z}}^{r,t_{\mathbf{z}}} - \hat{\underline{\mathbf{z}}}^r\right\| \leq \tilde{\epsilon}^c \tag{358}$$

$$\left\|\underline{\mathbf{w}}^{r,t_{\mathbf{w}}} - \hat{\underline{\mathbf{w}}}^r\right\| \leq \tilde{\epsilon}^c \tag{359}$$

Further choosing a sufficiently large $\tilde{c}$ and running the consensus protocol for $\mathbf{x}_i^r$'s for $\left(\frac{\tilde{c}\log(\frac{1}{\tilde{\epsilon}}) + \log(\frac{1}{\alpha}) + \log\left(4m^2(f(\mathbf{x}^0) - f^* + 2\mathcal{F})\right)}{\log(\frac{1}{\sigma})} + 1\right)$ rounds guarantees sufficient accuracy on the function value of the average iterate $\frac{1}{m}\sum_{i=1}^m f_i(\hat{\mathbf{x}})$.

**Lemma 29.** *Consider any iterates $\underline{\mathbf{x}}^r, \mathbf{y}^r$ following Gradient Tracking Update with $\eta_1, \alpha$ that satisfy conditions (41) - (44). Also assume that the potential function decreases between consecutive first phases. Consider a sufficiently large $\tilde{c}$ that guarantees $\max_i\left\|f_i(\mathbf{x}_i^{r,t_{\mathbf{x}}+1}) - f_i(\hat{\mathbf{x}}^r)\right\| \leq \frac{\tilde{\epsilon}^c}{2m}$ after running the consensus protocol on $\mathbf{x}_i^r$'s for $t_{\mathbf{x}} = \left(\frac{\tilde{c}\log(\frac{1}{\tilde{\epsilon}}) + \log(\frac{1}{\alpha}) + \log\left(4m^2(f(\mathbf{x}^0) - f^* + 2\mathcal{F})\right)}{\log(\frac{1}{\sigma})} + 1\right)$ rounds. Define $g_i^{r,0} := f_i(\mathbf{x}_i^{r,t_{\mathbf{x}}+1})$ and $\hat{g}^r := \frac{1}{m}\sum_{i=1}^m f_i(\mathbf{x}_i^{r,t_{\mathbf{x}}+1})$, let $t_{\mathbf{g}} = \left\lfloor\frac{c\log(\tilde{\epsilon}) - \log\left(2\left\|\mathbf{g}^{r,0} - \hat{\mathbf{g}}^r\right\|\right)}{\log\sigma}\right\rfloor$ and denote the true target function value by $\hat{g}_{tr}^r := \frac{1}{m}\sum_{i=1}^m f_i(\hat{\mathbf{x}}^r)$.*

*Then the following bound holds $\left\|\underline{\mathbf{g}}^{r,t_{\mathbf{g}}+1} - \hat{\underline{\mathbf{g}}}_{tr}^r\right\| \leq \tilde{\epsilon}^c$*

*Proof.* From the update of the consensus protocol we have $\left\|\underline{\mathbf{g}}^{r,t+1} - \hat{\underline{\mathbf{g}}}^r\right\| \leq \sigma^{t+1}\left\|\underline{\mathbf{g}}^{r,0} - \hat{\underline{\mathbf{g}}}^r\right\|$. Thus we solve for $t$ such that

$$\sigma^{t+1}\left\|\underline{\mathbf{g}}^{r,0} - \hat{\underline{\mathbf{g}}}^r\right\| \leq \frac{\tilde{\epsilon}^c}{2} \tag{360}$$

$$t \geq \frac{c\log(\tilde{\epsilon}) - \log\left(2\left\|\underline{\mathbf{g}}^{r,0} - \hat{\underline{\mathbf{g}}}^r\right\|\right)}{\log\sigma} - 1 \tag{361}$$

Thus for $t_{\mathbf{g}} = \left\lceil \frac{c\log(\tilde{\epsilon}) - \log\left(2\left\|\mathbf{g}^{r,0} - \hat{\mathbf{g}}^r\right\|\right)}{\log\sigma} \right\rceil$ we have $\left\|\mathbf{g}^{r,t_{\mathbf{g}}+1} - \hat{\mathbf{g}}^r\right\| \le \frac{\tilde{\epsilon}^c}{2}$.

Using the assumptions of the lemma we can also show that the estimation of the nodes is not far from the true function value.

$$\max_i \left\| f_i(\mathbf{x}_i^{r,t_{\mathbf{x}}+1}) - f_i(\hat{\mathbf{x}}^r) \right\| \le \frac{\tilde{\epsilon}^c}{2m} \tag{362}$$

$$\frac{1}{m}\sum_{i=1}^m \left\| f_i(\mathbf{x}_i^{r,t_{\mathbf{x}}+1}) - f_i(\hat{\mathbf{x}}^r) \right\| \le \frac{\tilde{\epsilon}^c}{2m} \tag{363}$$

$$\left\| \frac{1}{m}\sum_{i=1}^m f_i(\mathbf{x}_i^{r,t_{\mathbf{x}}+1}) - \frac{1}{m}\sum_{i=1}^m f_i(\hat{\mathbf{x}}^r) \right\| \le \frac{\tilde{\epsilon}^c}{2m} \tag{364}$$

$$\left\| \hat{\mathbf{g}}^r - \hat{\underline{\mathbf{g}}}^r_{tr} \right\| \le \frac{\tilde{\epsilon}^c}{2} \tag{365}$$

which implies that $\left\| \mathbf{g}^{r,t_{\mathbf{g}}+1} - \hat{\underline{\mathbf{g}}}^r_{tr} \right\| \le \tilde{\epsilon}^c$ □

**Corollary 10.** *Consider the assumption of Lemma 29 hold. Then after* $4\frac{\tilde{c}\log(\frac{1}{\tilde{\epsilon}}) + \log(\frac{1}{\alpha}) + \log\left(4m^2(f(\mathbf{x}^0) - f^* + 2\mathcal{F})\right)}{\log(\frac{1}{\sigma})} + 4 + \frac{c\log(\tilde{\epsilon}) - \log\left(2\left\|\mathbf{g}^{r,0} - \hat{\mathbf{g}}^r\right\|\right)}{\log\sigma}$ *rounds of consensus protocol we achieve the following accuracy:*

$$\left\| \mathbf{y}^{r,t_{\mathbf{y}}} - \hat{\mathbf{y}}^r \right\| \le \tilde{\epsilon}^c \tag{366}$$

$$\left\| \mathbf{x}^{r,t_{\mathbf{x}}} - \hat{\mathbf{x}}^r \right\| \le \tilde{\epsilon}^c \tag{367}$$

$$\left\| \mathbf{z}^{r,t_{\mathbf{z}}} - \hat{\mathbf{z}}^r \right\| \le \tilde{\epsilon}^c \tag{368}$$

$$\left\| \mathbf{w}^{r,t_{\mathbf{w}}} - \hat{\mathbf{w}}^r \right\| \le \tilde{\epsilon}^c \tag{369}$$

$$\left\| g_i^{r,t_{\mathbf{g}}} - \frac{1}{m}\sum_{i=1}^m f_i(\hat{\mathbf{x}}) \right\| \le \tilde{\epsilon}^c, \forall i \tag{370}$$

Similarly to section 11.2 utilizing Lemma 25 and equation (370) we get the following bounds :

**Corollary 11.** *Consider the assumptions of Lemma 29 hold. Then the following bounds also hold*

$$\frac{1}{m}\left\| \hat{\mathbf{y}}^r - \mathbf{y}^r \right\|^2 + \frac{1}{m}\tilde{\epsilon}^{2c} + \frac{2}{m}\tilde{\epsilon}^c \left\| \hat{\mathbf{y}}^r - \mathbf{y}^r \right\| \ge \frac{1}{m}\left\| \mathbf{y}^{r,t_{\mathbf{y}}+1} - \mathbf{y}^r \right\|^2 \ge \frac{1}{m}\left\| \hat{\mathbf{y}}^r - \mathbf{y}^r \right\|^2 - \frac{2}{m}\tilde{\epsilon}^c \left\| \hat{\mathbf{y}}^r - \mathbf{y}^r \right\| \tag{371}$$

$$\frac{1}{m}\left\| \hat{\mathbf{x}}^r - \mathbf{x}^r \right\|^2 + \frac{1}{m}\tilde{\epsilon}^{2c} + \frac{2}{m}\tilde{\epsilon}^c \left\| \hat{\mathbf{x}}^r - \mathbf{x}^r \right\| \ge \frac{1}{m}\left\| \mathbf{x}^{r,t_{\mathbf{x}}+1} - \mathbf{x}^r \right\|^2 \ge \frac{1}{m}\left\| \hat{\mathbf{x}}^r - \mathbf{x}^r \right\|^2 - \frac{2}{m}\tilde{\epsilon}^c \left\| \hat{\mathbf{x}}^r - \mathbf{x}^r \right\| \tag{372}$$

$$\frac{1}{m}\sum_{i=1}^m f_i(\hat{\mathbf{x}}) + \tilde{\epsilon}^c \ge g_i^{r,t_{\mathbf{g}}} \ge \frac{1}{m}\sum_{i=1}^m f_i(\hat{\mathbf{x}}) - \tilde{\epsilon}^c \tag{373}$$

The next lemma is used as an intermediate step in order to derive our final result.

**Lemma 30.** *Consider the assumptions of Lemma 29 hold. Further let* $\tilde{\epsilon} \le \min\left\{\frac{1}{8}, \left(4\sqrt{f(\hat{\mathbf{x}}^0) - f^* + 2\mathcal{F}}\right)^{\frac{c}{4}}\right\}$ *and* $c \ge 4$. *Then we can prove the following bounds:*

$$H(\mathbf{x}^r, \mathbf{y}^r) + \tilde{\epsilon}^{\frac{c}{2}} \ge g_i^{r,t_{\mathbf{g}}} + \frac{1}{m}\left\| \mathbf{x}^{r,t_{\mathbf{x}}+1} - \mathbf{x}^r \right\|^2 + \frac{\alpha}{m}\left\| \mathbf{y}^{r,t_{\mathbf{y}}+1} - \mathbf{y}^r \right\|^2 \ge H(\mathbf{x}^r, \mathbf{y}^r) - \tilde{\epsilon}^{\frac{c}{2}} \tag{374}$$

*Proof.* Towards proving the upper bound we utilize the results presented in Corollary 10.

$$g_i^{r,t_{\mathbf{g}}} + \frac{1}{m} \left\| \mathbf{x}^{r,t_{\mathbf{x}}+1} - \underline{\mathbf{x}}^r \right\|^2 + \frac{\alpha}{m} \left\| \mathbf{y}^{r,t_{\mathbf{y}}+1} - \underline{\mathbf{y}}^r \right\|^2$$

$$\leq \frac{1}{m} \sum_{i=1}^{m} f_i(\hat{\mathbf{x}}) + \tilde{\epsilon}^c + \frac{1}{m} \left\| \underline{\hat{\mathbf{x}}}^r - \underline{\mathbf{x}}^r \right\|^2 + \frac{1}{m} \tilde{\epsilon}^{2c} + \frac{2}{m} \tilde{\epsilon}^c \left\| \underline{\hat{\mathbf{x}}}^r - \underline{\mathbf{x}}^r \right\|$$

$$+ \frac{\alpha}{m} \left\| \underline{\hat{\mathbf{y}}}^r - \underline{\mathbf{y}}^r \right\|^2 + \frac{\alpha}{m} \tilde{\epsilon}^{2c} + \frac{2\alpha}{m} \tilde{\epsilon}^c \left\| \underline{\hat{\mathbf{y}}}^r - \underline{\mathbf{y}}^r \right\| \tag{375}$$

$$\leq H(\underline{\mathbf{x}}^r, \underline{\mathbf{y}}^r) + \tilde{\epsilon}^c + \left( \frac{1+\alpha}{m} \right) \tilde{\epsilon}^{2c} + \frac{4}{m} \tilde{\epsilon}^{2c} \left( \left\| \underline{\hat{\mathbf{x}}}^r - \underline{\mathbf{x}}^r \right\| + \alpha \left\| \underline{\hat{\mathbf{y}}}^r - \underline{\mathbf{y}}^r \right\| \right) \tag{376}$$

$$\leq H(\underline{\mathbf{x}}^r, \underline{\mathbf{y}}^r) + \tilde{\epsilon}^c + \left( \frac{2}{m} \right) \tilde{\epsilon}^{2c} + 4\tilde{\epsilon}^c \sqrt{f(\hat{\mathbf{x}}^0) - f^* + 2\mathcal{F}} \tag{377}$$

$$\leq H(\underline{\mathbf{x}}^r, \underline{\mathbf{y}}^r) + \tilde{\epsilon}^{\frac{c}{2}} \tag{378}$$

where third inequality comes from Lemma 21 aND the last inequality is due to $\tilde{\epsilon} \leq \min \left\{ \frac{1}{8}, \left( 4\sqrt{f(\hat{\mathbf{x}}^0) - f^* + 2\mathcal{F}} \right)^{\frac{c}{4}} \right\}$ and $c \geq 4$. Similarly for the lower bound we have

$$g_i^{r,t_{\mathbf{g}}} + \frac{1}{m} \left\| \mathbf{x}^{r,t_{\mathbf{x}}+1} - \underline{\mathbf{x}}^r \right\|^2 + \frac{\alpha}{m} \left\| \mathbf{y}^{r,t_{\mathbf{y}}+1} - \underline{\mathbf{y}}^r \right\|^2$$

$$\geq H(\underline{\mathbf{x}}^r, \underline{\mathbf{y}}^r) - \tilde{\epsilon}^c - \frac{4}{m} \tilde{\epsilon}^{2c} \left( \left\| \underline{\hat{\mathbf{x}}}^r - \underline{\mathbf{x}}^r \right\| + \alpha \left\| \underline{\hat{\mathbf{y}}}^r - \underline{\mathbf{y}}^r \right\| \right) \tag{379}$$

$$\geq H(\underline{\mathbf{x}}^r, \underline{\mathbf{y}}^r) - \tilde{\epsilon}^{\frac{c}{2}} \tag{380}$$

$\square$

From the above lemma and inequalities (369) and (368) we can bound the error of the estimation of the potential function by each node $i$.

**Corollary 12.** *Assume the conditions of Lemma 30 and inequalities* (369) *and* (368) *hold. Then the following bounds characterize the error on the estimation of the potential function after utilizing the average consensus protocol for* $4\frac{\tilde{c}\log(\frac{1}{\tilde{\epsilon}})+\log(\frac{1}{\alpha})+\log\left(4m^2(f(\mathbf{x}^0)-f^*+2\mathcal{F})\right)}{\log(\frac{1}{\sigma})} + 4 + \frac{c\log(\tilde{\epsilon})-\log\left(2\left\|\mathbf{g}^{r,0}-\hat{\mathbf{g}}^r\right\|\right)}{\log \sigma}$ *rounds.*

$$H(\underline{\mathbf{x}}^r, \underline{\mathbf{y}}^r) + \tilde{\epsilon}^{\frac{c}{2}} + \tilde{\epsilon}^c + \alpha\tilde{\epsilon}^c \geq g_i^{r,t_{\mathbf{g}}} + \mathbf{w}_i^{r,t_{\mathbf{w}}} + \alpha\mathbf{z}_i^{r,t_{\mathbf{z}}} \geq H(\underline{\mathbf{x}}^r, \underline{\mathbf{y}}^r) - \tilde{\epsilon}^{\frac{c}{2}} - \tilde{\epsilon}^c - \alpha\tilde{\epsilon}^c \tag{381}$$

*And for* $\tilde{\epsilon} \leq \frac{1}{8}$ *and* $c \geq 4$ *we also have*

$$H(\underline{\mathbf{x}}^r, \underline{\mathbf{y}}^r) + 2\tilde{\epsilon}^{\frac{c}{2}} \geq g_i^{r,t_{\mathbf{g}}} + \mathbf{w}_i^{r,t_{\mathbf{w}}} + \alpha\mathbf{z}_i^{r,t_{\mathbf{z}}} \geq H(\underline{\mathbf{x}}^r, \underline{\mathbf{y}}^r) - 2\tilde{\epsilon}^{\frac{c}{2}} \tag{382}$$

*Further after* $8\frac{\tilde{c}\log(\frac{1}{\tilde{\epsilon}})+\log(\frac{1}{\alpha})+\log\left(4m^2(f(\mathbf{x}^0)-f^*+2\mathcal{F})\right)}{\log(\frac{1}{\sigma})} + 8 + 2\frac{c\log(\tilde{\epsilon})-\log\left(2\left\|\mathbf{g}^{r,0}-\hat{\mathbf{g}}^r\right\|\right)}{\log \sigma}$ *of the consensus protocol on the iteration before the injection of noise and on the iteration at the end of phase two we have*

$$H(\underline{\mathbf{x}}^{-1}, \underline{\mathbf{y}}^{-1}) - H(\underline{\mathbf{x}}^{T_{cap}}, \underline{\mathbf{y}}^{T_{cap}}) + 4\tilde{\epsilon}^{\frac{c}{2}} \geq g_i^{-1,t_{\mathbf{g}}} + \mathbf{w}_i^{-1,t_{\mathbf{w}}} + \alpha\mathbf{z}_i^{-1,t_{\mathbf{z}}} - \left( g_i^{T_{cap},t_{\mathbf{g}}} + \mathbf{w}_i^{T_{cap},t_{\mathbf{w}}} + \alpha\mathbf{z}_i^{T_{cap},t_{\mathbf{z}}} \right) \tag{383}$$

$$H(\underline{\mathbf{x}}^{-1}, \underline{\mathbf{y}}^{-1}) - H(\underline{\mathbf{x}}^{T_{cap}}, \underline{\mathbf{y}}^{T_{cap}}) - 4\tilde{\epsilon}^{\frac{c}{2}} \leq g_i^{-1,t_{\mathbf{g}}} + \mathbf{w}_i^{-1,t_{\mathbf{w}}} + \alpha\mathbf{z}_i^{-1,t_{\mathbf{z}}} - \left( g_i^{T_{cap},t_{\mathbf{g}}} + \mathbf{w}_i^{T_{cap},t_{\mathbf{w}}} + \alpha\mathbf{z}_i^{T_{cap},t_{\mathbf{z}}} \right) \tag{384}$$

Combining all the above we can achieve our third objective

**Theorem 7.** *Assume the conditions of Corollary 12 hold and set* $\tilde{\epsilon}^{\frac{c}{2}} = \frac{\mathcal{F}}{40}$. *After* $8\frac{\tilde{c}\log(\frac{1}{\tilde{\epsilon}})+\log(\frac{1}{\alpha})+\log\left(4m^2(f(\mathbf{x}^0)-f^*+2\mathcal{F})\right)}{\log(\frac{1}{\sigma})} + 9 + 2\frac{c\log(\tilde{\epsilon})-\log\left(2\left\|\mathbf{g}^{r,0}-\hat{\mathbf{g}}^r\right\|\right)}{\log \sigma} + \frac{\log(2m^{\frac{3}{2}})}{\log(\frac{1}{\sigma})}$ *iterations of the consensus protocol the nodes decide whether enough progress has been made in phase II.*

*Proof.* First notice that by setting $\tilde{\epsilon}^{\frac{c}{2}} = \frac{\mathcal{F}}{40}$ the estimation of each node $i$ regarding the potential function decrease is off at most by $\frac{\mathcal{F}}{10}$. Thus the nodes can distinguish between second phases that achieve decrease at least $\mathcal{F}$ and second phases that achieve decrease less than $\frac{\mathcal{F}}{2}$. To do so we utilize Lemma 28 with $ind_i^0 = \mathbb{1}_{\left\{ g_i^{-1,t_\mathbf{g}} + \mathbf{w}_i^{-1,t_\mathbf{w}} + \alpha \mathbf{z}_i^{-1,t_\mathbf{z}} - \left( g_i^{T_{cap},t_\mathbf{g}} + \mathbf{w}_i^{T_{cap},t_\mathbf{w}} + \alpha \mathbf{z}_i^{T_{cap},t_\mathbf{z}} \right) \leq \frac{\mathcal{F}}{2} \right\}}$. Notice that if the potential function decrease in the current phase II is at least $\mathcal{F}$ then all nodes are going to approve and in the case the the current phase II achieves less than $\frac{\mathcal{F}}{2}$ decrease all nodes are going to disapprove. Finally, notice that if the decrease is between $\mathcal{F}$ and $\frac{\mathcal{F}}{2}$ both outcomes are possible; this is acceptable since enough progress have been made in this case as well. $\qquad\square$