[Reviews · NeurIPS 2020]

Review 1

Summary and Contributions: This manuscript proposed a Perturbed Decentralized Gradient Tracking (PDGT) Algorithm that achieves the second-order stationary in polynomial time, by assuming the objective function has first-order and second-order smoothness.

Strengths: The algorithm design and theoretical analysis are new. The gradient tracking technique and the construction of the potential function $H$ is interesting for the decentralized optimization.

Weaknesses: 1. I am concerned about the importance of this result. Since [15] says that perturbed gradient descent is able to find second-order stationary with almost-dimension free (with polylog factors of dimension) polynomial iteration complexity, it is not surprising to me that the decentralized algorithm with occasionally added noise is able to escape saddle point in polynomial time. In addition, the iteration complexity is no longer dimension-free anymore in Theorem 3 (there is a $d$ dependency instead of $log d$). 2. There is no empirical study in this paper. The authors should have constructed some synthetic examples where we know the exact location of saddle points and tried to verify the theoretical claims of the proposed algorithm. Furthermore, PGD [15] should also be compared. I know this is a theory paper, but given the presence of [15], the theoretical contribution is not strong enough from my perspective.

Correctness: To the best of my knowledge, the theoretical claims are correct. There is no empirical study in the paper.

Clarity: Yes, this paper is well written and easy to follow.

Relation to Prior Work: Yes.

Reproducibility: Yes

Additional Feedback: =======POST REBUTTAL======= I appreciate the authors for carefully addressing my concerns. I decide to change my score from 5 to 6.


Review 2

Summary and Contributions: This paper proposed a decentralized algorithm that provably converges to second-order stationary points in nonconvex optimization under standard assumptions. ===== Post-rebuttal edit: Thanks for the response. I like the newly added experiments, please incorporate it into the paper.

Strengths: The paper combines existing algorithmic ideas such as gradient tracking, perturbed gradient descent, consensus, to design an algorithm that converges first to a first-order stationary point, and then a second-order stationary point, with communication complexity bounds. Even the separate algorithmic ideas are not novel, putting them together and carefully balancing the trade-offs are nontrivial, where this work seems to be a solid contribution.

Weaknesses: The proposed algorithm is quite complicated and has many design parameters. It is unclear if which parts of the designs are necessary in practice, and which parts are for theoretical sake. Some numerical experiments would be helpful in verifying the theory and help to guide practice.

Correctness: The methods borrow tools in analyzing convergence in decentralized optimization and nonconvex optimization, and appear to be very solid.

Clarity: The paper reads very well and does a good job explaining the various steps in the algorithm design as well as theoretical bounds.

Relation to Prior Work: The discussions are adequate.

Reproducibility: Yes

Additional Feedback:


Review 3

Summary and Contributions: The paper considers the problem of finding minima for a non-convex objective function in a decentralized setting. In particular, it considers the case where the objective function is a sum of objective functions local to each node and where only adjacent nodes can communicate with each other. To this end, the paper proposes the Perturbed Decentralized Gradient Tracking (PDGT) algorithm, which is comprised of 2 phases: i) a first-order stationary point is found whilst avoiding consensus error; ii) then, noise is used to check for (and escape from) saddle points. Under standard assumptions, a non-asymptotic guarantee on convergence to a second-order stationary point is given, along with an upper bound on the required communication between nodes. Although this problem has been studied before, the theoretical analysis has only focused on asymptotic analysis or non-asymptotic analysis with stronger conditions than are used in this paper.

Strengths: The paper was very thorough in its theoretical analysis. A sensible algorithm was proposed, convergence rates (present in most similar papers) were calculated, and communication requirement bounds were found (not so common in similar papers). The paper was also written very clearly. Finally, the approach was novel and clearly explained role previous literature had played in motivating the proposed algorithm.

Weaknesses: The lack of any experiments was problematic, especially since a new algorithm was proposed and because decentralized machine learning generally has more moving parts and places to go wrong. Indeed, given the care taken in the theoretical analysis, I was a little surprised by this. It’s for this reason that I’ve rated this paper as a borderline reject. This is a good paper that could be published at NeurIPS, but please run this algorithm on a specific problem (even if the problem is just a toy problem), make sure it works as expected, and then put the results in the supplement.

Correctness: The approach is sensible and theoretical results seem to be correct. See above re: empirical methodology.

Clarity: The paper was well-written and easy to follow throughout. There were no obvious typos, notational errors, or formatting problems.

Relation to Prior Work: Papers published on similar topics were mentioned, and the differences between those and this paper were made clear. I would only add that I would like to have seen a longer discussion of why the two assumptions (particularly the first) used in [43] (as referenced in the paper) are overly restrictive given the target problems.

Reproducibility: Yes

Additional Feedback: # Post-rebuttal edit Thank you for adding an empirical investigation to support the good theoretical results. I have changed my score from 5 to 6.

[Author Response · NeurIPS 2020]

We thank the reviewers for their thorough feedback. We address each of their concerns as follows.

@R1, R3, R4 **Numerical experiments:** The main focus of this paper is to extend the "theory" of escaping from saddle points to the decentralized setting. However, following the reviewers' suggestion, we'll provide some experiments in the revised paper. Here, we briefly mention the setup and results and will include more details in the revised paper. Our goal is to show that PDGT is able to escape from saddle points quickly. We compare PDGT with D-GET which is a decentralized gradient tracking method that "doesn't use the perturbation idea" [37]. We focus on a matrix factorization problem for the MovieLens dataset, where the goal is to find a rank $r$ approximation of a matrix $\mathbf{M} \in \mathcal{M}^{l \times n}$, representing the ratings from 943 users to 1682 movies. Each user has rated at least 20 movies for a total of 9990 known ratings. This problem can be written as: $(\mathbf{U}^*, \mathbf{V}^*) := \underset{\mathbf{U} \in \mathcal{M}^{l \times r}, \mathbf{V} \in \mathcal{M}^{n \times r}}{\operatorname{argmin}} f(\mathbf{U}, \mathbf{V}) = \underset{\mathbf{U} \in \mathcal{M}^{l \times r}, \mathbf{V} \in \mathcal{M}^{n \times r}}{\operatorname{argmin}} \|\mathbf{M} - \mathbf{U}\mathbf{V}^\top\|_F^2$.

We consider different values of target rank and number of nodes. Both methods are given the same randomly generated connected graph, mixing matrix, and step size. Further, they are initialized at the same point which lies in the neighborhood of a saddle point. Note that in this problem all saddles are escapable and each local min is a global min.

In Fig. 1 the experiment is run for 10 nodes, and the target rank is 20. Initially both algorithms are stuck close to a saddle point and make very little progress. However, since the theoretical criterion for PDGT is satisfied in the very first rounds (small average gradient and consensus error) we have injection of noise. This nudge is sufficient to accelerate substantially the escape of PDGT. As we see in the plot, D-GET remains close to the saddle point at least until iteration 1400 where we can see the gradient increasing somewhat faster. At the same time PDGT escapes the saddle point, decreases the loss and approaches a local minimum.

In Fig. 2, the experiment is run for 30 nodes and the target rank is 30. Similarly, PDGT escapes from the saddle point much faster and decreases the loss substantially before it reaches the local minimum. We observe that D-GET also escapes the saddle point eventually following a similar trace to PDGT after spending a lot longer at the saddle. Interestingly, for this experiment, we observed that some parameters such as the stepsize of the first and the second phase, the injected noise and the threshold before we inject noise can afford to

Figure 1: Average loss (left) and squared norm of the average gradient (right) vs. iteration (10 nodes and target rank 20).

Figure 2: Average loss (left) and squared norm of the average gradient (right) vs. iteration (30 nodes and target rank 30).

be substantially greater than the theoretical propositions casting PDGT useful for a series of practical applications.

@R1 **Importance of the results considering [15]:** On a high level the PGD method in [15] (designed for centralized optimization) is not applicable to decentralized settings as it requires coordination between all nodes at each iteration which has a prohibitive communication cost. Moreover, notice that PGD is a descent algorithm whose behavior is rather well understood, whereas our proposed PDGT method is a non-monotonically decreasing algorithm which requires the *careful construction and analysis of a potential function* in order to argue about its convergence. Specifically, when the PDG algorithm adds noise to the iterate, there exist a specific number of steps and function decrease threshold that characterize the space topology at the point that the noise was injected. When the same injection happens in a distributed algorithm as PDGT, the consensus error (which potentially increases exponentially fast) and more generally the fact that the algorithm is not strictly descent deems the escape from the saddle point non trivial and the existence of appropriate thresholds unclear. Diving into more technical details, one can observe that in comparison to Lemma 17 in [15], our corresponding Lemma 15 utilizes a new potential function and controls the consensus error among the nodes deriving a novel and more complex analysis. Specifically, the last terms in (148) and (149) correspond to the consensus error of the escaping sequences. To control these term we need to prove an induction with two parts as shown in (151), (152) and invoke novel Lemma 14. It follows that equations (159)-(163) should hold, and an interesting connection is derived between the potential function parameter $\alpha$, the second phase step size $\eta_2$ and the dimension $d$. The aforementioned relation presents a trade-off between the first and second phase stepsizes, $\eta_1$ and $\eta_2$, through $\alpha$ and thus achieving the optimal overall convergence rate requires fine-tuning the stepsizes as well as coming up with the tightest bounds for quantities such as the target decrease of the potential function $\mathcal{F}$, the bound on the norm of the iterates $P$ and the radius of the noise ball $\mathcal{R}$. All these values in our analysis are different from the ones in [15] and our main proofs are more complex and technical confronting new challenges presented in the distributed framework. Finally, the polynomial dependence on $d$ appears to be crucial in controlling the consensus error after the injection of noise. To be more precise, notice that to prove the base of the induction in (151) we need to lower bound $\zeta\psi_0 = \zeta\mathcal{R}\mu$ where $\mu \in \left[\frac{\delta_2}{2\sqrt{d}}, 1\right]$, deriving the dependence on dimension. The importance of $\mu$ belonging in this interval becomes apparent in Lemma 16 and the corresponding Lemma 15 in [15], where the volume of the stuck region is bounded.

[Meta-Review · NeurIPS 2020]

This paper gives a decentralized non-convex optimization algorithm that is guaranteed to find a second order stationary point. The paper deals with consensus issues that may arise in decentralized settings while maintaining interesting theoretical guarantees. The response also provided interesting experiment results. The paper would be stronger if it can improve the d dependency in phase II or explain why a dimension dependency is necessary.